# Mitigating Participation Imbalance Bias in Asynchronous Federated Learning Under Client Heterogeneity

## Abstract

In Asynchronous Federated Learning (AFL), the central server immediately updates the global model with each arriving client's contribution. As a result, clients perform their local training on different model versions, causing information staleness (delay). In federated environments with non-IID local data distributions, this asynchronous pattern amplifies the adverse effect of client heterogeneity (due to different data distribution, local objectives, etc.), as faster clients contribute more frequent updates, biasing the global model. We term this phenomenon **heterogeneity amplification**. Our work provides a theoretical analysis that maps AFL design choices to their resulting error sources when heterogeneity amplification occurs. Guided by our analysis, we propose **ACE** (All-Client Engagement AFL), which mitigates participation imbalance through immediate, non-buffered updates that use the latest information available from *all* clients. We also introduce a delay-aware variant, **ACED**, to balance client diversity against update staleness. Experiments on different models for different tasks across diverse heterogeneity and delay settings validate our analysis and demonstrate the robust performance of our approaches.

## 1 Introduction

Federated Learning (FL) enables collaborative training of machine learning models across multiple clients (e.g., mobile devices) holding private data (Kairouz et al., 2021). In a typical FL process coordinated by a central server, clients receive the current global model, compute updates based on their local data, and send these updates back. The server aggregates these updates to refine the global model for the next round, keeping raw data local. A key challenge in FL is **client heterogeneity**: clients often have diverse characteristics, including non-IID local data distributions and potentially distinct local objectives or update computation processes. These variations can impact training speed and performance (Li et al., 2020; Kairouz et al., 2021). Another challenge is the presence of **stragglers**: synchronous FL algorithms, like FedAvg (Li et al., 2020), wait for a subset of clients to finish, creating bottlenecks from slower clients.

To address the straggler problem and reduce waiting times, **Asynchronous Federated Learning (AFL)** was proposed (Agarwal & Duchi, 2011; Recht et al., 2011; Nguyen et al., 2022). In AFL, the server incorporates each of the client updates immediately upon receipt without waiting. However, this solution introduces **update delays (staleness)** because slower clients compute updates locally based on older versions of the global model received earlier, while the server continues to evolve using updates from faster clients. This participation imbalance causes the global model to be *more* influenced by the data distributions and learning objectives of the faster clients. We formally define this phenomenon as **heterogeneity amplification** and provide a theoretical analysis to understand its impact on asynchronous FL. Specifically, our analysis shows that the challenges of AFL originate from two interconnected issues:

- **AFL Staleness and Dynamics:** The asynchronous nature of AFL results in widely varying client-server communication intervals. This variability leads to information staleness where gradients are computed on outdated models, introducing errors (Agarwal & Duchi, 2011). Additionally, updates formed from a subset of clients can introduce participation imbalance bias into the global model.

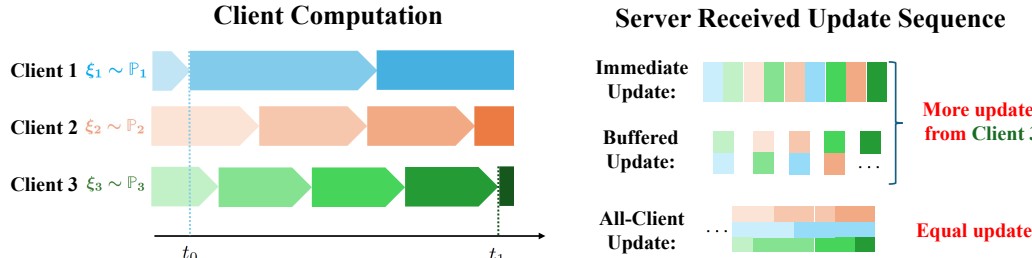

Figure 1: Staleness and Heterogeneity Amplification in AFL. **Left:** Clients compute at varying speeds (arrow lengths) on their local datasets with heterogeneous data distributions ($\mathbb{P}_i$, colors). Color intensity reflects staleness—the degree to which a client's model version is outdated due to infrequent client-server communication. **Right:** Update sequences (during $t_0$ to $t_1$): 'Immediate Update' applies client updates on arrival; 'Buffered Update' waits and aggregates multiple clients' updates before applying. However, both strategies demonstrate **heterogeneity amplification**: faster clients (e.g., Client 3) contribute more frequently, resulting in their imbalanced influence. In contrast, the 'All-Client Update' strategy aims to balance updates (despite staleness) from all the clients and thereby mitigate heterogeneity amplification.

- **Heterogeneity Amplification:** The interaction between client heterogeneity (including non-IID data distributions, different local objectives) and the dynamics of the AFL system (varying communication frequencies, partial participation) leads to faster and more frequent contributing clients having a greater influence on the global model, as shown in Figure 1. This affects convergence and degrades performance (Wang et al., 2024b; Koloskova et al., 2022).

Addressing these challenges requires a fundamental understanding of heterogeneity amplification, which can help mitigate its impact on convergence. To this end, we make the following contributions.

- **Theoretical Framework and Algorithm Design (Section 3).** We provide a theoretical framework that analyzes heterogeneity amplification by decomposing the *discrepancy between the server's aggregated update and the ideal gradient*. This connects AFL design choices to the resulting error and motivates our proposed algorithm, **ACE (All-Client Engagement AFL)**. It realizes an *all-client* aggregation through a *non-buffered, immediate update* to eliminate participation imbalance bias. We also introduce a practical delay-aware variant ACED, to handle clients with extreme delays by managing the trade-off between client diversity and update staleness.
- **Comparative Theoretical Analysis (Section 4).** Using our framework, we comparatively analyze ACE against recent AFL methods (FedBuff (Nguyen et al., 2022), CA$^2$FL (Wang et al., 2024b), Delay-adaptive ASGD (Koloskova et al., 2022), Vanilla ASGD (Mishchenko et al., 2022)). We show how its *all-client* design eliminates participation imbalance bias and mitigates the delay-heterogeneity interaction, resulting in a convergence rate robust to arbitrary heterogeneity (Theorem 1). In parallel, its *non-buffered, immediate update* mechanism improves communication efficiency and leads to faster convergence (Appendix E).
- **Experimental Validation (Section 5 and Appendix F)** We validate our findings through extensive experiments against the aforementioned baselines. Results across various models and tasks (Fig. 2, Fig. a.2, Table a.2) demonstrate ACE's robustly faster convergence and higher final accuracy, particularly under the challenging conditions of high client heterogeneity and high delay.

Overall, we provide a novel theoretical framework that provides valuable insights for mitigating biases in AFL. This guides our ACE algorithm which uses all-client aggregation for robust, communication-efficient convergence under heterogeneity. Our practical ACED variant manages the trade-off between client diversity and update staleness, and experiments validate our methods.

## 2 RELATED WORK

**Asynchronous FL and its Challenges.** Asynchronous federated learning (AFL) (Agarwal & Duchi, 2011; Recht et al., 2011) enhances training efficiency in large-scale distributed learning by eliminating costly synchronization steps in synchronous protocols like FedAvg (Li et al., 2020). While effective in reducing wall-clock time, especially in the presence of slow or straggling clients, asynchronicity

changes the learning dynamics and introduces several challenges (Lian et al., 2015). Beyond foundational FL challenges like *client heterogeneity* (Kairouz et al., 2021; Li et al., 2020) and *stochastic noise* (Bottou et al., 2018), AFL introduces the critical issue of *update staleness*. This problem arises as faster clients continuously update the server's global model, causing slower clients to compute gradients on outdated model versions. This imbalance in update frequency not only leads to *model staleness* but also reduces the influence of slower clients on the global model. When client data is non-IID, this dynamic gives faster clients a dominant influence, biasing the global model towards their local data distributions. Prior work(Wang et al., 2024b; Koloskova et al., 2022) has observed performance degradation in experiments under high heterogeneity and delay, but their theoretical analyses were derived on an algorithm-specific basis. These analyses reveal that the convergence guarantees of methods like FedBuff (Nguyen et al., 2022) and CA$^2$FL (Wang et al., 2024b) depend on the degree of client data heterogeneity, but a theoretical framework to analyze the delay-heterogeneity interaction and guide algorithm design was missing. We are the first to formally define this interaction as **heterogeneity amplification** and provide a theoretical analysis that identifies its cause in *partial client participation*, a common design choice in many AFL algorithms.

**Mitigation Strategies.** Given AFL's challenges, particularly client heterogeneity amplification, various mitigation strategies have been explored, focusing on different aspects of the problem.

First, some strategies, often adapted from synchronous FL, target client drift using methods like regularization (Li et al., 2020) (Acar et al., 2021)) or control variates (Karimireddy et al., 2020). However, the full participation assumption of methods such as SCAFFOLD (Karimireddy et al., 2020) only works in a limited number of scenarios where the server can actively control the queuing dynamics of the AFL system.

Second, other strategies directly address the impact of *model delays (staleness)*. These include adaptive step-sizing based on delay magnitude (Koloskova et al., 2022; Cohen et al., 2021; Aviv et al., 2021) and error feedback (Zheng et al., 2017; Stich & Karimireddy, 2020). While improving stability with stale updates, reacting primarily to delay magnitude does not always resolve the imbalanced client influence if heterogeneity amplification causes faster clients to dominate the update.

Third, aggregation strategies involving *state caching* and *buffering* have been explored to mitigate participation variance. One line of work utilizes client state caching to reuse historical gradients, such as MIFA (Gu et al., 2021) and FedVARP (Jhunjhunwala et al., 2022). While sharing the high-level concept of state reuse, these methods typically operate within synchronous or round-based protocols that impose synchronization barriers. Furthermore, their theoretical analysis generally focuses on proving the sufficiency of a heuristic algorithm, rather than deriving the necessary design conditions to eliminate bias from first principles. Another direction employs buffering or calibration, as seen in FedBuff (Nguyen et al., 2022) and CA$^2$FL (Wang et al., 2024b). While FedBuff simply aggregates updates from a subset, CA$^2$FL attempts to calibrate a cached all-client state using updates from a subset ($m < n$). However, as detailed in Appendix F.1.2, this calibration mechanism imposes non-uniform weighting on client updates, which structurally retains the participation imbalance bias and heterogeneity amplification. In contrast, our work establishes a prescriptive framework for truly asynchronous, non-buffered systems. We derive that aggregating updates from all clients ($m = n$) with equal weighting is a necessary condition to eliminate the participation bias term. Guided by this, our ACE algorithm maintains a server-side cache of the latest gradients from all clients and performs an immediate global update upon every single client arrival, thereby eliminating bias without reducing update frequency or enforcing synchronization.

## 3 PRELIMINARIES AND ANALYTICAL FRAMEWORK

Asynchronous Federated Learning (AFL) is designed to enhance system efficiency by allowing clients to operate without waiting for slower participants (stragglers). This section establishes the notations, problem setting, key assumptions for our analysis, and the analytical foundation motivating our method, ACE. More details can be found in Appendix A.

### 3.1 PROBLEM SETTING AND NOTATIONS

We consider $n$ clients orchestrated by a central server, minimizing a global objective $F(w) = \frac{1}{n} \sum_{i=1}^{n} F_i(w)$, where each $F_i(w) = \mathbb{E}_{\xi_i \sim \mathbb{P}_i}[f_i(w; \xi_i)]$ is the expected loss over client $i$'s true local data distribution $\mathbb{P}_i$. Clients compute stochastic gradients $\nabla f_i(w; \xi_i)$ from samples $\xi_i \sim \mathbb{P}_i$ as approximations to $\nabla F_i(w)$. The server maintains the global model $w^t \in \mathbb{R}^d$ at server iteration $t$ (up

to $T$ total iterations). In asynchronous settings, $w^{t+1} = w^t - \eta u^t$. The global update $u^t$ typically uses stale information; a contribution from client $i$ may be based on a model $w^{t-\tau_i^t}$, where $\tau_i^t \geq 0$ is its information staleness (delay) relative to $t$ (in server iterations).

We define: $\mathbb{E}[\cdot]$ as the total expectation over all sources of randomness (e.g., client data sampling, $\tau_i^t$ values). $\mathcal{F}^t$ is the $\sigma$-algebra of all information up to server iteration $t$ (including $w^t$). $\mathbb{E}_t[\cdot] := \mathbb{E}[\cdot|\mathcal{F}^t]$ is the conditional expectation given $\mathcal{F}^t$. The global update $u^t$ is formed from clients' stochastic contributions, where each client $i$ uses a fresh data sample $\xi_i$ with its respective stale model (e.g., $w^{t-\tau_i^t}$). Let $\mathcal{H}^t$ be the $\sigma$-algebra containing all information determining $u^t$ (including stale models used and aggregation rules) except the randomness from the set of these fresh data samples $\{\xi_i\}$ contributing to $u^t$. Then, $\bar{u}^t := \mathbb{E}_{\{\xi_i\}}[u^t|\mathcal{H}^t]$ is the expected update over these fresh samples.

## 3.2 ASSUMPTIONS

Our subsequent theoretical analysis relies on several standard assumptions common in the optimization literature (Stich & Karimireddy, 2020; Wang et al., 2024b; Nguyen et al., 2022), particularly for stochastic and asynchronous methods. We assume the following hold throughout the paper unless otherwise stated:

**Assumption 1** (Lower Boundedness). *The global objective function $F(w)$ (an expectation over true local distributions) is bounded below, i.e., $F(w) \geq F^* > -\infty$ for all $w \in \mathbb{R}^d$.*

**Assumption 2** (L-Smoothness). *Each local objective function $F_i(w)$ (an expectation) is L-smooth for some $L \geq 0$, implying $\|\nabla F_i(w) - \nabla F_i(w')\|_2 \leq L\|w - w'\|_2$ for all $w, w'$. This also implies $F(w)$ is L-smooth.*

**Assumption 3** (Unbiased Stochastic Gradients). *Given $t \leq t_2 < t_3$, let $\xi_i^{t_3}$ with $i \in [n]$ and $t_3 \geq 1$ be data sample drawn from $\mathbb{P}_i$, and $\mathcal{F}_{t_2}$ be the $\sigma$-algebra representing all information available up to server iteration $t_2$, then $\mathbb{E}\left[\nabla f_i(w^t; \xi_i^{t_3}) \mid \mathcal{F}_{t_2}\right] = \nabla F_i(w^t)$.*

**Assumption 4** (Bounded Sampling Noise). *The sampling noise of the local stochastic gradients is uniformly bounded: $\mathbb{E}_{\xi_i}\|\nabla f_i(w; \xi_i) - \nabla F_i(w)\|_2^2 \leq \sigma^2$ for some $\sigma \geq 0$.*

**Assumption 5** (Bounded Delay). *$\forall i, t : \tau_i^t \leq \tau_{\max}$, where $\tau_{\max}$ bounds the maximum interval between server iterations for any two consecutive global model updates triggered by any client $i$.*

Assumptions 1-4 characterize the optimization problem, and Assumption 5 constrains staleness. Beyond these, some analyses for algorithms with partial client participation (e.g., FedBuff (Nguyen et al., 2022)) or single-client updates (e.g., Vanilla ASGD (Mishchenko et al., 2022)) also assume **Bounded Data Heterogeneity (BDH)** , i.e., $\|\nabla F_i(w) - \nabla F(w)\|^2 \leq \zeta^2 < \infty$, which bounds how much any single client's local gradient can diverge from the true global gradient. This bound is required to analyze convergence in partial participation settings, as it controls the bias from averaging over a non-representative subset of clients. Our method, ACE, by employing full aggregation, is designed to eliminate the participation imbalance bias (see Section 3.3) from partial client participation, thereby eliminating the need for the BDH assumption in its convergence analysis.

## 3.3 THEORETICAL MOTIVATION FOR ACE: AN MSE DECOMPOSITION

In AFL, clients compute updates based on stale model versions. Client $i$ might use $w^{t-\tau_i^t}$ (where $\tau_i^t \geq 0$ is its information delay relative to server iteration $t$), while the server is at $w^t$. We denote the collection of stale models used by clients as $w_{\text{stale}}^t = \{w^{t-\tau_i^t}\}_{i=1}^n$. This presents a critical challenge: since the latest model versions available to clients for their local computations are at best these stale versions, any global gradient estimate $u^t$ (formed from their contributions) is inherently based on this outdated information when aiming to approximate $\nabla F(w^t)$. This creates a gap between the information used for client updates and the ideal current gradient at the server.

Our analysis starts from the standard descent lemma (details in Appendix B.1) for $L$-smooth functions (Assumption 2)). For an update $w^{t+1} = w^t - \eta u^t$, this lemma bounds the change in the objective as:

$$\mathbb{E}[F(w^{t+1})] \leq \mathbb{E}[F(w^t)] - \eta\mathbb{E}[\langle\nabla F(w^t), u^t\rangle] + \frac{L\eta^2}{2}\mathbb{E}\|u^t\|^2 \tag{1}$$

By summing this inequality over $T$ iterations and rearranging terms, we derive the following bound (2). This inequality bounds the average squared norm of the true gradient, a standard measure of conver-

gence for non-convex objectives, by terms including the Mean Squared Error (MSE) of our gradient estimates (details of the constants $\gamma_1, \gamma_2 > 0$ are in Appendix B.2):

$$\frac{1}{T}\sum_{t=0}^{T-1}\mathbb{E}\|\nabla F(w^t)\|^2 \leq \frac{\gamma_1(F(w^0) - \mathbb{E}[F(w^T)])}{T\eta} + \gamma_2\eta\left(\frac{1}{T}\sum_{t=0}^{T-1}\underbrace{\mathbb{E}\|u^t - \nabla F(w^t)\|_2^2}_{\text{MSE}_t}\right) \quad (2)$$

Bound (2) is important because its left-hand side, the average squared gradient norm, diminishes as an algorithm converges to a stationary point in non-convex optimization (Wang et al., 2024b; Mishchenko et al., 2022; Nguyen et al., 2022). It indicates that this convergence metric is upper-bounded by the average $\text{MSE}_t$. Therefore, controlling $\text{MSE}_t$ of the gradient estimates is key to improving the convergence guarantee, motivating its detailed analysis for algorithmic design.

**Decomposing the $\text{MSE}_t$ term.** To analyze the error sources contributing to $\text{MSE}_t$, we decompose the error $u^t - \nabla F(w^t)$. We introduce $\bar{u}^t$ (the expectation of $u^t$ over data sampling randomness, as defined in Section 3.1) and $\nabla F(w_{\text{stale}}^t) = \frac{1}{n}\sum_{i=1}^n \nabla F_i(w^{t-\tau_i^t})$ (the average true gradient on the latest stale models actually used by clients). With a telescoping sum, the error is decomposed as[1]:

$$u^t - \nabla F(w^t) = \underbrace{(u^t - \bar{u}^t)}_{:=A,\text{ Noise}} + \underbrace{(\bar{u}^t - \nabla F(w_{\text{stale}}^t))}_{:=B,\text{ Bias}} + \underbrace{(\nabla F(w_{\text{stale}}^t) - \nabla F(w^t))}_{:=C,\text{ Delay}} \quad (3)$$

Using the inequality $\|x + y + z\|^2 \leq 3(\|x\|^2 + \|y\|^2 + \|z\|^2)$ (an application of Lemma a.3 in Appendix B.1), we bound $\text{MSE}_t$ with the decomposition:

$$\text{MSE}_t = \mathbb{E}\|u^t - \nabla F(w^t)\|_2^2 \leq 3\mathbb{E}\|A\|_2^2 + 3\mathbb{E}\|B\|_2^2 + 3\mathbb{E}\|C\|_2^2 \quad (4)$$

Now let's analyze each error component (further discussions can be found in Section 4):

- **Term A (Sampling Noise):** $A = u^t - \bar{u}^t$, represents the stochastic error from using mini-batch gradient approximations (see Assumption 4). It depends on factors like the number of participating clients and the structure of $u^t$.
- **Term B (Bias Error):** $B = \bar{u}^t - \nabla F(w_{\text{stale}}^t)$, quantifies the deviation of the expected gradient estimate $\bar{u}^t$ from the true average gradient evaluated at the specific stale states $w^{t-\tau_i^t}$ used by the clients. This bias can arise from partial client participation in forming $u^t$ or from local training steps if the clients optimize local objectives.
- **Term C (Delay Error):** $C = \nabla F(w_{\text{stale}}^t) - \nabla F(w^t)$, captures the discrepancy between the average gradient on stale models and the gradient on the current server model. It generally grows with longer delays $\tau_i^t$.

This specific structure (A: Noise, B: Bias, C: Delay) helps isolate error sources relevant to AFL algorithm design. Noise (Term A) and Delay (Term C) are inherent to asynchronous optimization. In contrast, Bias Error (Term B), arises from a specific design choice: partial client participation. We therefore target Term B for complete elimination. The condition $B \equiv 0$ mathematically necessitates an **all-client aggregation** scheme. This principled design not only eliminates the primary source of heterogeneity amplification but also maximally reduces Noise (Term A) and helps contain Delay (Term C) by preventing bias accumulation in model drift (quantified in Section 4).

**Algorithm Design.** To achieve $B \equiv 0$ (Bias Error elimination), it requires $\bar{u}^t \equiv \nabla F(w_{\text{stale}}^t) = \frac{1}{n}\sum_{i=1}^n \nabla F_i(w^{t-\tau_i^t})$. Our *all-client aggregation* design for $u^t$ is $u^t := \frac{1}{n}\sum_{i=1}^n \nabla f_i(w^{t-\tau_i^t}; \xi_i^{\kappa_i})$. Here, for each client $i$ in the sum, $w^{t-\tau_i^t}$ is the stale model version upon which its currently cached gradient was computed. The superscript $\kappa_i$ on the sample $\xi_i^{\kappa_i}$ signifies that this specific sample was used by client $i$ to generate the gradient that was received and cached by the server at server iteration $\kappa_i$ (where $t - \tau_i^t < \kappa_i \leq t$). This sample $\xi_i^{\kappa_i}$ was drawn by client $i$ at the time of its local computation on $w^{t-\tau_i^t}$. Given Assumption 3, taking the expectation of $u^t$ over these respective fresh samples $\{\xi_i^{\kappa_i}\}$ yields $\bar{u}^t = \frac{1}{n}\sum_{i=1}^n \nabla F_i(w^{t-\tau_i^t}) = \nabla F(w_{\text{stale}}^t)$.

---

[1]This framework is extensible to other scenarios. For instance, by applying further telescoping sums, the Bias term $B$ can be decomposed to isolate *adversarial bias* (e.g., $\bar{u}^t - \bar{u}_{\text{honest}}^t$) in Byzantine settings, or the Noise term $A$ can be expanded to model errors from gradient compression.

This leads to our **ACE (All-Client Engagement AFL)** algorithm's core update principle Eq. 5, where this Bias Error (Term B) is eliminated:

$$\boxed{\text{Bias Error (Term B)} = 0 \iff u^t := \frac{1}{n} \sum_{i=1}^{n} \nabla f_i(w^{t-\tau_i^t}; \xi_i^{\kappa_i})} \tag{5}$$

By employing full aggregation, ACE aims to directly eliminate a key component of heterogeneity amplification related to imbalanced client influence from partial updates, potentially leading to a tighter convergence bound.

### 3.4 ACE ALGORITHM: CONCEPTUAL AND PRACTICAL VARIANTS

**(1) ACE Conceptual Implementation. (Algorithm 1)** The ACE algorithm primarily targets the Bias Error (Term B in InEq. 3) via a *full aggregation* strategy. In its main conceptual form (direct aggregation), the server computes the global update $u^t$ by averaging the latest available gradients $U_i^t$ from all $n$ clients:

$$u^t := \frac{1}{n} \sum_{i=1}^{n} U_i^t \in \mathbb{R}^d, \quad \text{where} \quad U_i^t = \nabla f_i(w^{t-\tau_i^t}; \xi_i) \tag{6}$$

Here, $U_i^t$ is the most recent gradient from client $i$, computed on its stale model $w^{t-\tau_i^t}$ using a fresh data sample $\xi_i$. To eliminate the participation bias from our analysis in Section 3.3, this method stores all $n$ gradients for an immediate update on each arrival. This offers higher communication efficiency than buffered methods (Nguyen et al., 2022; Wang et al., 2024b), which require similar storage but must wait for a buffer to fill. An alternative, efficient computation of $u^t$ for ACE uses an incremental rule (Algorithm a.5), $u^t = u^{t-1} + (u_{j_t}^{\text{new}} - u_{j_t}^{\text{prev}})/n$, and can reduce the server's cost from $\mathcal{O}(nd)$ to $\mathcal{O}(d)$ by distributing the overhead to clients. Appendix F.3.3 further explores a compression scheme to reduce the *total* system cost. For clarity, Algorithm 1 details only the direct aggregation method.

---

**Algorithm 1** Conceptual ACE (Direct Aggregation, Incremental Rule see Algorithm a.5)

---

1: **Server Initialization:**
2:     Initialize global model $w^0$.
3:     For each client $i \in [n]$: $U_i^{\text{cache}} \leftarrow \nabla f_i(w^0; \xi_i^0)$.    ▷ Initial gradients based on $w^0$, forming $u^0$
4:     $u^0 \leftarrow \frac{1}{n} \sum_{i=1}^{n} U_i^{\text{cache}}$.
5:     $w^1 \leftarrow w^0 - \eta u^0$.
6:     Server makes $w^1$ available to clients.
7: **Server Loop:** For $t = 1, \ldots, T-1$:                              ▷ To compute $u^t$ and model $w^{t+1}$
8:     Wait to receive a gradient $g_j$ from some client $j$. ▷ $g_j = \nabla f_j(w^{t-\tau_j^t}; \xi_j^t)$, where $w^{t-\tau_j^t}$ is the model client $j$ used and $\xi_j^t$ is its fresh sample for this contribution to $u^t$.
9:     Update server's cache for client $j$: $U_j^{\text{cache}} \leftarrow g_j$.
10:     Compute global update: $u^t \leftarrow \frac{1}{n} \sum_{i=1}^{n} U_i^{\text{cache}}$.    ▷ Uses latest $g_j$, cached $U_i^{\text{cache}}$ from others
11:     Update global model: $w^{t+1} \leftarrow w^t - \eta u^t$.
12:     Server makes $w^{t+1}$ available (e.g., to client $j$).
13: **Client $i$ Operation (runs continuously):**
14:     $w_{\text{local}} \leftarrow$ latest model version received from server.
15:     Compute gradient $g_i = \nabla f_i(w_{\text{local}}; \xi_i^{\text{new}})$.                              ▷ $\xi_i^{\text{new}}$ is a fresh sample
16:     Send $g_i$ to server.

---

**(2) Practical Variant: ACED (All-Client Engagement Bounded Delay-Aware AFL).** The conceptual ACE assumes bounded delays (Assumption 5) and active participation from all clients to ensure Term B elimination. However, this strict assumption becomes impractical in real-world scenarios with client dropouts or extreme delays.

To address this, ACED enforces a delay threshold $\tau_{\text{algo}}$ for including gradients in aggregation. The server caches the latest gradient $U_i^{\text{cache}}$ from each client and its model's dispatch time $t_i^{\text{start}}$. At iteration $t$, the active set $A(t) = \{i \in [n] \mid t - t_i^{\text{start}} \leq \tau_{\text{algo}}\}$ includes clients with sufficiently fresh

information. If $A(t)$ is non-empty ($n_t = |A(t)| > 0$), the server does a *bounded delay-aware* update:

$$u_{\text{BDA}}^t := \frac{1}{n_t} \sum_{i \in A(t)} U_i^{\text{cache}}, \quad \text{where} \quad A(t) = \{i \in [n] \mid t - t_i^{\text{start}} \leq \tau_{\text{algo}}\} \tag{7}$$

The model update is $w^{t+1} = w^t - \eta u_{\text{ACED}}^t$. This allows clients to rejoin $A(t)$ upon providing fresh updates. Algorithm a.1 (see Appendix D) details this. However, it is worth noting that if $n_t < n$, Term B may not be fully eliminated, and this variant needs separate convergence analysis. Due to the limited space, further discussion on ACED can be found in Appendix D.

# 4 THEORETICAL COMPARISON OF AFL ALGORITHMS

We apply our MSE decomposition (InEq. 4 from Section 3) to analyze the **Sampling Noise** ($\mathbb{E}\|A\|^2$), **Bias Error** ($\mathbb{E}\|B\|^2$), and **Delay Error** ($\mathbb{E}\|C\|^2$) for representative AFL algorithms (details for these baseline algorithms can be found in Appendix F.1). This analysis relies on Assumptions in Section 3.2. Key notation includes $\zeta^2$ for bounded data heterogeneity (if an algorithm assumes it), the set of participating clients $\mathcal{M}_t$ of size $m = |\mathcal{M}_t|$, the number of local steps $K$, and local learning rate $\eta_l$. We denote a weighted sum of $\{X_i\}_{i=1}^n$ as $\sum_i \overline{X_i}$ (detailed weights omitted), and $X \lesssim Y + Z$ to signify $X \leq aY + bZ$ for some constants $a, b > 0$.

**Term A: Sampling Noise Analysis** ($\mathbb{E}\|A\|^2 = \mathbb{E}\|u^t - \bar{u}^t\|_2^2$): This term reflects the variance from stochastic gradient estimation using mini-batches. Aggregation over more clients reduces this noise, while multiple local steps can accumulate it. (Details in Appendix B.3.)

- **Vanilla ASGD(Mishchenko et al., 2022) & Delay-Adaptive ASGD(Koloskova et al., 2022)** (single client update, $m = 1, K = 1$): $\mathbb{E}\|A\|^2 \leq \sigma^2$. With $m = 1$, there is no noise reduction from aggregation.
- **FedBuff(Nguyen et al., 2022) & CA²FL(Wang et al., 2024b)** (subset $m < n$ clients, local steps $K \geq 1$):$\mathbb{E}\|A\|^2 \lesssim \frac{K\eta_l^2}{m}\sigma^2$. Noise variance is reduced by averaging over $m$ clients but scales with local steps $K$ and the local learning rate $\eta_l$.
- **ACE (Ours)** (full aggregation over $n$ clients, $K = 1$): $\mathbb{E}\|A\|^2 \leq \frac{\sigma^2}{n}$. This achieves maximal sampling noise reduction by averaging across all $n$ clients.

**Term B: Bias Error Analysis** ($\mathbb{E}\|B\|^2 = \mathbb{E}\|\bar{u}^t - \nabla F(w_{\text{stale}}^t)\|_2^2$): This term measures the systematic deviation of the conditionally expected update $\bar{u}^t = \mathbb{E}_\xi[u^t|\mathcal{H}_t]$ from $\nabla F(w_{\text{stale}}^t)$, the ideal average gradient on the actual stale models clients used. Such bias primarily arises if $u^t$ is constructed using only a subset of clients ($m < n$) or involves multiple local steps ($K \geq 1$) that optimize divergent local objectives. (Details in Appendix B.4.)

- **Vanilla ASGD(Mishchenko et al., 2022) & Delay-Adaptive ASGD(Koloskova et al., 2022) & FedBuff(Nguyen et al., 2022)** ($m < n, K \geq 1$):

$$\mathbb{E}\|B\|^2 \lesssim \left( (\sigma^2 + K\zeta^2) + \sum_{i \in \mathcal{M}_t} \mathbb{E}\|\nabla F(w^{t-\tau_i^t})\|_2^2 \right) + (n - m)\left( \zeta^2 + \sum_{i=1}^n \overline{\mathbb{E}\|\nabla F(w^{t-\tau_i^t})\|_2^2} \right)$$

This suffers from both local steps ($K \geq 1$) and partial client participation ($m < n$).
- **CA²FL(Wang et al., 2024b)** ($m < n, K \geq 1$):

$$\mathbb{E}\|B\|^2 \lesssim \left( 1 + (1 - \frac{m}{n})^2 \right) \left( (\sigma^2 + K\zeta^2) + \sum_{i=1}^n \overline{\mathbb{E}\|\nabla F(w^{t-\tau_i^t})\|_2^2} \right).$$

Calibration reduces the partial participation component of bias, but bias related to the number of local steps ($K$) and imperfect calibration ($m < n$) persists.
- **ACE (Ours)** ($n$ clients, $K = 1$): By its design $u^t = \frac{1}{n}\sum \nabla f_i(w^{t-\tau_i^t}; \xi_i^{\kappa_i})$, Assumption 3 implies $\bar{u}^t = \frac{1}{n}\sum \nabla F_i(w^{t-\tau_i^t}) = \nabla F(w_{\text{stale}}^t)$. Therefore, $\mathbb{E}\|B\|^2 = 0$. This design eliminates this bias term by ensuring full aggregation and $K = 1$.

**Term C: Delay Error Analysis** ($\mathbb{E}\|C\|^2 = \mathbb{E}\|\nabla F(w_{\text{stale}}^t) - \nabla F(w^t)\|_2^2$): This term captures the error from using stale model versions. It is bounded by the average model drift clients experience, $D_i^t := \mathbb{E}\|w^{t-\tau_i^t} - w^t\|_2^2$, which measures how much the global model $w^t$ has changed during client

$i$'s effective delay interval $\tau_i^t$. Using Assumption 2 and Lemma a.3 in Appendix B.1:

$$\mathbb{E}\|C\|^2 = \mathbb{E}\left\|\frac{1}{n}\sum_{i=1}^n (\nabla F_i(w^{t-\tau_i^t}) - \nabla F_i(w^t))\right\|_2^2 \leq \frac{1}{n}\sum_{i=1}^n \mathbb{E}\|\nabla F_i(w^{t-\tau_i^t}) - \nabla F_i(w^t)\|_2^2 \leq \frac{L^2}{n}\sum_{i=1}^n D_i^t.$$

The bound on model drift $D_i^t = \mathbb{E}\|\sum_{s=t-\tau_i^t}^{t-1}\eta u^s\|_2^2$ (where $u^s$ is the server update at iteration $s$) highlights how different algorithm designs influence this drift: (Details in Appendix B.5.)

- **Vanilla ASGD(Mishchenko et al., 2022), Delay-Adaptive ASGD(Koloskova et al., 2022), FedBuff(Nguyen et al., 2022)** ($u^t$ from subset $\mathcal{M}_t$ of size $m \leq n$, local steps $K \geq 1$):

$$D_i^t \lesssim \tau_i^t \eta^2 \eta_l^2 \left(\frac{K\sigma^2}{m} + \frac{1}{m}\sum_{s'=t-\tau_i^t}^{t-1}\mathbb{E}\|\nabla F(w_{\text{stale}}^{s'})\|_2^2 + (n-m)K^2\zeta^2\right).$$

  The term $(n-m)K^2\zeta^2$ represents a per-iteration bias arising from partial participation ($m < n$), local steps ($K$), and client heterogeneity ($\zeta^2$). Its multiplication by $\tau_i^t$ illustrates how this bias accumulates. This $\tau\zeta^2$ interaction term is the direct mathematical representation of **heterogeneity amplification**.
- **CA$^2$FL(Wang et al., 2024b)** ($u^t$ from subset $\mathcal{M}_s$, local steps $K \geq 1$):

$$D_i^t \lesssim \tau_i^t \eta^2 \eta_l^2 \left(1 + (1-\frac{m}{n})^2\right)\left(\frac{K\sigma^2}{m} + \sum_{s'=t-\tau_i^t}^{t-1}\overline{\mathbb{E}\|\nabla F(w_{\text{stale}}^{s'})\|_2^2}\right).$$

  Calibration aims to remove the direct $\zeta^2$ term from partial participation bias found in FedBuff's drift, though effects of $K$ and incomplete calibration ($m < n$) remain.
- **ACE (Ours)** ($u^t$ averages over all $n$ clients, $K = 1$):

$$D_i^t \lesssim \tau_i^t \eta^2 \left(\frac{\sigma^2}{n} + \sum_{s'=t-\tau_i^t}^{t-1}\mathbb{E}\|\nabla F(w_{\text{stale}}^{s'})\|_2^2\right).$$

  Here, the $\zeta^2$ term from partial participation is absent because $u^t$ in ACE averages information from all $n$ clients, inherently balancing expected contributions during the drift calculation.

**Comparative Insights.** The impact of algorithmic design choices on the error terms is summarized in Table 1. (Green text indicates a positive impact, red a negative one). This comparison highlights:

- The number of participating clients $m$ affects Noise (Term A, reduced by larger $m$) and Bias (Term B, introduced if $m < n$).
- Eliminating Term B bias and mitigating the delay-heterogeneity interaction (often appearing in Term C analysis) necessitates using information from all $n$ clients, via full aggregation (ACE) or careful calibration (CA$^2$FL).
- Multiple local steps ($K > 1$) increase the bounds of all error components by accumulating sampling noise and multiplicatively amplifying the bias and delay errors.
- Adaptive learning rates mitigate the error accumulation captured by per-iteration model drift $D_i^t$ by down-weighting updates with large $\tau_i^t$ delays.

Table 1: Impact of Algorithmic Elements on Error Terms (A: Noise, B: Bias, C: Delay)

| Algorithm | Sampling Noise, $\mathbb{E}\|u^t - \bar{u}^t\|_2^2$ | Bias, $\mathbb{E}\|\bar{u}^t - \nabla F(w_{\text{stale}}^t)\|_2^2$ | Delay, $\mathbb{E}\|\nabla F(w_{\text{stale}}^t) - \nabla F(w^t)\|_2^2$ |
|---|---|---|---|
| Vanilla ASGD (Mishchenko et al., 2022) | Not Reduced (due to $m = 1$) | Contains bias from $K \geq 1$ and partial participation $m = 1$. | Contains $\tau\zeta^2$ interaction (from $m = 1$). |
| Delay-adapt ASGD (Koloskova et al., 2022) | Not Reduced (due to $m = 1$) | Contains bias from $K \geq 1$ and partial participation $m = 1$. | Contains $\tau\zeta^2$ interaction (from $m = 1$); Adaptive LR (smaller $\eta$) may reduce reduce delay error. |
| FedBuff (Nguyen et al., 2022) | Reduced by $m$, but increased by local steps $K \geq 1$. | Contains bias from $K \geq 1$ and partial participation $m < n$. | Contains $\tau\zeta^2$ interaction; Error increased by $K \geq 1$. |
| CA$^2$FL (Wang et al., 2024b) | Reduced by $m$, but increased by local steps $K \geq 1$. | Contains bias from $K \geq 1$ and partial participation $m < n$. | No $\tau\zeta^2$ interaction (Calibration); Error increased by $K \geq 1$. |
| **ACE (Ours)** | Max. Reduction (by $m = n$). | Eliminated (by $m = n, K = 1$). | No $\tau\zeta^2$ interaction (by $m = n$). |

**Convergence rate.** Plugging the bounds for ACE's $\text{MSE}_t$ components into the general convergence rate expression (Bound 2) indicates a key benefit of full aggregation. The resulting rate's upper bound is independent of the Bounded Data Heterogeneity (BDH) parameter $\zeta^2$, demonstrating ACE's theoretical robustness to arbitrarily high client heterogeneity. This rate is achieved by selecting an optimal practical learning rate $\eta \propto \sqrt{n/T}$ (see Appendix B.2), leading to Theorem 1:

**Theorem 1** (Convergence Rate of ACE (Alg. 1)). *Suppose Assumptions A1-A5 hold. By choosing an appropriate global step size $\eta$ proportional to $\sqrt{n/T}$, ACE (Algorithm 1) achieves the following convergence rate for smooth non-convex objectives:*

$$\frac{1}{T}\sum_{t=0}^{T-1}\mathbb{E}\|\nabla F(w^t)\|^2 \lesssim \frac{\Delta}{\sqrt{nT}} + \frac{L\sigma^2}{\sqrt{nT}} + \frac{L^2\tau_{max}\sigma^2}{T}$$

*where $\Delta = F(w^0) - F(w^T)$. (Proof can be found in Appendix C).*

## 5 EXPERIMENTAL RESULTS

**Experimental Setup.** We simulate Asynchronous Federated Learning (AFL) on CIFAR-10 dataset(Krizhevsky, 2009) with $N = 100$ clients. Non-IID conditions are created using a Dirichlet distribution (varying $\alpha$), and client delays follow an exponential distribution (varying mean $\beta$). We chose this synthetic heterogeneity setup specifically to independently control heterogeneity ($\alpha$) and delay ($\beta$), allowing us to isolate and verify the multiplicative "heterogeneity amplification" effect predicted by our theory. We compare our ACE algorithm against FedBuff (Nguyen et al., 2022), CA²FL (Wang et al., 2024b), Delay-adaptive ASGD (Koloskova et al., 2022) and Vanilla ASGD (Mishchenko et al., 2022), measuring over $T = 500$ server iterations. Appendix F.3 provides additional results on more models and tasks, including image classification across more heterogeneities and delay settings and Natural Language Processing (NLP) tasks with BERT(Sanh et al., 2019; Devlin et al., 2019) models.

**1. Impact of Non-IID Data (Client Heterogeneity).** Comparing Figure 2(a) with (b), or (c) with (d), increasing data heterogeneity (lower $\alpha$) typically degrades performance. ACE and CA²FL consistently achieve higher final accuracy and converge faster, especially under high heterogeneity ($\alpha = 0.1$). This aligns with our theory that mitigating aggregation bias (Term B in our analysis), as ACE does via its full participation logic, enhances robustness to client heterogeneity.

**2. Impact of Delay and Heterogeneity Amplification.** Increased system delay generally leads to a decline in accuracy for all methods due to larger model drift (as it scales with growing $\tau_{\max}$), as seen when comparing scenarios with higher delay and lower delay (Fig. 2(c) vs. (a) , or (d) vs. (b)).

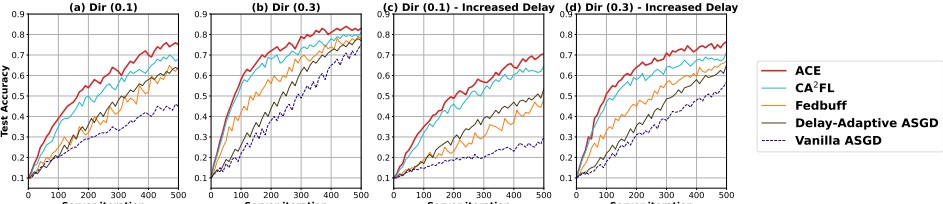

Figure 2: Impact of data heterogeneity (Dirichlet $\alpha$) and client delay (Exponential mean $\beta$) on CIFAR-10 test accuracy over 500 server iterations. (a) $\alpha = 0.1$, low delay ($\beta = 5$). (b) $\alpha = 0.3$, low delay. (c) $\alpha = 0.1$, increased delay ($\beta = 30$). (d) $\alpha = 0.3$, increased delay. ACE demonstrates robust performance toward various heterogeneity and delay. Extended results are in Appendix F.2.

Algorithms with partial client participation (e.g., FedBuff, Vanilla ASGD, Delay-adaptive ASGD), according to our theory (Section 4), are more vulnerable to the $\tau\zeta^2$ interaction within their Delay Error (Term C). Specifically, for these methods, the performance degradation caused by increased delay is more evident when data heterogeneity is high (accuracy difference between Fig. 2(a) and (c)) compared to when heterogeneity is lower (accuracy difference between Fig. 2(b) and (d)). This greater performance drop, along with the slower convergence of the baseline methods, illustrates the heterogeneity amplification effect. In contrast, the superior performance of ACE, supports our insight in Section 4 that all client participation mitigates heterogeneity amplification.

**3. Ablation Study on Local Steps ($K$).** To validate our choice of $K = 1$, we conducted an ablation study varying $K \in \{1, 5, 10\}$ (Table 2). While theoretically increasing $K$ can help reduce the initial suboptimality error term (related to $\Delta$) faster as $T$ increases (as indicated in Table a.1), our results demonstrate that this benefit is overwhelmed by the amplified local model drift in the asynchronous setting. As detailed in our analysis (Section 4 and Appendix B.5), delay $\tau$ interacts with $K$ in a

Table 2: Comprehensive Ablation Study on Local Steps ($K$) across Varying Heterogeneity ($\alpha$) and Delay ($\beta$). We report the final test accuracy (%) at $T = 500$ server iterations. The $K = 1$ column corresponds to the main results in Figure 2. Data for $K = 5$ and $K = 10$ demonstrates that increasing local steps consistently degrades performance in AFL due to drift amplification. ACE shows superior robustness, whereas single-client methods (Vanilla/Delay-Adaptive ASGD) and buffered methods (FedBuff) suffer significant drops, particularly under high delay conditions.

| Algorithm | (a) High Het., Low Delay ($\alpha = 0.1, \beta = 5$) | | | (b) Mod. Het., Low Delay ($\alpha = 0.3, \beta = 5$) | | | (c) High Het., High Delay ($\alpha = 0.1, \beta = 30$) | | | (d) Mod. Het., High Delay ($\alpha = 0.3, \beta = 30$) | | |
|---|---|---|---|---|---|---|---|---|---|---|---|---|
| | $K=1$ | $K=5$ | $K=10$ | $K=1$ | $K=5$ | $K=10$ | $K=1$ | $K=5$ | $K=10$ | $K=1$ | $K=5$ | $K=10$ |
| **ACE (Ours)** | **76.2** | **75.5** | **74.8** | **83.5** | **83.0** | **82.4** | **71.5** | **70.2** | **69.1** | **77.8** | **77.1** | **76.2** |
| CA²FL(Wang et al., 2024b) | 70.5 | 68.1 | 65.4 | 79.2 | 77.5 | 75.3 | 63.2 | 58.5 | 53.1 | 71.5 | 68.2 | 64.9 |
| FedBuff(Nguyen et al., 2022) | 63.8 | 60.2 | 56.5 | 75.8 | 73.2 | 69.8 | 51.5 | 45.8 | 39.5 | 66.5 | 62.1 | 57.4 |
| Delay-Adaptive ASGD(Koloskova et al., 2022) | 64.0 | 61.5 | 57.8 | 78.0 | 75.4 | 72.1 | 55.0 | 49.5 | 42.8 | 68.0 | 63.5 | 58.2 |
| Vanilla ASGD(Mishchenko et al., 2022) | 45.0 | 41.2 | 36.5 | 75.0 | 71.5 | 67.0 | 30.5 | 24.8 | 18.5 | 58.5 | 52.4 | 46.8 |

multiplicative, harmful way. For instance, the drift error bound for FedBuff(Nguyen et al., 2022) scales with $\mathcal{O}(\tau \cdot (n - m)K^2\zeta^2)$, and for CA²FL(Wang et al., 2024b) it involves terms scaling with $\mathcal{O}((1 + \frac{1}{n^2}(n - m)^2)\tau \cdot K\sigma^2)$. Consequently, increasing $K$ causes clients to accumulate larger deviation vectors based on outdated information. Empirically, this leads to consistent performance degradation for all algorithms as $K$ increases. Notably, ACE exhibits the greatest robustness, as its full aggregation ($m = n$) eliminates the $(n - m)$ multiplier, thereby minimizing this multiplicative drift error.

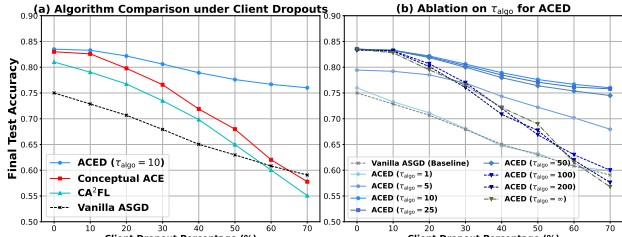

Figure 3: Final test accuracy ($T = 500$, Dir($\alpha = 0.3$), $\beta = 5$) vs. client dropout. (a) ACED ($\tau_{\text{algo}} = 10$) shows superior dropout robustness compared to Conceptual ACE, CA²FL, and Vanilla ASGD. (b) Ablation on ACED's $\tau_{\text{algo}}$: performance suffers if $\tau_{\text{algo}}$ is too small (partial participation bias) or too large (staleness error), but is stable across moderate $\tau_{\text{algo}}$ values.

**Delay-aware Aggregation under Client Dropouts.** We investigate ACED's robustness to client dropouts (from 0% to 70%) under Dir($\alpha = 0.3$) and $\beta = 5$, starting at $t = T/2 = 250$, as shown in Figure 3. Compared to other methods, ACED (using $\tau_{\text{algo}} = 10 = 2\beta$) exhibits enhanced resilience, highlighting the role of $\tau_{\text{algo}}$ in managing a trade-off between two error sources. Our ablation study quantifies this trade-off: an excessively small $\tau_{\text{algo}}$ (e.g., 1, resembling Vanilla ASGD) minimizes staleness but incurs high participation bias. Conversely, a very large $\tau_{\text{algo}}$ (e.g., $\geq 100 = T/5$) includes too many stale updates, leading to model drift. Therefore, since ACED allows dropped or delayed clients to contribute again once their delay recovers (Algorithm a.1), selecting $\tau_{\text{algo}}$ within a wide moderate range ($[10, 50]$) proves effective. This strategy maximizes participation to better address the common challenge of heterogeneity in AFL and reduce the impact of participation bias.

# 6 CONCLUSION

Our work introduces a general theoretical framework to analyze Asynchronous Federated Learning (AFL) algorithms by decomposing the total error. Our analysis using this framework identifies that client participation imbalance bias is the root cause of **heterogeneity amplification**. Based on this insight, we propose ACE; its immediate, non-buffered aggregation of all clients eliminates participation bias and ensures robust, communication-efficient convergence under high heterogeneity. For practical challenges like extreme delays, the delay-aware variant ACED uses a staleness threshold to manage the trade-off between maximizing client diversity (to reduce bias) and minimizing error from stale gradients (to reduce delay error). Experiments confirm our methods achieve more stable performance, particularly in challenging settings with high data heterogeneity and system delays.

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

Appendix Contents

## A   NOTATIONS

This section outlines the notation used and details the problem setup as presented in the paper.

### A.1   PROBLEM SETTING

The paper considers an Asynchronous Federated Learning (AFL) system with $n$ clients and a central server. The objective is to minimize a global function $F(w)$, which is an average of local client objectives $F_i(w)$:

$$F(w) = \frac{1}{n} \sum_{i=1}^{n} F_i(w)$$

Each local objective $F_i(w)$ is the expected loss over client $i$'s true local data distribution $\mathbb{P}_i$:

$$F_i(w) = \mathbb{E}_{\xi_i \sim \mathbb{P}_i}[f_i(w; \xi_i)]$$

Here, $w \in \mathbb{R}^d$ represents the model parameters, and $f_i(w; \xi_i)$ is the loss for a data sample $\xi_i$ from client $i$'s distribution. Clients compute stochastic gradients $\nabla f_i(w; \xi_i)$ as approximations to the true local gradients $\nabla F_i(w)$.

The server maintains the global model $w^t$ at server iteration $t$ (up to $T$ total iterations). In the asynchronous setting, the global model is updated, for example, via $w^{t+1} = w^t - \eta u^t$. The crucial aspect is that the global update $u^t$ is formed using potentially stale information. A contribution from client $i$ to $u^t$ might be based on a model version $w^{t-\tau_i^t}$ it received earlier, where $\tau_i^t \geq 0$ is the information staleness (or delay) of that client's information relative to the current server iteration $t$.

### A.2   NOTATIONS

The following notations are used:

- $n$: Total number of clients.
- $w^t \in \mathbb{R}^d$: Global model parameters at server iteration $t$.
- $d$: Dimensionality of the model parameters.
- $T$: Total number of server iterations.
- $F(w)$: Global objective function.
- $F_i(w)$: Local objective function for client $i$.
- $\mathbb{P}_i$: True local data distribution for client $i$.
- $f_i(w; \xi_i)$: Loss function for client $i$ on data sample $\xi_i$.
- $\nabla F(w)$: True gradient of the global objective function.
- $\nabla F_i(w)$: True gradient of the local objective function for client $i$.
- $\nabla f_i(w; \xi_i)$: Stochastic gradient computed by client $i$ from sample $\xi_i$ based on model $w$.
- $\eta$: Server-side learning rate (step size).
- $\eta_l$: Client-side local learning rate (mentioned in context of other algorithms like FedBuff in Section 4).
- $u^t$: Global update vector applied by the server at iteration $t$.
- $\tau_i^t, \rho_i^t$: Information staleness (delay) for client $i$'s contribution to the update at server iteration $t$. This is the difference in server iterations between when client $i$ received the model it used for computation and the current server iteration $t$. $\rho_i^t$ is specified for the delay of the models in the cache in CA$^2$FL (Wang et al., 2024b).
- $\mathbb{E}[\cdot]$: Total expectation over all sources of randomness.
- $\mathcal{F}^t$: The $\sigma$-algebra of all information available up to server iteration $t$ (including $w^t$).
- $\mathbb{E}_t[\cdot]$ or $\mathbb{E}[\cdot|\mathcal{F}^t]$: Conditional expectation given $\mathcal{F}^t$.

- $\mathcal{H}^t$: The $\sigma$-algebra containing all information determining $u^t$ (including stale models used and aggregation rules) except the randomness from the set of fresh data samples $\{\xi_i\}$ contributing to $u^t$.

- $\overline{u}^t := \mathbb{E}_{\{\xi_i\}}[u^t|\mathcal{H}^t]$: Expected global update over the fresh data samples $\{\xi_i\}$ used to form $u^t$, conditional on $\mathcal{H}^t$.

- $w^t_{\text{stale}} = \{w^{t-\tau_i^t}\}_{i=1}^n$: Collection of stale models used by clients whose updates contribute to $u^t$.

- $\nabla F(w^t_{\text{stale}}) = \frac{1}{n}\sum_{i=1}^n \nabla F_i(w^{t-\tau_i^t})$: The average true gradient evaluated on the specific stale model versions $w^{t-\tau_i^t}$ used by the clients.

- $\xi_i^{\kappa_i}$: A specific data sample used by client $i$ to generate the gradient that was received and cached by the server at server iteration $\kappa_i$ (where $t - \tau_i^t < \kappa_i \leq t$), computed on model $w^{t-\tau_i^t}$.

- $U_i^t$ or $U_i^{\text{cache}}$: The latest available (potentially stale) gradient from client $i$ cached at the server at iteration $t$. For ACE, $U_i^t = \nabla f_i(w^{t-\tau_i^t}; \xi_i^{\kappa_i})$.

- $F^*$: Lower bound of the global objective function, $F(w) \geq F^* > -\infty$.

- $L$: Lipschitz constant for the smoothness of local objective functions $F_i(w)$.

- $\sigma^2$: Bound on the variance of local stochastic gradients, $\mathbb{E}_{\xi_i}||\nabla f_i(w;\xi_i) - \nabla F_i(w)||_2^2 \leq \sigma^2$.

- $\tau_{\max}$: Bound on the maximum delay, $\tau_i^t \leq \tau_{\max}$.

- $\zeta^2$: Bound for Bounded Data Heterogeneity (BDH), $||\nabla F_i(w) - \nabla F(w)||^2 \leq \zeta^2$ (mentioned as an assumption in some other algorithms, but ACE aims to eliminate the need for it by full aggregation).

- $K$: Number of local steps (mentioned in context of other algorithms like FedBuff in Section 4).

- $t_i^{\text{start}}$: Server iteration when client $i$ obtained the model $w^{t_i^{\text{start}}}$ upon which its currently cached gradient $U_i^{\text{cache}}$ was computed (in ACED).

- $\tau_{\text{algo}}$: Maximum allowed delay threshold for gradient inclusion in ACED.

- $A(t)$: Set of active clients in ACED at server iteration $t$, defined as $\{i \in [n] | t - t_i^{\text{start}} \leq \tau_{\text{algo}}\}$.

- $n_t = |A(t)|$: Number of active clients in $A(t)$ for ACED.

- $n_{\min}$: Lower bound on $n_t$, i.e., $n_t \geq n_{\min} \geq 1$ (for ACED).

- $G$: Bound on the norm of expected local gradients, $||\nabla F_i(w)||_2 \leq G$ (Assumption a.7 for ACED analysis, noted as not strictly necessary but simplifying).

- $\Delta = F(w^0) - \mathbb{E}[F(w^T)]$ or $F(w^0) - F^*$: Initial suboptimality.

This list covers the primary notations introduced and used in the problem setup and for the analysis of ACE and related concepts within the specified paper. The paper also refers to notations from other algorithms (FedBuff, CA$^2$FL, Delay-adaptive ASGD) when making comparisons, which might have their own specific notations detailed in their respective original publications or the provided supplementary material.

# B  PROOFS FOR SECTION 4

All the notations used in this section are detailed in Appendix A. Details of the implementations of the baseline algorithms (FedBuff (Nguyen et al., 2022), CA$^2$FL (Wang et al., 2024b), Delay-adaptive ASGD (Koloskova et al., 2022) and Vanilla ASGD (Mishchenko et al., 2022)) can be found in F.

## B.1  USEFUL LEMMAS

**Lemma a.1.** *For two arbitrary vectors $a, b \in \mathbb{R}^d$, the inner product can be expressed as:*

$$\langle a, b \rangle = \frac{1}{2} \left( \|a\|^2 + \|b\|^2 - \|a - b\|^2 \right).$$

*Proof.* Expand $\|a - b\|^2$:

$$\begin{aligned} \|a - b\|^2 &= \langle a - b, a - b \rangle \\ &= \langle a, a \rangle - 2\langle a, b \rangle + \langle b, b \rangle \\ &= \|a\|^2 + \|b\|^2 - 2\langle a, b \rangle. \end{aligned}$$

Rearranging this gives $\langle a, b \rangle = \frac{1}{2}(\|a\|^2 + \|b\|^2 - \|a - b\|^2)$.  $\square$

**Lemma a.2.** *For vectors $x_i \in \mathbb{R}^d$, $i = 1, \ldots, n$:*

$$\left\| \sum_{k=1}^{n} x_k \right\|^2 = n \sum_{k=1}^{n} \|x_k\|^2 - \frac{1}{2} \sum_{i=1}^{n} \sum_{\substack{j=1 \\ j \neq i}}^{n} \|x_i - x_j\|^2$$

*Proof.* We first prove the following auxiliary identity: For vectors $x_1, \ldots, x_n \in \mathbb{R}^d$:

$$\sum_{i=1}^{n} \sum_{j=1}^{n} \|x_i - x_j\|^2 = 2n \sum_{k=1}^{n} \|x_k\|^2 - 2 \left\| \sum_{k=1}^{n} x_k \right\|^2$$

$$\begin{aligned} \sum_{i=1}^{n} \sum_{j=1}^{n} \|x_i - x_j\|^2 &= \sum_{i=1}^{n} \sum_{j=1}^{n} \left( \|x_i\|^2 - 2 \langle x_i, x_j \rangle + \|x_j\|^2 \right) \\ &= \sum_{i=1}^{n} \sum_{j=1}^{n} \|x_i\|^2 + \sum_{i=1}^{n} \sum_{j=1}^{n} \|x_j\|^2 - 2 \sum_{i=1}^{n} \sum_{j=1}^{n} \langle x_i, x_j \rangle \\ &= \sum_{j=1}^{n} \left( \sum_{i=1}^{n} \|x_i\|^2 \right) + \sum_{i=1}^{n} \left( \sum_{j=1}^{n} \|x_j\|^2 \right) - 2 \left\langle \sum_{i=1}^{n} x_i, \sum_{j=1}^{n} x_j \right\rangle \\ &= n \sum_{i=1}^{n} \|x_i\|^2 + n \sum_{j=1}^{n} \|x_j\|^2 - 2 \left\| \sum_{k=1}^{n} x_k \right\|^2 \\ &= 2n \sum_{k=1}^{n} \|x_k\|^2 - 2 \left\| \sum_{k=1}^{n} x_k \right\|^2. \end{aligned}$$

Note that when $i = j$, $\|x_i - x_j\|^2 = \|x_i - x_i\|^2 = \|\mathbf{0}\|^2 = 0$.
Therefore, the sum $\sum_{i=1}^{n} \sum_{j=1}^{n} \|x_i - x_j\|^2$ can be split based on whether $i = j$ or $i \neq j$:

$$\sum_{i=1}^{n} \sum_{j=1}^{n} \|x_i - x_j\|^2 = \sum_{i=1}^{n} \sum_{\substack{j=1 \\ j \neq i}}^{n} \|x_i - x_j\|^2 + \sum_{i=1}^{n} \|x_i - x_i\|^2 = \sum_{i=1}^{n} \sum_{\substack{j=1 \\ j \neq i}}^{n} \|x_i - x_j\|^2 + 0$$

So, $\sum_{i=1}^{n} \sum_{j=1}^{n} \|x_i - x_j\|^2 = \sum_{i=1}^{n} \sum_{\substack{j=1 \\ j \neq i}}^{n} \|x_i - x_j\|^2$. Thus, the auxiliary identity can be rewritten as:

$$\sum_{i=1}^{n} \sum_{\substack{j=1 \\ j \neq i}}^{n} \|x_i - x_j\|^2 = 2n \sum_{k=1}^{n} \|x_k\|^2 - 2 \left\| \sum_{k=1}^{n} x_k \right\|^2$$

Rearranging this equation to solve for $\|\sum_{k=1}^{n} x_k\|^2$:

$$2 \left\| \sum_{k=1}^{n} x_k \right\|^2 = 2n \sum_{k=1}^{n} \|x_k\|^2 - \sum_{i=1}^{n} \sum_{\substack{j=1 \\ j \neq i}}^{n} \|x_i - x_j\|^2$$

Dividing both sides by 2 yields the lemma:

$$\left\| \sum_{k=1}^{n} x_k \right\|^2 = n \sum_{k=1}^{n} \|x_k\|^2 - \frac{1}{2} \sum_{i=1}^{n} \sum_{\substack{j=1 \\ j \neq i}}^{n} \|x_i - x_j\|^2$$

$\square$

**Lemma a.3.** *For vectors $x_i \in \mathbb{R}^d$, $i = 1, \ldots, n$:*

$$\left\| \sum_{i=1}^{n} x_i \right\|^2 \leq n \sum_{i=1}^{n} \|x_i\|^2.$$

*A special case for two vectors $a, b \in \mathbb{R}^d$:*

$$\|a + b\|^2 \leq 2(\|a\|^2 + \|b\|^2).$$

*Proof.* This lemma is a corollary of Lemma a.2, since $\frac{1}{2} \sum_{i=1}^{n} \sum_{\substack{j=1 \\ j \neq i}}^{n} \|x_i - x_j\|^2 \geq 0$. $\square$

*Note: This lemma is very useful for "extracting the summation symbol" in a norm.*

**Lemma a.4** (Descent Lemma)**.** *For an L-smooth function $F : \mathbb{R}^d \to \mathbb{R}$, for any $x, y \in \mathbb{R}^d$:*

$$F(y) \leq F(x) + \langle \nabla F(x), y - x \rangle + \frac{L}{2} \|y - x\|^2.$$

*Proof.* By the Fundamental Theorem of Calculus:

$$F(y) - F(x) = \int_0^1 \langle \nabla F(x + \tau(y - x)), y - x \rangle d\tau.$$

Adding and subtracting $\langle \nabla F(x), y - x \rangle$:

$$F(y) - F(x) = \int_0^1 \langle \nabla F(x + \tau(y - x)) - \nabla F(x) + \nabla F(x), y - x \rangle d\tau$$

$$= \int_0^1 \langle \nabla F(x), y - x \rangle d\tau + \int_0^1 \langle \nabla F(x + \tau(y - x)) - \nabla F(x), y - x \rangle d\tau$$

$$= \langle \nabla F(x), y - x \rangle + \int_0^1 \langle \nabla F(x + \tau(y - x)) - \nabla F(x), y - x \rangle d\tau.$$

Using Cauchy-Schwarz inequality and $L$-smoothness ($L$-gradient Lipschitz property, which states $\|\nabla F(a) - \nabla F(b)\| \leq L\|a - b\|$):

$$\int_0^1 \langle \nabla F(x + \tau(y - x)) - \nabla F(x), y - x \rangle d\tau \leq \int_0^1 \|\nabla F(x + \tau(y - x)) - \nabla F(x)\| \|y - x\| d\tau$$

$$\leq \int_0^1 L\|(x + \tau(y - x)) - x\| \|y - x\| d\tau$$

$$= \int_0^1 L\|\tau(y - x)\| \|y - x\| d\tau$$

$$= \int_0^1 L\tau \|y - x\|^2 d\tau$$

$$= L\|y - x\|^2 \int_0^1 \tau d\tau$$

$$= L\|y - x\|^2 \left[\frac{\tau^2}{2}\right]_0^1 = \frac{L}{2}\|y - x\|^2.$$

Therefore,

$$F(y) - F(x) \leq \langle \nabla F(x), y - x \rangle + \frac{L}{2}\|y - x\|^2,$$

which implies the statement of the lemma:

$$F(y) \leq F(x) + \langle \nabla F(x), y - x \rangle + \frac{L}{2}\|y - x\|^2.$$

*Application to algorithm analysis:* This lemma is frequently applied in the analysis of iterative optimization algorithms. For an algorithm with an update rule of the form $w^{t+1} = w^t - \eta u^t$, where $w^t$ is the model at iteration $t$, $u^t$ is the update direction (possibly stochastic), and $\eta$ is the step size, we can set $x = w^t$ and $y = w^{t+1}$. Then $y - x = w^{t+1} - w^t = -\eta u^t$. Substituting these into the lemma:

$$F(w^{t+1}) \leq F(w^t) + \langle \nabla F(w^t), -\eta u^t \rangle + \frac{L}{2}\| - \eta u^t\|^2$$

$$F(w^{t+1}) \leq F(w^t) - \eta \langle \nabla F(w^t), u^t \rangle + \frac{L\eta^2}{2}\|u^t\|^2.$$

If $u^t$ involves randomness (e.g., from stochastic gradients or client sampling), we typically take the total expectation $\mathbb{E}[\cdot]$ over all sources of randomness:

$$\mathbb{E}[F(w^{t+1})] \leq \mathbb{E}[F(w^t)] - \eta \mathbb{E}[\langle \nabla F(w^t), u^t \rangle] + \frac{L\eta^2}{2}\mathbb{E}\|u^t\|^2.$$

This inequality then forms the basis for analyzing the expected decrease in the objective function per iteration. $\square$

**Lemma a.5** ((Reddi et al., 2021), Model Drift from Local Steps). *For local learning rate which satisfying $\eta_l \leq \frac{1}{8KL}$, the local model difference after $k$ ($\forall k \in \{0, 1, \ldots, K-1\}$) steps local updates satisfies*

$$\frac{1}{n}\sum_{i=1}^n \mathbb{E}[\|w_i^{t,k} - w^t\|^2] \leq 5K\eta_l^2(\sigma^2 + 6K\zeta^2) + 30K^2\eta_l^2\mathbb{E}[\|\nabla F(w^t)\|^2]. \tag{a.1}$$

*Proof.* The proof of Lemma a.5 is exactly same as the proof of Lemma 3 in (Reddi et al., 2021). $\square$

**Lemma a.6** (Cross-Iteration Gradient Error Independence). *Let $\delta_k^s = \nabla f_k(w^{s-\tau_k^s}; \xi_k^{\kappa_k(s)}) - \nabla F_k(w^{s-\tau_k^s})$ denote the stochastic error of the gradient for client $k$'s contribution at server iteration $s$. The gradient is computed by client $k$ using model $w^{s-\tau_k^s}$ and a data sample $\xi_k^{\kappa_k(s)}$. This specific sample $\xi_k^{\kappa_k(s)}$ was used by client $k$ to generate the gradient that was received and cached by*

*the server at its iteration $\kappa_k(s)$, where $s - \tau_k^s < \kappa_k(s) \leq s$. The sample $\xi_k^{\kappa_k(s)}$ was drawn fresh by client $k$ at the time of its local computation on $w^{s - \tau_k^s}$.*

*Under Assumption 3 (Unbiased Stochastic Gradients), for any two distinct server iterations $s_1 \neq s_2$, the expected inner product of the sum of these errors is zero:*

$$\mathbb{E}\left[\langle \delta_i^{s_1}, \delta_j^{s_2}\rangle\right] = 0 \qquad (\forall i, j \in [n])$$

*Proof.* Without loss of generality, assume $s_1 < s_2$. Let $\mathcal{F}_{s_2-1}$ be the $\sigma$-algebra generated by all information available up to and including server iteration $s_2 - 1$. By definition, $w^{s_1 - \tau_i^{s_1}}$ and the sample $\xi_i^{\kappa_i(s_1)}$ used to compute $\delta_i^{s_1}$ are contained within this information set. Thus, $\delta_i^{s_1}$ is $\mathcal{F}_{s_2-1}$-measurable.

The sample $\xi_j^{\kappa_j(s_2)}$ is drawn fresh by client $j$ for its local computation on model $w^{s_2 - \tau_j^{s_2}}$. This sample draw is independent of the history $\mathcal{F}_{s_2-1}$ (conditional on $w^{s_2 - \tau_j^{s_2}}$). By Assumption 3 (Unbiased Stochastic Gradients), for a given model $w^{s_2 - \tau_j^{s_2}}$, the gradient computed using the fresh sample $\xi_j^{\kappa_j(s_2)}$ is unbiased:

$$\mathbb{E}_{\xi_j^{\kappa_j(s_2)}}[\nabla f_j(w^{s_2 - \tau_j^{s_2}}; \xi_j^{\kappa_j(s_2)})|w^{s_2 - \tau_j^{s_2}}] = \nabla F_j(w^{s_2 - \tau_j^{s_2}})$$

Therefore, the conditional expectation of $\delta_j^{s_2}$ given $w^{s_2 - \tau_j^{s_2}}$ is:

$$\mathbb{E}_{\xi_j^{\kappa_j(s_2)}}[\delta_j^{s_2}|w^{s_2 - \tau_j^{s_2}}] = \mathbb{E}_{\xi_j^{\kappa_j(s_2)}}[\nabla f_j(w^{s_2 - \tau_j^{s_2}}; \xi_j^{\kappa_j(s_2)})|w^{s_2 - \tau_j^{s_2}}] - \nabla F_j(w^{s_2 - \tau_j^{s_2}}) = 0$$

Now consider the conditional expectation of $\delta_j^{s_2}$ with respect to $\mathcal{F}_{s_2-1}$. By the tower property (law of total expectation), and noting that given $w^{s_2 - \tau_j^{s_2}}$, the randomness of $\xi_j^{\kappa_j(s_2)}$ is independent of other information in $\mathcal{F}_{s_2-1}$:

$$\mathbb{E}[\delta_j^{s_2}|\mathcal{F}_{s_2-1}] = \mathbb{E}\left[\mathbb{E}[\delta_j^{s_2}|w^{s_2 - \tau_j^{s_2}}, \mathcal{F}_{s_2-1}]|\mathcal{F}_{s_2-1}\right]$$

$$= \mathbb{E}\left[\mathbb{E}_{\xi_j^{\kappa_j(s_2)}}[\delta_j^{s_2}|w^{s_2 - \tau_j^{s_2}}]|\mathcal{F}_{s_2-1}\right]$$

$$= \mathbb{E}[0|\mathcal{F}_{s_2-1}]$$

$$= 0$$

Next, we use the law of total expectation to evaluate $\mathbb{E}[\langle \delta_i^{s_1}, \delta_j^{s_2}\rangle]$:

$$\mathbb{E}[\langle \delta_i^{s_1}, \delta_j^{s_2}\rangle] = \mathbb{E}\left[\mathbb{E}[\langle \delta_i^{s_1}, \delta_j^{s_2}\rangle|\mathcal{F}_{s_2-1}]\right]$$

Since $\delta_i^{s_1}$ is $\mathcal{F}_{s_2-1}$-measurable:

$$\mathbb{E}[\langle \delta_i^{s_1}, \delta_j^{s_2}\rangle] = \mathbb{E}\left[\langle \delta_i^{s_1}, \mathbb{E}[\delta_j^{s_2}|\mathcal{F}_{s_2-1}]\rangle\right] = \mathbb{E}[\langle \delta_i^{s_1}, 0\rangle] = 0$$

This holds for all pairs of clients $(i, j)$ when $s_1 < s_2$. A symmetric argument applies if $s_2 < s_1$. Therefore, when $s_1 \neq s_2$, all cross-terms $\mathbb{E}[\langle \delta_i^{s_1}, \delta_j^{s_2}\rangle]$ are zero. Consequently,

$$\mathbb{E}\left[\langle \delta_i^{s_1}, \delta_j^{s_2}\rangle\right] = 0 \quad \text{when } s_1 \neq s_2$$

This completes the proof. $\qquad\qquad\square$

## B.2 MSE CONVERGENCE CONTROL AND THE OPTIMAL LEARNING RATE

**Theorem a.1** (MSE controls the AFL convergence). *By summing the inequality in the Descent Lemma a.4 over $T$ iterations and rearranging terms, we derive the following bound. This inequality bounds the average squared norm of the true gradient, a standard measure of convergence for non-convex objectives, by terms including the Mean Squared Error (MSE) of our gradient estimates (constants $\gamma_1, \gamma_2 > 0$):*

$$\frac{1}{T}\sum_{t=0}^{T-1}\mathbb{E}\|\nabla F(w^t)\|^2 \leq \frac{\gamma_1(F(w^0) - \mathbb{E}[F(w^T)])}{T\eta} + \gamma_2\eta\left(\frac{1}{T}\sum_{t=0}^{T-1}\underbrace{\mathbb{E}\|u^t - \nabla F(w^t)\|_2^2}_{MSE_t}\right) \quad \text{(a.2)}$$

*Proof.* We start from the expected Descent Lemma (Lemma a.4 applied to $w^{t+1} = w^t - \eta u^t$ and taking expectation):

$$\mathbb{E}[F(w^{t+1})] \leq \mathbb{E}[F(w^t)] - \eta\mathbb{E}[\langle\nabla F(w^t), u^t\rangle] + \frac{L\eta^2}{2}\mathbb{E}\|u^t\|^2 \quad \text{(a.3)}$$

Rearranging a.3, we get:

$$\eta\mathbb{E}[\langle\nabla F(w^t), u^t\rangle] \leq \mathbb{E}[F(w^t)] - \mathbb{E}[F(w^{t+1})] + \frac{L\eta^2}{2}\mathbb{E}\|u^t\|^2 \quad \text{(a.4)}$$

We relate the inner product term to $\|\nabla F(w^t)\|^2$ and $MSE_t = \mathbb{E}\|u^t - \nabla F(w^t)\|_2^2$. Consider the term $\langle\nabla F(w^t), u^t\rangle$:

$$\langle\nabla F(w^t), u^t\rangle = \langle\nabla F(w^t), \nabla F(w^t) + u^t - \nabla F(w^t)\rangle$$
$$= \|\nabla F(w^t)\|^2 + \langle\nabla F(w^t), u^t - \nabla F(w^t)\rangle$$

Taking expectation:

$$\mathbb{E}[\langle\nabla F(w^t), u^t\rangle] = \mathbb{E}[\|\nabla F(w^t)\|^2] + \mathbb{E}[\langle\nabla F(w^t), u^t - \nabla F(w^t)\rangle]$$

Using Young's inequality $\langle a, b\rangle \geq -\frac{1}{2}\|a\|^2 - \frac{1}{2}\|b\|^2$ for the second term (by negating it: $-\langle a, b\rangle \leq \frac{1}{2}\|a\|^2 + \frac{1}{2}\|b\|^2$):

$$\mathbb{E}[\langle\nabla F(w^t), u^t - \nabla F(w^t)\rangle] \geq -\frac{1}{2}\mathbb{E}[\|\nabla F(w^t)\|^2] - \frac{1}{2}\mathbb{E}[\|u^t - \nabla F(w^t)\|^2]$$

So,

$$\mathbb{E}[\langle\nabla F(w^t), u^t\rangle] \geq \mathbb{E}[\|\nabla F(w^t)\|^2] - \frac{1}{2}\mathbb{E}[\|\nabla F(w^t)\|^2] - \frac{1}{2}MSE_t = \frac{1}{2}\mathbb{E}[\|\nabla F(w^t)\|^2] - \frac{1}{2}MSE_t$$

Substitute this back into a.4:

$$\eta\left(\frac{1}{2}\mathbb{E}[\|\nabla F(w^t)\|^2] - \frac{1}{2}MSE_t\right) \leq \mathbb{E}[F(w^t)] - \mathbb{E}[F(w^{t+1})] + \frac{L\eta^2}{2}\mathbb{E}\|u^t\|^2$$

$$\frac{\eta}{2}\mathbb{E}[\|\nabla F(w^t)\|^2] \leq \mathbb{E}[F(w^t)] - \mathbb{E}[F(w^{t+1})] + \frac{\eta}{2}MSE_t + \frac{L\eta^2}{2}\mathbb{E}\|u^t\|^2 \quad \text{(a.5)}$$

We bound $\mathbb{E}\|u^t\|^2$ using $\|a + b\|^2 \leq 2\|a\|^2 + 2\|b\|^2$ (Lemma a.3):

$$\mathbb{E}\|u^t\|^2 = \mathbb{E}\|u^t - \nabla F(w^t) + \nabla F(w^t)\|^2 \leq 2\mathbb{E}\|u^t - \nabla F(w^t)\|^2 + 2\mathbb{E}\|\nabla F(w^t)\|^2 = 2MSE_t + 2\mathbb{E}\|\nabla F(w^t)\|^2$$

Substitute this into a.5:

$$\frac{\eta}{2}\mathbb{E}[\|\nabla F(w^t)\|^2] \leq \mathbb{E}[F(w^t)] - \mathbb{E}[F(w^{t+1})] + \frac{\eta}{2}MSE_t + \frac{L\eta^2}{2}(2MSE_t + 2\mathbb{E}\|\nabla F(w^t)\|^2)$$

$$= \mathbb{E}[F(w^t)] - \mathbb{E}[F(w^{t+1})] + \left(\frac{\eta}{2} + L\eta^2\right)MSE_t + L\eta^2\mathbb{E}\|\nabla F(w^t)\|^2$$

Rearranging terms to isolate $\mathbb{E}[\|\nabla F(w^t)\|^2]$:

$$\left(\frac{\eta}{2} - L\eta^2\right)\mathbb{E}[\|\nabla F(w^t)\|^2] \leq \mathbb{E}[F(w^t)] - \mathbb{E}[F(w^{t+1})] + \eta\left(\frac{1}{2} + L\eta\right)MSE_t$$

Assume the step size $\eta$ is chosen such that $\eta \leq \frac{1}{4L}$. Then $\frac{1}{2} - L\eta \geq \frac{1}{2} - \frac{1}{4} = \frac{1}{4}$. So, $\frac{\eta}{2} - L\eta^2 = \eta(\frac{1}{2} - L\eta) \geq \frac{\eta}{4}$. Also, $\frac{1}{2} + L\eta \leq \frac{1}{2} + \frac{1}{4} = \frac{3}{4}$. Thus,

$$\frac{\eta}{4}\mathbb{E}[\|\nabla F(w^t)\|^2] \leq \mathbb{E}[F(w^t)] - \mathbb{E}[F(w^{t+1})] + \frac{3\eta}{4}\text{MSE}_t$$

Multiplying by $\frac{4}{\eta}$:

$$\mathbb{E}[\|\nabla F(w^t)\|^2] \leq \frac{4}{\eta}(\mathbb{E}[F(w^t)] - \mathbb{E}[F(w^{t+1})]) + 3\eta \cdot \text{MSE}_t$$

Summing from $t = 0$ to $T - 1$:

$$\sum_{t=0}^{T-1} \mathbb{E}[\|\nabla F(w^t)\|^2] \leq \frac{4}{\eta} \sum_{t=0}^{T-1} (\mathbb{E}[F(w^t)] - \mathbb{E}[F(w^{t+1})]) + 3\eta \sum_{t=0}^{T-1} \text{MSE}_t$$

$$= \frac{4}{\eta}(\mathbb{E}[F(w^0)] - \mathbb{E}[F(w^T)]) + 3\eta \sum_{t=0}^{T-1} \text{MSE}_t \quad \text{(telescoping sum)}$$

Dividing by $T$:

$$\frac{1}{T}\sum_{t=0}^{T-1} \mathbb{E}[\|\nabla F(w^t)\|^2] \leq \frac{4(\mathbb{E}[F(w^0)] - \mathbb{E}[F(w^T)])}{T\eta} + 3\eta\left(\frac{1}{T}\sum_{t=0}^{T-1} \text{MSE}_t\right)$$

This matches the desired form with $\gamma_1 = 4$ and $\gamma_2 = 3$, under the condition $\eta \leq \frac{1}{4L}$. Note that $F(w^T)$ is often replaced by $F^* = \min_w F(w)$ since $F(w^T) \geq F^*$, which makes the bound looser but independent of $F(w^T)$. The term $\mathbb{E}[F(w^0)] - \mathbb{E}[F(w^T)]$ is used for a finite $T$. If $F(w^T)$ is simply written as $F(w^T)$, and expectation is dropped for $F(w^0)$ (if $w^0$ is deterministic), then we have:

$$\frac{1}{T}\sum_{t=0}^{T-1} \mathbb{E}\|\nabla F(w^t)\|^2 \leq \frac{4(F(w^0) - \mathbb{E}[F(w^T)])}{T\eta} + 3\eta\left(\frac{1}{T}\sum_{t=0}^{T-1} \underbrace{\mathbb{E}\|u^t - \nabla F(w^t)\|_2^2}_{\text{MSE}_t}\right)$$

The constants $\gamma_1, \gamma_2$ might differ based on the specific choices made in applying Young's inequality (underlined part) or in setting learning rate. Here we only give the existence proof of $\gamma_1, \gamma_2$ by taking certain values. $\square$

**Theorem a.2** (Optimal Learning Rate Scaling for ACE). *The optimal learning rate $\eta^*$ that minimizes the convergence upper bound for ACE is proportional to $\sqrt{n/T}$.*

*Proof.* Our goal is to select a learning rate $\eta$ that minimizes the convergence rate's upper bound from Theorem a.1:

$$\mathcal{R}(\eta) = \frac{\gamma_1 \Delta}{T\eta} + \gamma_2 \eta \underbrace{\left(\frac{1}{T}\sum_{t=0}^{T-1} \mathbb{E}\|u^t - \nabla F(w^t)\|_2^2\right)}_{\overline{\text{MSE}}}$$

It is worth noting that the $\overline{\text{MSE}}$ term can be a function of $\eta$. From our MSE decomposition 4, the error consists of Term A (Noise), Term B (Bias), and Term C (Delay).
For ACE,

- (Appendix B.3) The upper bound of Sampling Noise (Term A) is $\mathbb{E}\|A\|_2^2 \leq \frac{\sigma^2}{n}$.

- (Appendix B.4) Term B is zero.

- (Appendix B.5) As shown in the analysis of model drift $D_i^t$, the Delay Error contains a component that scales with $\eta^2$:

$$\mathbb{E}\|\text{Term C}\|^2 \leq \frac{L^2}{n}\sum_i D_i^t \leq \eta^2 \frac{L^2}{n}\tau_{\max}\left(\frac{\sigma^2}{n} + \dots\right)$$

Thus, the full upper bound $\mathcal{R}(\eta)$ contains terms proportional to $1/\eta$, $\eta$, and $\eta^3$:

$$\mathcal{R}(\eta) \lesssim \frac{1}{T\eta} + \eta\left(\frac{\sigma^2}{n}\right) + \eta^3\left(\frac{L^2\tau_{\max}\sigma^2}{n} + \cdots\right) \tag{a.6}$$

For the algorithm to converge, the left-hand side of Bound 2 must approach zero as $T \to \infty$. Consequently, its upper bound, $\mathcal{R}(\eta)$, must also approach zero. For $\mathcal{R}(\eta) \to 0$, the learning rate $\eta$ must be a vanishing quantity, i.e., $\eta(T) \to 0$ as $T \to \infty$. If $\eta$ were a constant, the terms proportional to $\eta$ and $\eta^3$ would prevent the bound from converging to zero.

Since we have established that $\eta$ must be a small quantity for large $T$, higher-order terms in $\eta$ become negligible. Specifically, the $\eta^3$ term diminishes much faster than the $\eta$ term. Therefore, for the purpose of finding the optimal *scaling rate* of the learning rate, the behavior of $\mathcal{R}(\eta)$ is dominated by the first two terms. The problem simplifies to minimizing the dominant part of the bound:

$$\mathcal{R}_{\text{dom}}(\eta) \lesssim \frac{1}{T\eta} + \frac{\eta}{n} \tag{a.7}$$

To find the optimal $\eta$ that minimizes this simplified expression, we take the derivative with respect to $\eta$ and set it to zero, which yields:

$$(\eta^*)^2 \propto \frac{1/T}{1/n} \implies \eta^* \propto \sqrt{\frac{n}{T}} \tag{a.8}$$

This demonstrates that while the exact value of the optimal learning rate depends on multiple constants, its scaling with respect to $n$ and $T$ is robustly determined by balancing the two dominant terms in the convergence bound. This justifies our choice of $\eta$ proportional to $\sqrt{n/T}$ to achieve the rate in Theorem 1. $\qquad\square$

### B.3 THEOREM ON SAMPLING NOISE (TERM A)

**Theorem a.3** (Sampling Noise Term $A = u^t - \overline{u}^t$). *Let $u^t$ be the global update at server iteration $t$, and $\overline{u}^t = \mathbb{E}_{\{\xi\}}[u^t|\mathcal{H}^t]$ be its expectation conditional on all information $\mathcal{H}^t$ (including stale models used for gradient computation) except the fresh data samples $\{\xi\}$ used to compute the gradients that form $u^t$. Under Assumptions 3 and 4, the expected squared norm of the sampling noise term $A = u^t - \overline{u}^t$ is bounded as follows for different asynchronous algorithms:*

1. *Vanilla ASGD(Mishchenko et al., 2022) & Delay-adaptive ASGD(Koloskova et al., 2022): If the server update $u^t = \nabla f_{j_t}(w^{t-\tau^t_{j_t}}; \xi^t_{j_t})$ is based on the stochastic gradient from a single client $j_t$ (performing $K = 1$ local step, local learning rate $\eta_l = 1$ effectively for the gradient itself), then*
$$\mathbb{E}\|A\|_2^2 \le \sigma^2$$

2. *FedBuff(Nguyen et al., 2022): If the server update $u^t = \frac{1}{m}\sum_{i\in\mathcal{M}_t}\Delta_i^t$, where $\Delta_i^t = \eta_l\sum_{k=0}^{K-1}g_{i,k}^t$ is derived from $K$ local SGD steps with local learning rate $\eta_l$ by clients in a set $\mathcal{M}_t$ of $m$ clients, and $g_{i,k}^t = \nabla f_i(w_{i,k}^{t-\tau_i^t}; \xi_{i,k}^t)$ is the stochastic gradient (computed on local model $w_{i,k}^{t-\tau_i^t}$ which is based on global model $w^{t-\tau_i^t}$), then*
$$\mathbb{E}\|A\|_2^2 \le \frac{K\eta_l^2\sigma^2}{m}$$

3. *$CA^2FL$(Wang et al., 2024b) (Cache-Aided Asynchronous Federated Learning): If the server update is $v^t = h^t + \frac{1}{m}\sum_{i\in\mathcal{S}_t}(\Delta_i^t - h_i^t)$, where $\Delta_i^t = \eta_l\sum_{k=0}^{K-1}g_{i,k}^t$ is the model difference from client $i \in \mathcal{S}_t$ (a set of $m$ clients) after $K$ local SGD steps with learning rate $\eta_l$, and $h^t, h_i^t$ are cached values. The sampling noise $A = v^t - \overline{v^t}$ where $\overline{v^t} = \mathbb{E}_{\{\xi_{i,k}^t\}}[v^t|\mathcal{H}^t]$ is bounded by:*
$$\mathbb{E}\|A\|_2^2 \le \frac{K\eta_l^2\sigma^2}{m}$$
   *(This arises because $A = \frac{1}{m}\sum_{i\in\mathcal{S}_t}(\Delta_i^t - \mathbb{E}[\Delta_i^t|\mathcal{H}^t])$.)*

4. *ACE (Ours): If the server update $u^t = \frac{1}{n}\sum_{i=1}^n\nabla f_i(w^{t-\tau_i^t}; \xi_i^{\kappa_i})$ is an average of the latest available (potentially stale) stochastic gradients from all $n$ clients (each performing $K = 1$ step for the gradient computation, with $\eta_l = 1$ for the gradient itself), then*
$$\mathbb{E}\|A\|_2^2 \le \frac{\sigma^2}{n}$$

*Proof.* The general structure for Term A is $A = u^t - \overline{u}^t$. We need to calculate $\mathbb{E}\|A\|_2^2$.

1. **Vanilla ASGD & Delay-adaptive ASGD:**
   Here, the update is $u^t = \nabla f_{j_t}(w^{t-\tau^t_{j_t}}; \xi^t_{j_t})$ from a single client $j_t$. The expected update, conditioned on the stale model $w^{t-\tau^t_{j_t}}$ (which is in $\mathcal{H}^t$), is $\overline{u}^t = \nabla F_{j_t}(w^{t-\tau^t_{j_t}})$. Thus, the sampling noise is $A = \nabla f_{j_t}(w^{t-\tau^t_{j_t}}; \xi^t_{j_t}) - \nabla F_{j_t}(w^{t-\tau^t_{j_t}})$. Then, its expected squared norm is:
$$\mathbb{E}\|A\|_2^2 = \mathbb{E}\left[\|\nabla f_{j_t}(w^{t-\tau^t_{j_t}}; \xi^t_{j_t}) - \nabla F_{j_t}(w^{t-\tau^t_{j_t}})\|_2^2\right]$$
   By Assumption 4 (Bounded Sampling Noise), this is directly bounded by $\sigma^2$.

2. **FedBuff:**
   The update is $u^t = \frac{1}{m}\sum_{i\in\mathcal{M}_t}\Delta_i^t = \frac{\eta_l}{m}\sum_{i\in\mathcal{M}_t}\sum_{k=0}^{K-1}g_{i,k}^t$. The expected update is $\overline{u}^t = \frac{\eta_l}{m}\sum_{i\in\mathcal{M}_t}\sum_{k=0}^{K-1}\nabla F_i(w_{i,k}^{t-\tau_i^t})$. The sampling noise is $A = u^t - \overline{u}^t = \frac{\eta_l}{m}\sum_{i\in\mathcal{M}_t}\sum_{k=0}^{K-1}(g_{i,k}^t - \nabla F_i(w_{i,k}^{t-\tau_i^t}))$. Let $\delta_{i,k}^t = g_{i,k}^t - \nabla F_i(w_{i,k}^{t-\tau_i^t})$. By Assumption 3

(Unbiased Stochastic Gradients), $\mathbb{E}[\delta_{i,k}^t | w_{i,k}^{t-\tau_i^t}] = 0$. We have:

$$\mathbb{E}\|A\|_2^2 = \mathbb{E}\left\| \frac{\eta_l}{m} \sum_{i \in \mathcal{M}_t} \sum_{k=0}^{K-1} \delta_{i,k}^t \right\|_2^2$$

Expanding the square:

$$\mathbb{E}\|A\|_2^2 = \frac{\eta_l^2}{m^2} \mathbb{E}\left[ \sum_{i \in \mathcal{M}_t} \sum_{k=0}^{K-1} \|\delta_{i,k}^t\|_2^2 + \sum_{\substack{(i,k) \neq (j,l) \\ i,j \in \mathcal{M}_t}} \langle \delta_{i,k}^t, \delta_{j,l}^t \rangle \right]$$

The samples $\xi_{i,k}^t$ used to compute $g_{i,k}^t$ are drawn independently for each client $i$ and each local step $k$. Therefore, for $(i,k) \neq (j,l)$, the terms $\delta_{i,k}^t$ and $\delta_{j,l}^t$ are (conditionally) independent given the respective models they were computed on. Since $\mathbb{E}[\delta_{i,k}^t | \mathcal{H}_k^t] = 0$ (where $\mathcal{H}_k^t$ includes $w_{i,k}^{t-\tau_i^t}$), the expectation of the cross terms $\langle \delta_{i,k}^t, \delta_{j,l}^t \rangle$ is zero. Specifically, $\mathbb{E}[\langle \delta_{i,k}^t, \delta_{j,l}^t \rangle] = \mathbb{E}[\mathbb{E}[\langle \delta_{i,k}^t, \delta_{j,l}^t \rangle | \mathcal{H}_{kl}^t]]$ where $\mathcal{H}_{kl}^t$ contains $w_{i,k}^{t-\tau_i^t}$ and $w_{j,l}^{t-\tau_j^t}$. If $i \neq j$, or $i = j$ but $k \neq l$ (implying different samples $\xi_{i,k}^t$ and $\xi_{j,l}^t$), then $\mathbb{E}[\delta_{i,k}^t | \mathcal{H}_{kl}^t]$ and $\mathbb{E}[\delta_{j,l}^t | \mathcal{H}_{kl}^t]$ are zero, making the cross term zero. Thus,

$$\mathbb{E}\|A\|_2^2 = \frac{\eta_l^2}{m^2} \sum_{i \in \mathcal{M}_t} \sum_{k=0}^{K-1} \mathbb{E}\|\delta_{i,k}^t\|_2^2$$

$$\leq \frac{\eta_l^2}{m^2} \sum_{i \in \mathcal{M}_t} \sum_{k=0}^{K-1} \sigma^2 \quad \text{(by Assumption 4)}$$

$$= \frac{\eta_l^2}{m^2} (m \cdot K \cdot \sigma^2) = \frac{K \eta_l^2 \sigma^2}{m}$$

3. **CA$^2$FL:**
   The global update is $v^t = h^t + \frac{1}{m} \sum_{i \in \mathcal{S}_t} (\Delta_i^t - h_i^t)$. The randomness from the current set of samples $\{\xi_{i,k}^t\}$ for $i \in \mathcal{S}_t$ comes from $\Delta_i^t = \eta_l \sum_{k=0}^{K-1} g_{i,k}^t$. The cached terms $h^t$ and $h_i^t$ are considered fixed with respect to the expectation over these current fresh samples (i.e., they are in $\mathcal{H}^t$). Let $A = v^t - \overline{v^t}$. Then $\overline{v^t} = \mathbb{E}_{\{\xi_{i,k}^t\}_{i \in \mathcal{S}_t}}[v^t | \mathcal{H}^t] = h^t + \frac{1}{m} \sum_{i \in \mathcal{S}_t} (\mathbb{E}_{\{\xi_{i,k}^t\}}[\Delta_i^t | \mathcal{H}^t] - h_i^t)$. Let $\overline{\Delta_i^t} = \mathbb{E}_{\{\xi_{i,k}^t\}}[\Delta_i^t | \mathcal{H}^t] = \eta_l \sum_{k=0}^{K-1} \nabla F_i(w_{i,k}^{t-\tau_i^t})$. So, $A = v^t - \overline{v^t} = \frac{1}{m} \sum_{i \in \mathcal{S}_t} (\Delta_i^t - \overline{\Delta_i^t})$. This simplifies to:

$$A = \frac{1}{m} \sum_{i \in \mathcal{S}_t} \eta_l \sum_{k=0}^{K-1} (g_{i,k}^t - \nabla F_i(w_{i,k}^{t-\tau_i^t})).$$

   This expression for $A$ is identical in form to that of FedBuff, with $\mathcal{S}_t$ corresponding to $\mathcal{M}_t$ and $m = |\mathcal{S}_t|$. Thus, the subsequent steps of the proof are the same as for FedBuff, yielding:

$$\mathbb{E}\|A\|_2^2 \leq \frac{K \eta_l^2 \sigma^2}{m}$$

4. **ACE:**
   Here $u^t = \frac{1}{n} \sum_{i=1}^n \nabla f_i(w^{t-\tau_i^t}; \xi_i^{\kappa_i})$ and $\overline{u}^t = \frac{1}{n} \sum_{i=1}^n \nabla F_i(w^{t-\tau_i^t})$. So, $A = \frac{1}{n} \sum_{i=1}^n (\nabla f_i(w^{t-\tau_i^t}; \xi_i^{\kappa_i}) - \nabla F_i(w^{t-\tau_i^t}))$. Let $\delta_i^t = \nabla f_i(w^{t-\tau_i^t}; \xi_i^{\kappa_i}) - \nabla F_i(w^{t-\tau_i^t})$. By Assumption 3, $\mathbb{E}[\delta_i^t | \mathcal{H}^t] = 0$. The samples $\xi_i^{\kappa_i}$ are drawn independently by each client $i$ for its respective gradient computation, conditional on $\mathcal{H}^t$ (which includes all $w^{t-\tau_j^t}$).

$$\mathbb{E}\|A\|_2^2 = \mathbb{E}\left\| \frac{1}{n} \sum_{i=1}^n \delta_i^t \right\|_2^2$$

$$= \frac{1}{n^2} \mathbb{E}\left[ \sum_{i=1}^n \|\delta_i^t\|_2^2 + \sum_{i \neq j} \langle \delta_i^t, \delta_j^t \rangle \right]$$

For $i \neq j$, $\delta_i^t$ and $\delta_j^t$ are (conditionally) independent given $\mathcal{H}^t$ because the samples $\xi_i^{\kappa_i}$ and $\xi_j^{\kappa_j}$ are drawn by different clients independently. Thus, $\mathbb{E}[\langle \delta_i^t, \delta_j^t \rangle | \mathcal{H}^t] = \langle \mathbb{E}[\delta_i^t | \mathcal{H}^t], \mathbb{E}[\delta_j^t | \mathcal{H}^t] \rangle = \langle 0, 0 \rangle = 0$. So the cross terms vanish:

$$\mathbb{E}\|A\|_2^2 = \frac{1}{n^2} \sum_{i=1}^n \mathbb{E}\|\delta_i^t\|_2^2$$

$$\leq \frac{1}{n^2} \sum_{i=1}^n \sigma^2 \quad \text{(by Assumption 4)}$$

$$= \frac{1}{n^2}(n \cdot \sigma^2) = \frac{\sigma^2}{n}$$

$\square$

### B.4 THEOREM ON BIAS ERROR (TERM B)

**Theorem a.4** (Bias Error Term $B = \overline{u}^t - \nabla F(w_{\text{stale}}^t)$)**.** *Let $u^t$ be the global update at server iteration $t$, $\overline{u}^t = \mathbb{E}_{\{\xi\}}[u^t | \mathcal{H}^t]$ its conditional expectation, and $\nabla F(w_{stale}^t) = \frac{1}{n} \sum_{i=1}^n \nabla F_i(w^{t-\tau_i^t})$. The expected squared norm of the bias error term $B$ is bounded as follows for different asynchronous algorithms, under relevant assumptions (primarily 2, 3, 4, and bounded data heterogeneity (BDH) assumption where applicable):*

1. ***Vanilla ASGD(Mishchenko et al., 2022) & Delay-Adaptive ASGD(Koloskova et al., 2022) & FedBuff(Nguyen et al., 2022):** If the server update $u^t = \frac{1}{m} \sum_{i \in \mathcal{M}_t} \Delta_i^t$, where $\Delta_i^t$ is derived from $K \geq 1$ local SGD steps with local learning rate $\eta_l$. Then $\overline{u}^t = \frac{1}{m} \sum_{i \in \mathcal{M}_t} \eta_l \sum_{k=0}^{K-1} \nabla F_i(w_{i,k}^{t-\tau_i^t})$. The bias $B = \overline{u}^t - \nabla F(w_{stale}^t)$ arises from both multiple local steps ($K \geq 1$) and partial client participation ($m < n$). A representative bound (cf. ACE paper's analysis of FedBuff) is:*

$$\mathbb{E}\|B\|^2 \lesssim \left( (\sigma^2 + K\zeta^2) + \sum_{i \in \mathcal{M}_t} \mathbb{E}\|\nabla F(w^{t-\tau_i^t})\|_2^2 \right) + (n-m)\left( \zeta^2 + \sum_{i=1}^n \overline{\mathbb{E}\|\nabla F(w^{t-\tau_i^t})\|_2^2} \right)$$

*The term reflects client drift due to local steps and bias due to averaging over a subset $m$ of clients.*

2. ***CA$^2$FL(Wang et al., 2024b):** The server update is $v^t = h^t + \frac{1}{m} \sum_{i \in \mathcal{S}_t}(\Delta_i^t - h_i^t)$, leading to $\overline{v^t} = h^t + \frac{1}{m} \sum_{i \in \mathcal{S}_t}(\overline{\Delta_i^t} - h_i^t)$. The bias $B = \overline{v^t} - \nabla F(w_{stale}^t)$ is reduced by the calibration mechanism but still affected by local steps and imperfect calibration if $m < n$. A representative bound (cf. ACE paper's analysis of CA$^2$FL) is:*

$$\mathbb{E}\|B\|^2 \lesssim \left( 1 + (1 - \frac{m}{n})^2 \right) \left( (\sigma^2 + K\zeta^2) + \sum_{i=1}^n \overline{\mathbb{E}\|\nabla F(w^{t-\tau_i^t})\|_2^2} \right)$$

3. ***ACE (Ours):** The server update $u^t = \frac{1}{n} \sum_{i=1}^n \nabla f_i(w^{t-\tau_i^t}; \xi_i^{\kappa_i})$.*

   *Which leads to*
$$\mathbb{E}\|B\|_2^2 = 0.$$

*Proof.* The general structure for Term B is $B = \overline{u}^t - \nabla F(w_{\text{stale}}^t)$. We need to calculate $\mathbb{E}\|B\|_2^2$.

1. **Vanilla ASGD & Delay-Adaptive ASGD & FedBuff:**
   Deviation of the sum of true local gradients over $K$ steps from $K$ times the initial true local

gradient, averaged over participating clients.

$$B = \overline{u}^t - \nabla F(w_{\text{stale}}^t)$$

$$= \frac{1}{mK} \sum_{i \in \mathcal{M}_t} \sum_{k=0}^{K-1} \nabla F_i(w_{i,k}^{t-\tau_i^t}) - \frac{1}{n} \sum_{i=1}^{n} \nabla F_i(w^{t-\tau_i^t})$$

$$= \underbrace{\left(\frac{1}{m} - \frac{1}{n}\right) \sum_{i \in \mathcal{M}_t} \nabla F_i(w^{t-\tau_i^t})}_{\text{Part 1}} - \underbrace{\frac{1}{n} \sum_{i \notin \mathcal{M}_t} \nabla F_i(w^{t-\tau_i^t})}_{\text{Part 2}}$$

$$+ \underbrace{\frac{1}{mK} \sum_{i \in \mathcal{M}_t} \left(\sum_{k=0}^{K-1} \nabla F_i(w_{i,k}^{t-\tau_i^t}) - K\nabla F_i(w^{t-\tau_i^t})\right)}_{\mathcal{E}_{\text{drift}}}$$

By Lemma a.3, $\mathbb{E}\|B\|_2^2 \le 3\mathbb{E}\|\text{Part 1}\|_2^2 + 3\mathbb{E}\|\text{Part 2}\|_2^2 + 3\mathbb{E}\|\mathcal{E}_{\text{drift}}\|_2^2$

For the *partial participation bias* Part 1 and 2, the client subset $\mathcal{S}$ to determine the sum is $\mathcal{M}_t$ or $[n]/\mathcal{M}_t$

$$\mathbb{E}\|\sum_{i \in \mathcal{S}} \nabla F_i(w^{t-\tau_i^t})\|_2^2 = \mathbb{E}\|\sum_{i \in \mathcal{S}} \nabla F_i(w^{t-\tau_i^t}) - \sum_{i \in \mathcal{S}} \nabla F(w^{t-\tau_i^t}) + \sum_{i \in \mathcal{S}} \nabla F(w^{t-\tau_i^t})\|_2^2$$

$$\le \underbrace{2\mathbb{E}\|\sum_{i \in \mathcal{S}} \nabla F_i(w^{t-\tau_i^t}) - \sum_{i \in \mathcal{S}} \nabla F(w^{t-\tau_i^t})\|_2^2}_{\text{Can be determined by BDH Assumption and Lemma a.3}}$$

$$+ 2\mathbb{E}\|\sum_{i \in \mathcal{S}} \nabla F(w^{t-\tau_i^t})\|_2^2 \quad \text{(By Lemma a.3)}$$

$$\le 2|\mathcal{S}| \sum_{i \in \mathcal{S}} \zeta^2 + 2|\mathcal{S}| \sum_{i \in \mathcal{S}} \mathbb{E}\|\nabla F(w^{t-\tau_i^t})\|_2^2$$

Therefore,

$$\mathbb{E}\|\text{Part 1}\|_2^2 \le 2\left(\frac{1}{m} - \frac{1}{n}\right)^2 \left(m^2\zeta^2 + m \sum_{i \in \mathcal{M}_t} \mathbb{E}\|\nabla F(w^{t-\tau_i^t})\|_2^2\right)$$

$$\mathbb{E}\|\text{Part 2}\|_2^2 \le \frac{2}{n^2} \left((n-m)^2\zeta^2 + (n-m) \sum_{i \notin \mathcal{M}_t} \mathbb{E}\|\nabla F(w^{t-\tau_i^t})\|_2^2\right)$$

For the drift error $\mathcal{E}_{\text{drift}}$,

$$\mathcal{E}_{\text{drift}} = \frac{1}{mK} \sum_{i \in \mathcal{M}_t} \left(\sum_{k=0}^{K-1} \nabla F_i(w_{i,k}^{t-\tau_i^t}) - K\nabla F_i(w^{t-\tau_i^t})\right)$$

Taking its squared norm and expectation:

$$\mathbb{E}\|\mathcal{E}_{\text{drift}}\|_2^2 = \mathbb{E}\left\|\frac{1}{mK} \sum_{i \in \mathcal{M}_t} \sum_{k=0}^{K-1} \left(\nabla F_i(w_{i,k}^{t-\tau_i^t}) - \nabla F_i(w^{t-\tau_i^t})\right)\right\|_2^2$$

$$\le \frac{1}{mK^2} \sum_{i \in \mathcal{M}_t} \mathbb{E}\left\|\sum_{k=0}^{K-1} (\nabla F_i(w_{i,k}^{t-\tau_i^t}) - \nabla F_i(w^{t-\tau_i^t}))\right\|_2^2 \quad \text{(by Lemma a.3 for outer sum)}$$

$$\le \frac{1}{mK} \sum_{i \in \mathcal{M}_t} \sum_{k=0}^{K-1} \mathbb{E}\|\nabla F_i(w_{i,k}^{t-\tau_i^t}) - \nabla F_i(w^{t-\tau_i^t})\|_2^2 \quad \text{(by Lemma a.3 for inner sum)}$$

$$\le \frac{L^2}{mK} \sum_{i \in \mathcal{M}_t} \sum_{k=0}^{K-1} \mathbb{E}\|w_{i,k}^{t-\tau_i^t} - w^{t-\tau_i^t}\|_2^2 \quad \text{(by L-smoothness Assumption 2)}$$

By Lemma a.5:

$$\mathbb{E}\|w_{i,k'}^{t-\tau_i^t} - w^{t-\tau_i^t}\|^2 \leq 5K\eta_l^2(\sigma^2 + 6K\zeta^2) + 30K^2\eta_l^2\mathbb{E}[\|\nabla F(w^{t-\tau_i^t})\|^2]$$

Merging all the pieces. For simplicity of notation, the terms in the partial participation bias involve $\mathbb{E}[\|\nabla F(\cdot)\|^2$ are merged in an (weighted) average sense:

$$\mathbb{E}\|B\|^2 \lesssim \left((\sigma^2 + K\zeta^2) + \sum_{i\in\mathcal{M}_t} \mathbb{E}\|\nabla F(w^{t-\tau_i^t})\|_2^2\right) + (n-m)\left(\zeta^2 + \sum_{i=1}^n \overline{\mathbb{E}\|\nabla F(w^{t-\tau_i^t})\|_2^2}\right)$$

2. **CA$^2$FL:** The server update is $v^t = h^t + \frac{1}{m}\sum_{i\in\mathcal{S}_t}(\Delta_i^t - h_i^t)$, leading to $\overline{v^t} = h^t + \frac{1}{m}\sum_{i\in\mathcal{S}_t}\overline{(\Delta_i^t - h_i^t)}$. The model delay of the non-participating client $i$ at server iteration $t$ is denoted as $\rho_i^t$, since $\zeta$ is used for denoting the BDH assumption bound.

$$B = \overline{u}^t - \nabla F(w_{\text{stale}}^t)$$

$$= \frac{1}{mK}\sum_{i\in\mathcal{S}_t}\sum_{k=0}^{K-1}[\nabla F_i(w_{i,k}^{t-\tau_i^t}) - \nabla F_i(w^{t-\tau_i^t})] \quad \text{(Denoted as } X_1\text{)}$$

$$+ \left(\frac{1}{nK} - \frac{1}{mK}\right)\sum_{i\in\mathcal{S}_t}\sum_{k=0}^{K-1}[\nabla F_i(w_{i,k}^{t-\rho_i^t}) - \nabla F_i(w^{t-\rho_i^t})] \quad \text{(Denoted as } X_2\text{)}$$

$$+ \frac{1}{nK}\sum_{i\notin\mathcal{S}_t}\sum_{k=0}^{K-1}[\nabla F_i(w_{i,k}^{t-\rho_i^t}) - \nabla F_i(w^{t-\rho_i^t})] \quad \text{(Denoted as } X_3\text{)}$$

The core of bounding $\mathbb{E}\|B\|^2$ involves:

(a) Using $\|X_1 + X_2 + X_3\|^2 \leq 3(\|X_1\|^2 + \|X_2\|^2 + \|X_3\|^2)$, where $X_1, X_2, X_3$ are the three main summations in $B$.

(b) For each component, say $X_1 = \frac{1}{mK}\sum_{i\in\mathcal{S}_t}\sum_{k=0}^{K-1}[\nabla F_i(w^{t-\tau_i^t}) - \nabla F_i(w_{i,k}^{t-\tau_i^t})]$:

$$\mathbb{E}\|X_1\|^2 \leq \frac{1}{(mK)^2}\mathbb{E}\left\|\sum_{i\in\mathcal{S}_t}\sum_{k=0}^{K-1}[\nabla F_i(w^{t-\tau_i^t}) - \nabla F_i(w_{i,k}^{t-\tau_i^t})]\right\|^2$$

$$\leq \frac{1}{mK^2}\sum_{i\in\mathcal{S}_t}\mathbb{E}\left\|\sum_{k=0}^{K-1}[\nabla F_i(w^{t-\tau_i^t}) - \nabla F_i(w_{i,k}^{t-\tau_i^t})]\right\|^2 \quad \text{(by Lemma a.3 for outer sum)}$$

$$\leq \frac{1}{mK}\sum_{i\in\mathcal{S}_t}\sum_{k=0}^{K-1}\mathbb{E}\|\nabla F_i(w^{t-\tau_i^t}) - \nabla F_i(w_{i,k}^{t-\tau_i^t})\|_2^2 \quad \text{(by Lemma a.3 for inner sum)}$$

$$\leq \frac{L^2}{mK}\sum_{i\in\mathcal{S}_t}\sum_{k=0}^{K-1}\mathbb{E}\|w^{t-\tau_i^t} - w_{i,k}^{t-\tau_i^t}\|_2^2 \quad \text{(by L-smoothness Assumption 2)}$$

(c) By Lemma a.5 (adapting notation: $w^{\text{start}} = w^{t-\tau_i^t}$ or $w^{t-\rho_i^t}$):

$$\mathbb{E}[\|w_{i,k'}^{\text{start}} - w^{\text{start}}\|^2] \leq 5K\eta_l^2(\sigma^2 + 6K\zeta^2) + 30K^2\eta_l^2\mathbb{E}[\|\nabla F(w^{\text{start}})\|^2]$$

Note that

$$\underbrace{\frac{1}{m^2}}_{\text{Coefficient}} \cdot \underbrace{m}_{|\mathcal{S}_t|} \cdot \underbrace{m}_{\text{Sum of }|\mathcal{S}_t|\text{ components}} + \left(\frac{1}{m} - \frac{1}{n}\right)^2 \cdot m \cdot m + \frac{1}{m^2} \cdot (m-n) \cdot (m-n)$$

$$= 1 + 2(1 - \frac{m}{n})^2$$

For simplicity of notation, the terms involve $\mathbb{E}[\|\nabla F(\cdot)\|^2$ are merged in an (weighted) average sense (we treat two sources of delay $\tau, \rho$ equivalently):

$$\mathbb{E}\|B\|^2 \lesssim \left(1 + (1 - \frac{m}{n})^2\right)(\sigma^2 + K\zeta^2) + \frac{1}{m}\sum_{i \in \mathcal{S}_t} \mathbb{E}\|\nabla F(w^{t-\tau_i^t})\|_2^2$$

$$+ m\left(\frac{1}{m} - \frac{1}{n}\right)^2 \sum_{i \in \mathcal{S}_t} \mathbb{E}\|\nabla F(w^{t-\rho_i^t})\|_2^2$$

$$+ \frac{(n-m)^2}{m^2} \sum_{i \notin \mathcal{S}_t} \mathbb{E}\|\nabla F(w^{t-\rho_i^t})\|_2^2$$

$$\lesssim \left(1 + (1 - \frac{m}{n})^2\right)\left((\sigma^2 + K\zeta^2) + \sum_{i=1}^{n} \overline{\mathbb{E}\|\nabla F(w^{t-\tau_i^t})\|_2^2}\right)$$

3. **ACE:** The server update $u^t = \frac{1}{n}\sum_{i=1}^{n} \nabla f_i(w^{t-\tau_i^t}; \xi_i^{\kappa_i})\ (t - \tau_i^t < \kappa_i \leq t)$.

Thus, $\overline{u}^t = \frac{1}{n}\sum_{i=1}^{n} \nabla F_i(w^{t-\tau_i^t})$. Therefore:

$$B = \overline{u}^t - \nabla F(w_{\text{stale}}^t) = \frac{1}{n}\sum_{i=1}^{n} \nabla F_i(w^{t-\tau_i^t}) - \frac{1}{n}\sum_{i=1}^{n} \nabla F_i(w^{t-\tau_i^t}) = 0$$

Which leads to

$$\mathbb{E}\|B\|_2^2 = 0.$$

$\square$

### B.5 THEOREM ON DELAY ERROR (TERM C)

**Theorem a.5** (Delay Error Term $C = \nabla F(w_{\text{stale}}^t) - \nabla F(w^t)$). *Let $w^t$ be the global model at server iteration $t$, and $w_{\text{stale}}^t = \{w^{t-\tau_i^t}\}_{i=1}^{n}$ be the collection of stale models used by clients, where $\tau_i^t$ is the information delay for client $i$. The expected squared norm of the delay error term $C$ is $\mathbb{E}\|C\|_2^2 = \mathbb{E}\|\nabla F(w_{\text{stale}}^t) - \nabla F(w^t)\|_2^2$. Under Assumption 2 (L-Smoothness), this can be bounded in terms of model drift $D_i^t = \mathbb{E}\|w^{t-\tau_i^t} - w^t\|_2^2$:*

$$\mathbb{E}\|C\|_2^2 = \mathbb{E}\left\|\frac{1}{n}\sum_{i=1}^{n}(\nabla F_i(w^{t-\tau_i^t}) - \nabla F_i(w^t))\right\|_2^2 \leq \frac{1}{n}\sum_{i=1}^{n} \mathbb{E}\|\nabla F_i(w^{t-\tau_i^t}) - \nabla F_i(w^t)\|_2^2 \leq \frac{L^2}{n}\sum_{i=1}^{n} D_i^t$$

*The model drift $D_i^t = \mathbb{E}\|\sum_{s=t-\tau_i^t}^{t-1} \eta u^s\|_2^2$ (where $u^s$ is the server update at step $s$) is bounded as follows for different asynchronous algorithms, under relevant assumptions (including Assumption 5 for $\tau_{max}$, and bounded data heterogeneity $\zeta^2$ where applicable):*

1. ***Vanilla ASGD(Mishchenko et al., 2022), Delay-Adaptive ASGD(Koloskova et al., 2022), FedBuff(Nguyen et al., 2022):*** *If the server update $u^s$ is formed from a subset $\mathcal{M}_s$ of $m \leq n$ clients, potentially with $K \geq 1$ local steps and local learning rate $\eta_l$:*

$$D_i^t \lesssim \tau_i^t \eta^2 \eta_l^2 \left(\frac{K\sigma^2}{m} + \frac{1}{m}\sum_{s'=t-\tau_i^t}^{t-1} \mathbb{E}\|\nabla F(w_{stale}^{s'})\|_2^2 + (n-m)K^2\zeta^2\right)$$

*The term $(n-m)K^2\zeta^2$ highlights drift arising from client heterogeneity when $m < n$.*

2. ***CA$^2$FL(Wang et al., 2024b):*** *If the server update $u^s$ is from a subset $\mathcal{M}_s$ of $m$ clients, calibrated using all-client history, with $K \geq 1$ local steps and local learning rate $\eta_l$:*

$$D_i^t \lesssim \tau_i^t \eta^2 \eta_l^2 (1 + (1 - \frac{m}{n})^2)\left(\frac{K\sigma^2}{m} + \sum_{s'=t-\tau_i^t}^{t-1} \overline{\mathbb{E}\|\nabla F(w_{stale}^{s'})\|_2^2}\right)$$

*Calibration aims to remove the direct $\zeta^2$ term from partial participation bias found in FedBuff's drift.*

3. **ACE(Ours)**: *If the server update $u^s$ averages information from all $n$ clients ($m = n$), with $K = 1$ effective local step for the gradient:*

$$D_i^t \lesssim \tau_i^t \eta^2 \left( \frac{\sigma^2}{n} + \sum_{s'=t-\tau_i^t}^{t-1} \mathbb{E}\|\nabla F(w_{stale}^{s'})\|_2^2 \right)$$

*Here, the $\zeta^2$ term from partial participation is absent because $u^s$ in ACEaverages information from all $n$ clients.*

*Note: The core idea is that the structure of $u^s$ (full vs. partial aggregation, number of local steps $K$) influences the terms within $D_i^t$, and thus Term C.*

*Proof.* The general structure for Term C is $C = \nabla F(w_{stale}^t) - \nabla F(w^t)$. We need to calculate $\mathbb{E}\|C\|_2^2$, w.r.t. the original definition or w.r.t. the model drift $D_i^t$.

1. **Vanilla ASGD & Delay-Adaptive ASGD & FedBuff:**
   The update is $u^t = \frac{1}{m} \sum_{i \in \mathcal{M}_t} \Delta_i^t = \frac{\eta_l}{m} \sum_{i \in \mathcal{M}_t} \sum_{k=0}^{K-1} g_{i,k}^t$. Note that by Lemma a.6, the sum of the cross-iteration gradient error is zero. The model drift can be calculated as:

$$D_i^t = \mathbb{E}[\|w^t - w^{t-\tau_t^l}\|^2] = \mathbb{E}\left[ \left\| \sum_{s=t-\tau_t^l}^{t-1} (w^{s+1} - w^s) \right\|^2 \right]$$

$$= \mathbb{E}\left[ \left\| \eta \sum_{s=t-\tau_t^l}^{t-1} \frac{1}{m} \sum_{j \in \mathcal{M}_s} \sum_{k=0}^{K-1} \eta^l g_{s-\tau_j^s,k}^j \right\|^2 \right]$$

$$= \mathbb{E}\left[ \left\| \eta \sum_{s=t-\tau_t^l}^{t-1} \frac{1}{m} \sum_{j \in \mathcal{M}_s} \sum_{k=0}^{K-1} \eta^l (g_{s-\tau_j^s,k}^j - \nabla F_j(w_j^{s-\tau_j^s,k}) + \nabla F_j(w_j^{s-\tau_j^s,k})) \right\|^2 \right]$$

$$= 2\mathbb{E}\left[ \left\| \eta \underbrace{\sum_{s=t-\tau_t^l}^{t-1}}_{\text{Expand by Lemma a.6}} \frac{1}{m} \sum_{j \in \mathcal{M}_s} \sum_{k=0}^{K-1} \eta^l (g_{s-\tau_j^s,k}^j - \nabla F_j(w_j^{s-\tau_j^s,k})) \right\|^2 \right]$$

$$+ 2\mathbb{E}\left[ \left\| \eta \sum_{s=t-\tau_t^l}^{t-1} \frac{1}{m} \sum_{j \in \mathcal{M}_s} \sum_{k=0}^{K-1} \eta^l \nabla F_j(w_j^{s-\tau_j^s,k}) \right\|^2 \right]$$

$$\leq \frac{2\tau_i^t K \eta^2 \eta_l^2}{m} \sigma^2 + \frac{2\tau_i^t \eta^2 \eta_l^2}{m^2} \sum_{s=t-\tau_t^l}^{t-1} \mathbb{E}\left[ \left\| \sum_{j \in \mathcal{M}_s} \sum_{k=0}^{K-1} \nabla F_j(w_j^{s-\tau_j^s,k}) \right\|^2 \right]$$

Note that we have

$$\left\| \sum_{i=1}^n \sum_{k=0}^{K-1} \nabla F_i(w_i^{t,k}) \right\|^2 = \sum_{i=1}^n \left\| \sum_{k=0}^{K-1} \nabla F_i(w_i^{t,k}) \right\|^2 + \sum_{i \neq j} \left\langle \sum_{k=0}^{K-1} \nabla F_i(w_i^{t,k}), \sum_{k=0}^{K-1} \nabla F_j(w_j^{t,k}) \right\rangle$$

$$= \sum_{i=1}^n n \left\| \sum_{k=0}^{K-1} \nabla F_i(w_i^{t,k}) \right\|^2 - \frac{1}{2} \sum_{i \neq j} \left\| \sum_{k=0}^{K-1} \nabla F_i(w_i^{t,k}) - \sum_{k=0}^{K-1} \nabla F_j(w_j^{t,k}) \right\|^2,$$

Where the second equation, $\|\sum_{i=1}^n x_i\|^2 = \sum_{i=1}^n n \|x_i\|^2 - \frac{1}{2} \sum_{i \neq j} \|x_i - x_j\|^2$, holds due to Lemma a.2. And $\langle a, b \rangle = \frac{1}{2} (\|a\|^2 + \|b\|^2 - \|a - b\|^2)$, holds due to Lemma a.1

For simplicity, we assume a uniform partial participation of the clients, i.e. $\mathbb{P}\{i \in \mathcal{M}_t\} = \frac{m}{n}, \mathbb{P}\{i, j \in \mathcal{M}_t\} = \frac{m(m-1)}{n(n-1)}$.

$$
\left\| \sum_{j \in \mathcal{M}_s} \sum_{k=0}^{K-1} \nabla F_j(w_j^{s-\tau_j^s,k}) \right\|^2 = \left\| \sum_{i=1}^{n} \sum_{k=0}^{K-1} \mathbb{P}\{i \in \mathcal{M}_t\} \nabla F_i(w_i^{t,k}) \right\|^2
$$

$$
= \sum_{i=1}^{n} \mathbb{P}\{i \in \mathcal{M}_t\} \left\| \sum_{k=0}^{K-1} \nabla F_i(w_i^{t,k}) \right\|^2 + \sum_{i \neq j}^{n} \mathbb{P}\{i, j \in \mathcal{M}_t\} \left\langle \sum_{k=0}^{K-1} \nabla F_i(w_i^{t,k}), \sum_{k=0}^{K-1} \nabla F_j(w_j^{t,k}) \right\rangle
$$

$$
= \frac{m}{n} \sum_{i=1}^{n} \left\| \sum_{k=0}^{K-1} \nabla F_i(w_i^{t,k}) \right\|^2 + \frac{m(m-1)}{n(n-1)} \sum_{i \neq j} \left\langle \sum_{k=0}^{K-1} \nabla F_i(w_i^{t,k}), \sum_{k=0}^{K-1} \nabla F_j(w_j^{t,k}) \right\rangle
$$

$$
= \frac{m^2}{n} \sum_{i=1}^{n} \left\| \sum_{k=0}^{K-1} \nabla F_i(w_i^{t,k}) \right\|^2 - \frac{m(m-1)}{2n(n-1)} \sum_{i \neq j} \left\| \sum_{k=0}^{K-1} \nabla F_i(w_i^{t,k}) - \sum_{k=0}^{K-1} \nabla F_j(w_j^{t,k}) \right\|^2
$$

$$
= \frac{m(n-m)}{n(n-1)} \sum_{i=1}^{n} \left\| \sum_{k=0}^{K-1} \nabla F_i(w_i^{t,k}) \right\|^2 + \frac{m(m-1)}{n(n-1)} \left\| \sum_{i=1}^{n} \sum_{k=0}^{K-1} \nabla F_i(w_i^{t,k}) \right\|^2 ,
$$

Thus, by Lemma a.5 :

$$
\mathbb{E} \left[ \left\| \sum_{j \in \mathcal{M}_s} \sum_{k=0}^{K-1} \nabla F_j(w_j^{s-\tau_j^s,k}) \right\|^2 \right]
$$

$$
= \frac{m(m-1)}{n(n-1)} \sum_{j=1}^{n} \mathbb{E} \left[ \left\| \sum_{k=0}^{K-1} \nabla F_j(w_j^{s-\tau_j^s,k}) \right\|^2 \right] + \frac{m(m-1)}{n(n-1)} \mathbb{E} \left[ \left\| \sum_{j=1}^{n} \sum_{k=0}^{K-1} \nabla F_j(w_j^{s-\tau_j^s,k}) \right\|^2 \right]
$$

$$
\leq \frac{m(n-m)}{n(n-1)} \left[ 15nK^3\eta_l^2(\sigma^2 + 6K\zeta^2) + (90K^4L^2\eta_l^2 + 3K^2) \sum_{j=1}^{n} \mathbb{E}[\|\nabla F(w^{s-\tau_j^s})\|^2] + 3nK^2\zeta^2 \right]
$$

$$
+ \frac{2m(m-1)}{n(n-1)} \sum_{j=1}^{n} \mathbb{E} \left[ \left\| \sum_{k=0}^{K-1} \nabla F_j(w_j^{s-\tau_j^s,k}) - \sum_{k=0}^{K-1} \nabla F_j(w_j^{s-\tau_s^l,k}) \right\|^2 \right]
$$

$$
+ \frac{2m(m-1)}{n-1} K^2 \sum_{j=1}^{n} \mathbb{E}[\|\nabla F(w^{s-\tau_j^s})\|^2] \qquad \text{(Lemma a.3)}
$$

$$
\leq \frac{m(n-m)}{n(n-1)} \left[ 15nK^3\eta_l^2(\sigma^2 + 6K\zeta^2) + (90K^4L^2\eta_l^2 + 3K^2) \sum_{j=1}^{n} \mathbb{E}[\|\nabla F(w^{s-\tau_j^s})\|^2] + 3nK^2\zeta^2 \right]
$$

$$
+ \frac{2m(m-1)KL^2}{n-1} \sum_{j=1}^{n} \mathbb{E}[\|w_j^{s-\tau_j^s,k} - w_j^{s-\tau_s^l,k}\|^2] \qquad \text{(L-smoothness, Assumption 2)}
$$

$$
+ \frac{2m(m-1)K^2}{n-1} \sum_{j=1}^{n} \mathbb{E}[\|\nabla F(w^{s-\tau_j^s})\|^2]
$$

$$
\leq \left[ \frac{3m(n-m)}{n(n-1)} + \frac{2nm(m-1)}{n(n-1)} \right] \left[ 5K^3L^2\eta_l^2(\sigma^2 + 6K\zeta^2) + (30K^4L^2\eta_l^2 + K^2)\frac{1}{n} \sum_{j=1}^{n} \mathbb{E}[\|\nabla F(w^{s-\tau_j^s})\|^2] \right]
$$

$$
+ \frac{3m(n-m)}{n-1} K^2\zeta^2 ,
$$

Take back to $D_i^t = \mathbb{E}\left[\|w^t - w^{t-\tau_i^t}\|^2\right]$. For the simplicity of notations, the terms are merged in an (weighted) average sense,

$$D_i^t = \mathbb{E}\left[\|w^t - w^{t-\tau_i^t}\|^2\right] \leq \frac{2\tau_i^t K \eta^2 \eta_l^2}{m}\sigma^2 + \frac{2\tau_i^t \eta^2 \eta_l^2}{m^2}\sum_{s=t-\tau_i^t}^{t-1}\left\{\left[\frac{3m(n-m)}{n-1} + \frac{2nm(m-1)}{n-1}\right]\right.$$

$$\cdot\left[5K^3 L^2 \eta_l^2(\sigma^2 + 6K\zeta^2) + (30K^4 L^2 \eta_l^2 + K^2)\mathbb{E}\left[\|\nabla F(w^{s-\tau_j^s})\|^2\right]\right]$$

$$+ \left.\frac{3m(n-m)}{n-1}K^2\zeta^2\right\}$$

$$\lesssim \tau_i^t \eta^2 \eta_l^2 \left(\frac{K\sigma^2}{m} + \frac{1}{m}\sum_{s'=t-\tau_i^t}^{t-1}\overline{\mathbb{E}\|\nabla F(w_{\text{stale}}^{s'})\|_2^2} + (n-m)K^2\zeta^2\right)$$

2. **CA$^2$FL:**

The server update is $v^t = h^t + \frac{1}{m}\sum_{i\in\mathcal{S}_t}(\Delta_i^t - h_i^t)$, leading to $\overline{v^t} = h^t + \frac{1}{m}\sum_{i\in\mathcal{S}_t}(\overline{\Delta_i^t} - h_i^t)$.

The model delay of the non-participating client $i$ at server iteration $t$ is denoted as $\rho_i^t$, since $\zeta$ is used for denoting the BDH assumption bound.

$$v^t = \frac{1}{n}\sum_{i\notin S_t}h_i^{t-1} + \frac{1}{n}\sum_{i\in S_t}h_i^{t-1} + \frac{1}{m}\sum_{i\in S_t}\left(\Delta_i^{t-\tau_i^t} - h_i^{t-1}\right)$$

$$= \frac{1}{n}\sum_{i\notin S_t}h_i^{t-1} + \sum_{i\in S_t}\left[\left(\frac{1}{n}-\frac{1}{m}\right)h_i^{t-1} + \frac{1}{m}\Delta_i^{t-\tau_i^t}\right]$$

Take into the definition of $\mathbb{E}\|C\|^2 = \mathbb{E}\|\nabla F(w_{\text{stale}}^t) - \nabla F(w^t)\|^2$,

$$\mathbb{E}\left[\left\|\frac{1}{m}\sum_{i\in S_t}[\nabla F_i(w^t) - \nabla F_i(w^{t-\tau_i^t})] + \left(\frac{1}{n}-\frac{1}{m}\right)\sum_{i\in S_t}[\nabla F_i(w^t) - \nabla F_i(w^{t-\rho_i^t})]\right.\right.$$

$$\left.\left. + \frac{1}{n}\sum_{i\notin S_t}[\nabla F_i(w^t) - \nabla F_i(w^{t-\rho_i^t})]\right\|^2\right]$$

$$\leq \frac{3}{m}\mathbb{E}\left[\sum_{i\in S_t}\|\nabla F_i(w^t) - \nabla F_i(w^{t-\tau_i^t})\|^2\right] + \frac{3(n-m)^2}{n^2 m}\mathbb{E}\left[\sum_{i\in S_t}\|\nabla F_i(w^t) - \nabla F_i(w^{t-\rho_i^t})\|^2\right]$$

$$+ \frac{3(n-m)}{n^2}\mathbb{E}\left[\sum_{i\notin S_t}\|\nabla F_i(w^t) - \nabla F_i(w^{t-\rho_i^t})\|^2\right]$$

$$\leq \frac{3L^2}{m}\mathbb{E}\left[\sum_{i\in S_t}\left\|\sum_{s=t-\tau_i^t}^{t-1}(w^{s+1} - w^s)\right\|^2\right] + \frac{3(n-m)^2 L^2}{n^2 m}\mathbb{E}\left[\sum_{i\in S_t}\left\|\sum_{s=t-\rho_i^t}^{t-1}(w^{s+1} - w^s)\right\|^2\right]$$

$$+ \frac{3(n-m)L^2}{n^2}\mathbb{E}\left[\sum_{i\notin S_t}\left\|\sum_{s=t-\rho_i^t}^{t-1}(w^{s+1} - w^s)\right\|^2\right]$$

Note that

$$\underbrace{\frac{1}{m}}_{\text{Coefficient}} \cdot \underbrace{m}_{|\mathcal{S}_t|} + \frac{(n-m)^2}{n^2 m}\cdot m + \frac{n-m}{n^2}\cdot(n-m) = 1 + 2\left(1 - \frac{m}{n}\right)^2.$$

Similar to the above proof for the partial participation methods (Vanilla ASGD & Delay-Adaptive ASGD & FedBuff); and note that by Lemma a.6, the sum of the cross-iteration

gradient error is zero:

$$\mathbb{E}\left[\left\|\sum_{s=t-\tau_t^i}^{t-1}(w^{s+1}-w^s)\right\|^2\right] = \mathbb{E}\left[\|w^t - w^{t-\tau_t^i}\|^2\right]$$

$$\leq \frac{2\tau_i^t K \eta^2 \eta_l^2}{m}\sigma^2 + \frac{2\tau_i^t \eta^2 \eta_l^2}{m^2}\sum_{s=t-\tau_t^i}^{t-1}\mathbb{E}\left[\left\|\sum_{j\in S_s}\sum_{k=0}^{K-1}\left(\frac{1}{m}\nabla F_i(w_i^{s-\tau_j^s,k})\right.\right.\right.$$

$$\left.\left.\left.+\left(\frac{1}{n}-\frac{1}{m}\right)\nabla F_i(w_i^{s-\rho_j^s,k})\right)\frac{1}{n}\sum_{j\notin S_s}\sum_{k=0}^{K-1}\nabla F_i(w_i^{s-\rho_j^s,k})\right\|^2\right].$$

Similarly,

$$\mathbb{E}\left[\left\|\sum_{s=t-\rho_i^t}^{t-1}(w^{s+1}-w^s)\right\|^2\right] = \mathbb{E}\left[\|w^t - w^{t-\rho_i^t}\|^2\right]$$

$$\leq \frac{2\rho_i^t K \eta^2 \eta_l^2}{m}\sigma^2 + \frac{2\rho_i^t \eta^2 \eta_l^2}{m^2}\sum_{s=t-\rho_i^t}^{t-1}\mathbb{E}\left[\left\|\sum_{j\in S_s}\sum_{k=0}^{K-1}\left(\frac{1}{m}\nabla F_i(w_i^{s-\tau_j^s,k})\right.\right.\right.$$

$$\left.\left.\left.+\left(\frac{1}{n}-\frac{1}{m}\right)\nabla F_i(w_i^{s-\rho_j^s,k})\right)\frac{1}{n}\sum_{j\notin S_s}\sum_{k=0}^{K-1}\nabla F_i(w_i^{s-\rho_j^s,k})\right\|^2\right].$$

Merging all the pieces. For the simplicity of notations, the terms are merged in an (weighted) average sense (we treat two sources of delay $\tau, \rho$ equivalently):

$$D_i^t \leq \tau_i^t \eta^2 \eta_l^2 (1 + (1-\frac{m}{n})^2)\left(\frac{K\sigma^2}{m} + \sum_{s'=t-\tau_i^t}^{t-1}\overline{\mathbb{E}\|\nabla F(w_{\text{stale}}^{s'})\|_2^2}\right)$$

3. **ACE (Ours):** For $\mathbb{E}\|w^t - w^{t-\tau_i^t}\|^2$, it can be decomposed as a telescoping sum:

$$w^t - w^{t-\tau_i^t} = \sum_{s=t-\tau_i^t}^{t-1}(w^{s+1}-w^s) = \sum_{s=t-\tau_i^t}^{t-1}(-\eta u^s).$$

$$\mathbb{E}\|w^t - w^{t-\tau_i^t}\|^2 = \eta^2 \mathbb{E}\left\|\sum_{s=t-\tau_i^t}^{t-1}u^s\right\|^2$$

Decompose $u^s = (u^s - \bar{u}^s) + \bar{u}^s$ and by Lemma a.3,

$$\eta^2 \mathbb{E}\left\|\sum_{s=t-\tau_i^t}^{t-1}((u^s-\bar{u}^s)+\bar{u}^s)\right\|^2 \leq 2\eta^2 \mathbb{E}\left\|\sum_{s=t-\tau_i^t}^{t-1}(u^s-\bar{u}^s)\right\|^2 + 2\eta^2 \mathbb{E}\left\|\sum_{s=t-\tau_i^t}^{t-1}\bar{u}^s\right\|^2$$

$$\leq \frac{2\eta^2}{n^2}\underbrace{\mathbb{E}\left\|\sum_{s=t-\tau_i^t}^{t-1}\sum_{i=1}^{n}(\nabla f_i(w^{s-\tau_i^s},\xi_i^{\kappa_i})-\nabla F_i(w^{s-\tau_i^s}))\right\|^2}_{\text{term I}}$$

$$+ \frac{2\eta^2}{n^2}\underbrace{\mathbb{E}\left\|\sum_{s=t-\tau_i^t}^{t-1}\sum_{i=1}^{n}\nabla F_i(w^{s-\tau_i^s})\right\|^2}_{\text{term II}}$$

For term I: Let $\delta_i^s = \nabla f_i(w^{s-\tau_i^s}, \xi_i^\kappa) - \nabla F_i(w^{s-\tau_i^s})$.

$$\text{term I} = \mathbb{E}\left\|\sum_{s=t-\tau_i^t}^{t-1}\sum_{i=1}^n \delta_i^s\right\|^2 = \sum_{s=t-\tau_i^t}^{t-1}\mathbb{E}\left\|\sum_{i=1}^n \delta_i^s\right\|^2 + \sum_{\substack{s_1 \neq s_2 \\ t-\tau_i^t \leq s_1, s_2 \leq t-1}} \mathbb{E}\langle\sum_i \delta_i^{s_1}, \sum_j \delta_j^{s_2}\rangle$$

By Lemma a.6, the sum of these cross-iteration gradient error is zero:

$$\sum_{s_1 \neq s_2} \mathbb{E}\left[\left\langle\sum_{i=1}^n \delta_i^{s_1}, \sum_{j=1}^n \delta_j^{s_2}\right\rangle\right] = \sum_{s_1 \neq s_2}\sum_{i=1}^n\sum_{j=1}^n \mathbb{E}\left[\langle\delta_i^{s_1}, \delta_j^{s_2}\rangle\right] = 0$$

Therefore, by Lemma a.3,

$$\text{term I} = \mathbb{E}\left\|\sum_{s=t-\tau_i^t}^{t-1}\sum_{i=1}^n \delta_i^s\right\|^2 = \sum_{s=t-\tau_i^t}^{t-1}\mathbb{E}\|\sum_{i=1}^n \delta_i^s\|^2 \leq \sum_{s=t-\tau_i^t}^{t-1} n\sum_{i=1}^n \mathbb{E}\|\delta_i^s\|^2$$

$$\leq \tau_i^t n\sum_{i=1}^n \frac{\sigma^2}{n} = \tau_i^t n\sigma^2$$

For term II:

$$\mathbb{E}\|\underbrace{\sum_{s=t-\tau_i^t}^{t-1}}_{\tau_i^t \text{ terms}}\sum_{i=1}^n \nabla F_i(w^{s-\tau_i^s})\|^2 \leq \tau_i^t \sum_{s=t-\tau_i^t}^{t-1}\mathbb{E}\|\sum_{i=1}^n \nabla F_i(w^{s-\tau_i^s})\|^2$$

Therefore, merge term I and term II, we have

$$\mathbb{E}\|w^t - w^{t-\tau_i^t}\|^2 \leq 2\frac{\eta^2}{n^2}(\tau_i^t n\sigma^2) + 2\frac{\eta^2}{n^2}(n^2\tau_i^t \sum_{s=t-\tau_i^t}^{t-1}\mathbb{E}\|\bar{u}^s\|^2)$$

$$\Rightarrow \mathbb{E}\|w^t - w^{t-\tau_i^t}\|^2 \leq 2\eta^2\tau_i^t\left(\frac{\sigma^2}{n} + \sum_{s=t-\tau_i^t}^{t-1}\mathbb{E}\|\bar{u}^s\|^2\right)$$

In ACE, $\bar{u}^s = \nabla F(w_{\text{stale}}^s)$:

$$D_i^t = \mathbb{E}\|w^t - w^{t-\tau_i^t}\|^2 \lesssim \tau_i^t\eta^2\left(\frac{\sigma^2}{n} + \sum_{s'=t-\tau_i^t}^{t-1}\mathbb{E}\|\nabla F(w_{\text{stale}}^{s'})\|_2^2\right).$$

$\square$

## C  CONVERGENCE RATE OF ACE

### C.1  PROOF OF THE RATE

**Theorem 1** (Convergence Rate of ACE (Alg. 1)). *Suppose Assumptions A1-A5 hold. By choosing a step size $\eta \le \frac{1}{8L\tau_{max}}$, ACE achieves the following convergence rate:*

$$\frac{1}{T}\sum_{t=0}^{T-1}\mathbb{E}\|\nabla F(w^t)\|^2 \le \frac{2\Delta}{T\eta} + \frac{4L\tau_{max}\eta\sigma^2}{n} + \frac{2L^2\tau_{max}^2\eta^2\sigma^2}{n}$$

*where $\Delta = F(w^0) - F^*$. Substituting $\eta \simeq \frac{1}{\sqrt{nT}}$, the RHS converges to $0$ as $T \to \infty$.*

*Proof.* Start from the Descent Lemma a.4:

$$\mathbb{E}[F(w^{t+1})] - \mathbb{E}[F(w^t)] \le -\eta\mathbb{E}\langle \nabla F(w^t), u^t\rangle + \frac{L\eta^2}{2}\mathbb{E}\|u^t\|^2 \tag{a.9}$$

We analyze the two terms on the RHS separately to strictly handle the coefficients.

**Term 1: The Inner Product.** Using the property $\mathbb{E}_\xi[u^t] = \bar{u}^t$, we have $\mathbb{E}\langle \nabla F(w^t), u^t\rangle = \langle \nabla F(w^t), \bar{u}^t\rangle$. Using the identity $-\langle a, b\rangle = \frac{1}{2}\|a-b\|^2 - \frac{1}{2}\|a\|^2 - \frac{1}{2}\|b\|^2$:

$$-\eta\langle \nabla F(w^t), \bar{u}^t\rangle = -\frac{\eta}{2}\|\nabla F(w^t)\|^2 - \frac{\eta}{2}\|\bar{u}^t\|^2 + \frac{\eta}{2}\|\nabla F(w^t) - \bar{u}^t\|^2$$

(Note: By taking the expectation *first*, the variance term $\mathbb{E}\|u^t - \bar{u}^t\|^2$ does not appear here, avoiding the negative coefficient issue).

**Term 2: The Smoothness Term.** Using the exact variance decomposition $\mathbb{E}\|u^t\|^2 = \|\bar{u}^t\|^2 + \mathbb{E}\|u^t - \bar{u}^t\|^2$:[2]

$$\frac{L\eta^2}{2}\mathbb{E}\|u^t\|^2 = \frac{L\eta^2}{2}\|\bar{u}^t\|^2 + \frac{L\eta^2}{2}\mathbb{E}\|u^t - \bar{u}^t\|^2$$

**Combine Term 1 and Term 2:** Substituting these back into a.9:

$$\mathbb{E}[F(w^{t+1})] - \mathbb{E}[F(w^t)] \le -\frac{\eta}{2}\|\nabla F(w^t)\|^2 + \left(\frac{L\eta^2}{2} - \frac{\eta}{2}\right)\|\bar{u}^t\|^2$$

$$+ \frac{\eta}{2}\|\nabla F(w^t) - \bar{u}^t\|^2 + \frac{L\eta^2}{2}\mathbb{E}\|u^t - \bar{u}^t\|^2$$

Now, all error terms in the second line of the above inequality have **positive coefficients**, allowing for valid upper bound substitutions:

- **Noise Term:** The coefficient is $\frac{L\eta^2}{2} > 0$. Using Theorem B.3 ($\mathbb{E}\|u^t - \bar{u}^t\|^2 \le \sigma^2/n$):

$$\frac{L\eta^2}{2}\mathbb{E}\|u^t - \bar{u}^t\|^2 \le \frac{L\eta^2\sigma^2}{2n}$$

---

[2]The validity of this decomposition depends on the cross-term $2\mathbb{E}\langle u^t - \bar{u}^t, \bar{u}^t\rangle$ being zero. We prove this using the Law of Iterated Expectations:

$$\mathbb{E}[\langle u^t - \bar{u}^t, \bar{u}^t\rangle] = \mathbb{E}_{\mathcal{H}^t}\left[\langle\mathbb{E}_\xi[u^t - \bar{u}^t \mid \mathcal{H}^t], \bar{u}^t\rangle\right]$$

$$= \mathbb{E}_{\mathcal{H}^t}\left[\langle\underbrace{\mathbb{E}_\xi[u^t \mid \mathcal{H}^t]}_{\bar{u}^t} - \bar{u}^t, \bar{u}^t\rangle\right] = \mathbb{E}_{\mathcal{H}^t}[\langle 0, \bar{u}^t\rangle] = 0.$$

Here, we utilize the fact that $\bar{u}^t$ is measurable with respect to the filtration $\mathcal{H}^t$ (history), allowing it to be pulled out of the inner conditional expectation.

- **Delay Term:** The coefficient is $\frac{\eta}{2} > 0$. For ACE, $\bar{u}^t = \frac{1}{n} \sum \nabla F_i(w^{t-\tau_i^t})$.

$$\|\nabla F(w^t) - \bar{u}^t\|^2 = \left\| \frac{1}{n} \sum_{i=1}^n (\nabla F_i(w^t) - \nabla F_i(w^{t-\tau_i^t})) \right\|^2$$

$$\leq \frac{1}{n} \sum_{i=1}^n \|\nabla F_i(w^t) - \nabla F_i(w^{t-\tau_i^t})\|^2 \leq \frac{L^2}{n} \sum_{i=1}^n \|w^t - w^{t-\tau_i^t}\|^2$$

Substituting these bounds:

$$\mathbb{E}[F(w^{t+1})] - \mathbb{E}[F(w^t)] \leq -\frac{\eta}{2} \mathbb{E}\|\nabla F(w^t)\|^2 + \left( \frac{L\eta^2}{2} - \frac{\eta}{2} \right) \|\bar{u}^t\|^2$$

$$+ \frac{\eta L^2}{2n} \sum_{i=1}^n \mathbb{E}\|w^t - w^{t-\tau_i^t}\|^2 + \frac{L\eta^2 \sigma^2}{2n} \tag{a.10}$$

For $\mathbb{E}\|w^t - w^{t-\tau_i^t}\|^2$, it can be decomposed as a telescoping sum:

$$w^t - w^{t-\tau_i^t} = \sum_{s=t-\tau_i^t}^{t-1} (w^{s+1} - w^s) = \sum_{s=t-\tau_i^t}^{t-1} (-\eta u^s).$$

$$\mathbb{E}\|w^t - w^{t-\tau_i^t}\|^2 = \eta^2 \mathbb{E} \left\| \sum_{s=t-\tau_i^t}^{t-1} u^s \right\|^2$$

Decompose $u^s = (u^s - \bar{u}^s) + \bar{u}^s$ and by Lemma a.3,

$$\eta^2 \mathbb{E} \left\| \sum_{s=t-\tau_i^t}^{t-1} ((u^s - \bar{u}^s) + \bar{u}^s) \right\|^2 \leq 2\eta^2 \mathbb{E} \left\| \sum_{s=t-\tau_i^t}^{t-1} (u^s - \bar{u}^s) \right\|^2 + 2\eta^2 \mathbb{E} \left\| \sum_{s=t-\tau_i^t}^{t-1} \bar{u}^s \right\|^2$$

$$\leq \frac{2\eta^2}{n^2} \underbrace{\mathbb{E} \left\| \sum_{s=t-\tau_i^t}^{t-1} \sum_{i=1}^n (\nabla f_i(w^{s-\tau_i^s}, \xi_i^{\kappa_i}) - \nabla F_i(w^{s-\tau_i^s})) \right\|^2}_{\text{term I}}$$

$$+ \frac{2\eta^2}{n^2} \underbrace{\mathbb{E} \left\| \sum_{s=t-\tau_i^t}^{t-1} \sum_{i=1}^n \nabla F_i(w^{s-\tau_i^s}) \right\|^2}_{\text{term II}}$$

For term I: Let $\delta_i^s = \nabla f_i(w^{s-\tau_i^s}, \xi_i^\kappa) - \nabla F_i(w^{s-\tau_i^s})$.

$$\text{term I} = \mathbb{E} \left\| \sum_{s=t-\tau_i^t}^{t-1} \sum_{i=1}^n \delta_i^s \right\|^2 = \sum_{s=t-\tau_i^t}^{t-1} \mathbb{E} \left\| \sum_{i=1}^n \delta_i^s \right\|^2 + \sum_{\substack{s_1 \neq s_2 \\ t-\tau_i^t \leq s_1, s_2 \leq t-1}} \mathbb{E} \langle \sum_i \delta_i^{s_1}, \sum_j \delta_j^{s_2} \rangle$$

By Lemma a.6, the sum of these cross-iteration gradient error is zero:

$$\sum_{s_1 \neq s_2} \mathbb{E} \left[ \left\langle \sum_{i=1}^n \delta_i^{s_1}, \sum_{j=1}^n \delta_j^{s_2} \right\rangle \right] = \sum_{s_1 \neq s_2} \sum_{i=1}^n \sum_{j=1}^n \mathbb{E} \left[ \langle \delta_i^{s_1}, \delta_j^{s_2} \rangle \right] = 0$$

Therefore, by Lemma a.3,

$$\text{term I} = \mathbb{E}\left\|\sum_{s=t-\tau_i^t}^{t-1}\sum_{i=1}^{n}\delta_i^s\right\|^2 = \sum_{s=t-\tau_i^t}^{t-1}\mathbb{E}\|\sum_{i=1}^{n}\delta_i^s\|^2 \leq \sum_{s=t-\tau_i^t}^{t-1}n\sum_{i=1}^{n}\mathbb{E}\|\delta_i^s\|^2$$

$$\leq \tau_i^t n\sum_{i=1}^{n}\frac{\sigma^2}{n} = \tau_i^t n\sigma^2 \leq \tau_{\max}n\sigma^2$$

For term II:

$$\mathbb{E}\|\underbrace{\sum_{s=t-\tau_i^t}^{t-1}}_{\tau_i^t \text{ terms}}\sum_{i=1}^{n}\nabla F_i(w^{s-\tau_i^s})\|^2 \leq \tau_i^t\sum_{s=t-\tau_i^t}^{t-1}\mathbb{E}\|\sum_{i=1}^{n}\nabla F_i(w^{s-\tau_i^s})\|^2$$

$$\leq \tau_{\max}\sum_{s=t-\tau_{\max}}^{t-1}\mathbb{E}\|\sum_{i=1}^{n}\nabla F_i(w^{s-\tau_i^s})\|^2$$

Therefore, merge term I and term II, we have

$$\mathbb{E}\|w^t - w^{t-\tau_i^t}\|^2 \leq 2\frac{\eta^2}{n^2}(\tau_i^t n\sigma^2) + 2\frac{\eta^2}{n^2}(n^2\tau_i^t\sum_{s=t-\tau_i^t}^{t-1}\mathbb{E}\|\bar{u}^s\|^2)$$

$$\Rightarrow \mathbb{E}\|w^t - w^{t-\tau_i^t}\|^2 \leq 2\eta^2\tau_{\max}\left(\frac{\sigma^2}{n} + \sum_{s=t-\tau_i^t}^{t-1}\mathbb{E}\|\bar{u}^s\|^2\right)$$

Substitute this back into the main inequality a.10 for $\mathbb{E}[F(w^{t+1})] - \mathbb{E}[F(w^t)]$:

$$\mathbb{E}[F(w^{t+1})] - \mathbb{E}[F(w^t)] \leq -\frac{\eta}{2}\mathbb{E}\|\nabla F(w^t)\|^2 + \frac{L\eta^2\sigma^2}{2n} + \left(\frac{L\eta^2}{2} - \frac{\eta}{2}\right)\|\bar{u}^t\|^2$$

$$+ \frac{\eta L^2}{2n}n\left[2\eta^2\tau_{\max}\left(\frac{\sigma^2}{n} + \underbrace{\sum_{s=t-\tau_{\max}}^{t-1}}_{\tau_{\max} \text{ terms}}\mathbb{E}\|\bar{u}^s\|^2\right)\right]$$

$$\leq -\frac{\eta}{2}\mathbb{E}\|\nabla F(w^t)\|^2 + \left(\frac{L\eta^2}{2} + L^2\eta^3\tau_{\max}\right)\frac{\sigma^2}{n}$$

$$+ \left(\frac{L\eta^2}{2} - \frac{\eta}{2} + L^2\eta^3\tau_{\max}^2\right)\max_{t}\mathbb{E}\|\bar{u}^t\|^2$$

$$\leq -\frac{\eta}{2}\mathbb{E}\|\nabla F(w^t)\|^2 + \left(\frac{L\eta^2}{2} + L^2\eta^3\tau_{\max}\right)\frac{\sigma^2}{n} \quad\quad (a.11)$$

The last step holds when $(\frac{L\eta^2}{2} - \frac{\eta}{2} + L^2\eta^3\tau_{\max}^2) \leq 0$. This means $\eta(2L^2\tau_{\max}^2\eta^2 + L\eta - 1) \leq 0$.

Since $\eta > 0$, we need

$$f(\eta) = 2L^2\tau_{\max}^2\eta^2 + L\eta - 1 \leq 0$$

The roots of $f(\eta) = 0$ are $\eta_{+,-} = \frac{-1\pm\sqrt{1+8\tau_{\max}^2}}{4L\tau_{\max}^2}$.

So $0 < \eta \leq \eta_+ = \frac{-1+\sqrt{1+8\tau_{\max}^2}}{4L\tau_{\max}^2}$.

A looser but simpler condition is by decompsing $-\eta/2 = -\frac{\eta}{4} - \frac{\eta}{4}$ in $\frac{L\eta^2}{2} - \frac{\eta}{2} + L^2\eta^3\tau_{\max}^2$ and assign these two $-\frac{\eta}{4}$ separately:

$$\begin{cases} \frac{L\eta^2}{2} - \eta/4 \leq 0 \implies \eta \leq \frac{1}{2L} \\ L^2\eta^3\tau_{\max}^2 - \eta/4 \leq 0 \implies \eta \leq \frac{1}{2L\tau_{\max}} \end{cases}$$

This leads to $\eta \leq \frac{1}{2L\tau_{\max}}$ (assuming $\tau_{\max} \geq 1$).

If we apply a practical learning rate $\eta = c\sqrt{n/T}$, then we require $T \geq 4c^2L^2n\tau_{\max}^2$. This is an **implicit relationship** between $\tau_{\max}$ and $T$. This relationship suggests that in practice, a sufficiently large total number of server iterations $T$ can mitigate the negative impact on convergence caused by a delay $\tau_{\max}$.

Go back to a.11 (simplified equation after dropping $\mathbb{E}\|\bar{u}^t\|^2$ term):

$$\mathbb{E}[F(w^{t+1})] - \mathbb{E}[F(w^t)] \leq -\frac{\eta}{2}\mathbb{E}\|\nabla F(w^t)\|^2 + (\frac{L\eta^2}{2} + L^2\eta^3\tau_{\max})\frac{\sigma^2}{n}$$

Sum over $t = 0$ to $T - 1$:

$$\mathbb{E}[F(w^T)] - F(w^0) \leq -\frac{\eta}{2}\sum_{t=0}^{T-1}\mathbb{E}\|\nabla F(w^t)\|^2 + T(\frac{L\eta^2}{2} + L^2\eta^3\tau_{\max})\frac{\sigma^2}{n}$$

Denote $F(w^0) - \mathbb{E}[F(w^T)]$ as $\Delta$.

$$\frac{\eta}{2}\sum_{t=0}^{T-1}\mathbb{E}\|\nabla F(w^t)\|^2 \leq \Delta + T\eta(\frac{L\eta}{2} + L^2\eta^2\tau_{\max})\frac{\sigma^2}{n}$$

Divide by $T\eta/2$:

$$\frac{1}{T}\sum_{t=0}^{T-1}\mathbb{E}\|\nabla F(w^t)\|^2 \leq \frac{2\Delta}{T\eta} + (L\eta + 2L^2\eta^2\tau_{\max})\frac{\sigma^2}{n}$$

Take $\eta = c\sqrt{n/T}$:

$$\frac{1}{T}\sum_{t=0}^{T-1}\mathbb{E}\|\nabla F(w^t)\|^2 \leq \frac{2\Delta}{Tc\sqrt{n/T}} + \left(Lc\sqrt{n/T} + 2L^2c^2\frac{n}{T}\tau_{\max}\right)\frac{\sigma^2}{n}$$

$$= \frac{2\Delta}{c\sqrt{nT}} + \frac{cL\sigma^2}{\sqrt{nT}} + \frac{2c^2L^2\tau_{\max}\sigma^2}{T}$$

$$\lesssim \frac{\Delta}{\sqrt{nT}} + \frac{L\sigma^2}{\sqrt{nT}} + \frac{L^2\tau_{\max}\sigma^2}{T}$$

$\square$

### C.1.1 Alternative Convergence Analysis with Explicit Independence

In the primary analysis (Appendix C), we utilized the Law of Iterated Expectations to handle the stochasticity of data sampling conditioned on the filtration of the model history. To address potential theoretical concerns regarding the subtle statistical dependency between the current model trajectory $w^t$ and the historical data samples embedded in the aggregated update $u^t$, we provide an alternative proof in this section.

This alternative analysis adopts a stricter "decomposition technique" (Wang et al., 2024a). Instead of evaluating errors relative to the current iterate $w^t$, we anchor the analysis to the **"oldest possible model"** currently influencing the system, denoted as $w^{t-\tau_{\max}}$. By definition, all stochastic gradients involved in the aggregation at iteration $t$ are computed using models generated *after* $w^{t-\tau_{\max}}$ was fixed. This ensures that the specific data batches used for these gradients are statistically independent of the reference point $w^{t-\tau_{\max}}$, thereby eliminating correlation issues without relying on conditional expectations.

It is worth noting that while this technique offers explicit independence, it treats intermediate updates as model drift, leading to an accumulation of error terms scaling with $\tau_{\max}$. Consequently, this results in a **looser upper bound** (with larger constant coefficients) compared to our primary proof. However, it rigorously serves as a robustness check, confirming that the asymptotic convergence rate order of ACE remains valid even under this framework.

*Proof.* Let $s^t := \max(0, t - \tau_{\max})$ be the delayed time index used for decoupling. Note that $t - s^t \leq \tau_{\max}$ for all $t$. By the $L$-smoothness of $F$ and the update rule $w^{t+1} = w^t - \eta u^t$, we have the descent inequality:

$$\mathbb{E}[F(w^{t+1})] - \mathbb{E}[F(w^t)] \leq -\eta \mathbb{E}\langle \nabla F(w^t), u^t \rangle + \frac{L\eta^2}{2} \mathbb{E}\|u^t\|^2 \tag{a.12}$$

**Step 1: Decomposition of the Inner Product.** Since $w^t$ is coupled with the historical gradients in $u^t$, we introduce the delayed iterate $w^{s^t}$ which is independent of the stochastic noise in $u^t$ (conditioned on $\mathcal{F}_{s^t}$). We decompose the inner product as:

$$-\eta \mathbb{E}\langle \nabla F(w^t), u^t \rangle = -\eta \mathbb{E}\langle \nabla F(w^{s^t}), u^t \rangle - \eta \mathbb{E}\langle \nabla F(w^t) - \nabla F(w^{s^t}), u^t \rangle$$

For the first term, we validly apply the conditional expectation $\mathbb{E}[u^t | \mathcal{F}_{s^t}] = \bar{u}^t$. Substituting this back and rearranging terms to recover $\nabla F(w^t)$:

$$-\eta \mathbb{E}\langle \nabla F(w^{s^t}), u^t \rangle = -\eta \mathbb{E}\langle \nabla F(w^{s^t}), \bar{u}^t \rangle$$
$$= -\eta \mathbb{E}\langle \nabla F(w^t), \bar{u}^t \rangle + \eta \mathbb{E}\langle \nabla F(w^t) - \nabla F(w^{s^t}), \bar{u}^t \rangle$$

Combining these, we isolate the coupling error term $\mathcal{E}_{couple}$:

$$-\eta \mathbb{E}\langle \nabla F(w^t), u^t \rangle = -\eta \mathbb{E}\langle \nabla F(w^t), \bar{u}^t \rangle + \underbrace{\eta \mathbb{E}\langle \nabla F(w^t) - \nabla F(w^{s^t}), \bar{u}^t - u^t \rangle}_{\mathcal{E}_{couple}}$$

Let $\delta^t := u^t - \bar{u}^t$. We bound $\mathcal{E}_{couple}$ using Cauchy-Schwarz and the update rule $w^t - w^{s^t} = \sum_{j=1}^{t-s^t}(-\eta u^{t-j})$:

$$\mathcal{E}_{couple} \leq \eta \mathbb{E}[\|\nabla F(w^t) - \nabla F(w^{s^t})\| \|\delta^t\|] \leq \eta L \mathbb{E}\left[\left\|\sum_{j=1}^{t-s^t} \eta u^{t-j}\right\| \|\delta^t\|\right]$$
$$\leq \eta^2 L \sum_{j=1}^{\tau_{\max}} \mathbb{E}[\|u^{t-j}\| \|\delta^t\|] \tag{a.13}$$

Applying Young's inequality $xy \leq \frac{1}{2}x^2 + \frac{1}{2}y^2$ to each term in the sum:

$$\mathcal{E}_{couple} \leq \frac{\eta^2 L}{2} \sum_{j=1}^{\tau_{\max}} \left(\mathbb{E}\|u^{t-j}\|^2 + \mathbb{E}\|\delta^t\|^2\right) = \frac{\eta^2 L}{2} \sum_{j=1}^{\tau_{\max}} \mathbb{E}\|u^{t-j}\|^2 + \frac{\eta^2 L \tau_{\max}}{2} \mathbb{E}\|\delta^t\|^2 \tag{a.14}$$

For the main descent term, we use the identity $-\langle a, b \rangle = \frac{1}{2}\|a-b\|^2 - \frac{1}{2}\|a\|^2 - \frac{1}{2}\|b\|^2$:

$$-\eta \mathbb{E}\langle \nabla F(w^t), \bar{u}^t \rangle = -\frac{\eta}{2}\mathbb{E}\|\nabla F(w^t)\|^2 - \frac{\eta}{2}\mathbb{E}\|\bar{u}^t\|^2 + \frac{\eta}{2}\mathbb{E}\|\nabla F(w^t) - \bar{u}^t\|^2 \tag{a.15}$$

**Step 2: Combining Terms and Variance Bound.** Using Young's inequality for the quadratic term in (a.12), $\mathbb{E}\|u^t\|^2 = \mathbb{E}\|\bar{u}^t + \delta^t\|^2 \leq 2\mathbb{E}\|\bar{u}^t\|^2 + 2\mathbb{E}\|\delta^t\|^2$. Substituting (a.14) and (a.15) into (a.12):

$$\mathbb{E}[F(w^{t+1})] - \mathbb{E}[F(w^t)] \leq -\frac{\eta}{2}\mathbb{E}\|\nabla F(w^t)\|^2 + \left(L\eta^2 - \frac{\eta}{2}\right)\mathbb{E}\|\bar{u}^t\|^2$$
$$+ \frac{\eta}{2} \underbrace{\mathbb{E}\|\nabla F(w^t) - \bar{u}^t\|^2}_{\text{Delay Error}}$$
$$+ \left(L\eta^2 + \frac{\eta^2 L \tau_{\max}}{2}\right)\mathbb{E}\|\delta^t\|^2$$
$$+ \frac{\eta^2 L}{2}\sum_{j=1}^{\tau_{\max}} \mathbb{E}\|u^{t-j}\|^2 \quad \text{(Coupling Drift)} \tag{a.16}$$

**Step 3: Bounding Specific Terms.**

1. **Noise Term:** By Theorem B.3, $\mathbb{E}\|\delta^t\|^2 \leq \frac{\sigma^2}{n}$.

2. **Delay Error:** In ACE, $\bar{u}^t = \frac{1}{n}\sum_{i=1}^n \nabla F_i(w^{t-\tau_i^t})$. Since Term B is strictly zero:

$$\mathbb{E}\|\nabla F(w^t) - \bar{u}^t\|^2 = \mathbb{E}\left\|\frac{1}{n}\sum_{i=1}^n (\nabla F_i(w^t) - \nabla F_i(w^{t-\tau_i^t}))\right\|^2$$

$$\leq \frac{L^2}{n}\sum_{i=1}^n \mathbb{E}\|w^t - w^{t-\tau_i^t}\|^2$$

Using Jensen's inequality on the update sum, $\|w^t - w^{t-k}\|^2 = \|\sum_{j=1}^k \eta u^{t-j}\|^2 \leq k\eta^2 \sum_{j=1}^k \|u^{t-j}\|^2$. Since $\tau_i^t \leq \tau_{\max}$:

$$\mathbb{E}\|\nabla F(w^t) - \bar{u}^t\|^2 \leq L^2\tau_{\max}\eta^2 \sum_{k=1}^{\tau_{\max}} \mathbb{E}\|u^{t-k}\|^2$$

**Step 4: Global Summation and Coefficient Analysis.** Summing (a.16) from $t=0$ to $T-1$ and substituting the bounds:

$$F(w^0) - F^* \geq \frac{\eta}{2}\sum_{t=0}^{T-1} \mathbb{E}\|\nabla F(w^t)\|^2 - \sum_{t=0}^{T-1}\left(\frac{\eta}{2} - L\eta^2\right)\mathbb{E}\|\bar{u}^t\|^2$$

$$- \left(L\eta^2 + \frac{L\tau_{\max}\eta^2}{2}\right)\frac{T\sigma^2}{n}$$

$$- \underbrace{\left(\frac{\eta}{2}L^2\tau_{\max}\eta^2 + \frac{\eta^2 L}{2}\right)}_{\text{Drift Coeff } C_{\text{drift}}} \sum_{t=0}^{T-1}\sum_{k=1}^{\tau_{\max}} \mathbb{E}\|u^{t-k}\|^2$$

We regroup the historical update terms using the property $\sum_{t=0}^{T-1}\sum_{k=1}^{\tau_{\max}} \mathbb{E}\|u^{t-k}\|^2 \leq \tau_{\max}\sum_{t=0}^{T-1} \mathbb{E}\|u^t\|^2$. Expanding $\mathbb{E}\|u^t\|^2 \leq 2\mathbb{E}\|\bar{u}^t\|^2 + \frac{2\sigma^2}{n}$:

$$C_{\text{drift}}\sum_{t=0}^{T-1}\sum_{k=1}^{\tau_{\max}} \mathbb{E}\|u^{t-k}\|^2 \leq C_{\text{drift}}\tau_{\max}\sum_{t=0}^{T-1}\left(2\mathbb{E}\|\bar{u}^t\|^2 + \frac{2\sigma^2}{n}\right)$$

Substituting this back, we analyze the total coefficient $C_{\bar{u}}$ for the $\sum_{t=0}^{T-1} \mathbb{E}\|\bar{u}^t\|^2$ term:

$$C_{\bar{u}} = \left(L\eta^2 - \frac{\eta}{2}\right) + 2\tau_{\max}C_{\text{drift}} = L\eta^2 - \frac{\eta}{2} + L^2\tau_{\max}^2\eta^3 + L\tau_{\max}\eta^2$$

To ensure $C_{\bar{u}} \leq 0$, we factor out $-\eta/2$:

$$C_{\bar{u}} = -\frac{\eta}{2}\left(1 - 2L\eta - 2L\tau_{\max}\eta - 2L^2\tau_{\max}^2\eta^2\right)$$

By choosing $\eta \leq \frac{1}{8L\tau_{\max}}$ (and assuming $\tau_{\max} \geq 1$), we have $2L\eta \leq \frac{1}{4}$, $2L\tau_{\max}\eta \leq \frac{1}{4}$, and $2L^2\tau_{\max}^2\eta^2 \leq 2(\frac{1}{64}) < \frac{1}{4}$. The term in parenthesis is $\geq 1 - 0.25 - 0.25 - 0.04 > 0$, so $C_{\bar{u}} \leq 0$. Thus, we can drop the $\mathbb{E}\|\bar{u}^t\|^2$ terms.

**Step 5: Final Rate.** We collect all remaining noise terms (all proportional to $\sigma^2/n$):

$$\text{Total Noise} = \frac{T\sigma^2}{n}\left[\underbrace{\left(L\eta^2 + \frac{L\tau_{\max}\eta^2}{2}\right)}_{\text{Direct Noise}} + \underbrace{2\tau_{\max}C_{\text{drift}}}_{\text{From Drift}}\right]$$

$$= \frac{T\sigma^2}{n}\left[L\eta^2 + \frac{1}{2}L\tau_{\max}\eta^2 + 2\tau_{\max}\left(\frac{1}{2}L^2\tau_{\max}\eta^3 + \frac{1}{2}L\eta^2\right)\right]$$

$$= \frac{T\sigma^2}{n}\eta^2\left[L + \frac{3}{2}L\tau_{\max} + L^2\tau_{\max}^2\eta\right]$$

$$\leq \frac{T\sigma^2}{n}\eta^2\left[2L\tau_{\max} + L^2\tau_{\max}^2\eta\right] \quad (\text{using } \tau_{\max} \geq 2, \text{terms bounded})$$

Rearranging the main inequality:

$$\frac{\eta}{2} \sum_{t=0}^{T-1} \mathbb{E}\|\nabla F(w^t)\|^2 \leq \Delta + \frac{T\sigma^2}{n}\eta^2(2L\tau_{\max} + L^2\tau_{\max}^2\eta)$$

Multiplying by $\frac{2}{T\eta}$:

$$\frac{1}{T} \sum_{t=0}^{T-1} \mathbb{E}\|\nabla F(w^t)\|^2 \leq \frac{2\Delta}{T\eta} + \frac{2\sigma^2}{n}\eta(2L\tau_{\max} + L^2\tau_{\max}^2\eta)$$

$$= \frac{2\Delta}{T\eta} + \frac{4L\tau_{\max}\eta\sigma^2}{n} + \frac{2L^2\tau_{\max}^2\eta^2\sigma^2}{n}$$

All error terms on the RHS contain the factor $\eta$. By substituting $\eta \propto 1/\sqrt{T}$, the RHS converges to $0$ as $T \to \infty$. $\square$

## D DETAILED DISCUSSION ON THE DELAY-AWARE VARIANT

### D.1 PSEUDO-CODE OF ACED

---

**Algorithm a.1** ACE Variant: ACED (All-Client Engagement Bounded Delay-Aware AFL)

---

**Require:** Maximum allowed delay $\tau_{\text{algo}}$, step size $\eta$.

1: **Server Initialization:** Initialize $w^0$. Server cache stores $(U_i^{\text{cache}}, t_i^{\text{start}})$ for each $i$. Obtain $u_i^0 = \nabla f_i(w^0; \xi_i)$. Set $U_i^{\text{cache}} \leftarrow u_i^0$, $t_i^{\text{start}} \leftarrow 1$ for all $i$. Broadcast $w^1 = w^0 - \eta \frac{1}{n} \sum_{i=1}^n U_i^{\text{cache}}$.

2: **Server Loop:** For $t = 1, \ldots, T - 1$:

3:      Receive $u_{j_t}^{\text{new}} = \nabla f_{j_t}(w^{t_{j_t}^{\text{start}}}; \xi_{j_t}^{\text{new}})$ from client $j_t$.

4:      Update: $U_{j_t}^{\text{cache}} \leftarrow u_{j_t}^{\text{new}}$.

5:      Define active set: $A(t) = \{i \in [n] \mid t - t_i^{\text{start}} \leq \tau_{\text{algo}}\}$.

6:      Compute $n_t = |A(t)|$.

7:      **If** $n_t > 0$: $w^{t+1} = w^t - \eta \frac{1}{n_t} \sum_{i \in A(t)} U_i^{\text{cache}}$.          ▷ Direct sum over active set

8:      **Else**: $w^{t+1} = w^t$.          ▷ Skip update if no valid "fresh" gradients

9:      Send $w^{t+1}$ to client $j_t$ and update: $t_{j_t}^{\text{start}} \leftarrow t+1$.

10: **Client $i$ Operation:**

11:      Initialize: Compute $u_i^1 = \nabla f_i(w^0; \xi_i^1)$, send to server.

12:      Loop: Receive $w^{\text{received}}$, compute $u_i^{\text{new}} = \nabla f_i(w^{\text{received}}; \xi_i^{\text{new}})$, send to server.

---

There are some important details to be noticed for the ACED algorithm:

- **Active Set Formation:** At each server iteration $t$, the server forms an active set $A(t)$ by checking a condition for every client.
  - **If** a client's information is fresh (i.e., the elapsed time since it received its model, $t - t_i^{\text{start}}$, is within the $\tau_{\text{algo}}$ threshold), it is included in the active set for the current update.
  - **Otherwise**, if the client is too slow and its information becomes stale ($t - t_i^{\text{start}} > \tau_{\text{algo}}$), it is temporarily excluded from the aggregation.

- **Rejoin Mechanism:** The algorithm enables clients to rejoin after being excluded.
  - When any client (even one previously excluded for being too slow) sends its completed gradient to the server, the server accepts the update.
  - Crucially, the server then resets that client's timestamp to the current time ($t_i^{\text{start}} \leftarrow t+1$). This action makes the client's information "fresh" again.
  - This reset guarantees the client will be included in the active set in the next iteration, allowing it to rejoin the training process.

### D.2 ASSUMPTIONS FOR ACED

Let $n$ be the total number of clients. The convergence analysis of ACED relies on the following assumptions, adapted from the main ACE paper and the provided analysis sketch .

**Assumption a.1** (*Lower Boundedness*). *The global objective function $F(w) = \frac{1}{n} \sum_{i=1}^n F_i(w)$ is bounded below, i.e., $F(w) \geq F^* > -\infty$ for all $w \in \mathbb{R}^d$. Let $\Delta_F = F(w^0) - F^*$.*

**Assumption a.2** (*L-Smoothness*). *Each local objective function $F_i(w)$ is L-smooth for some $L \geq 0$. This implies $F(w)$ is also L-smooth.*

$$\|\nabla F_i(w) - \nabla F_i(w')\|_2 \leq L \|w - w'\|_2, \quad \forall w, w' \in \mathbb{R}^d.$$

**Assumption a.3** (*Unbiased Stochastic Gradients*). *For any client $i$, its cached gradient $U_i^{cache}$ (used in $u_{BDA}^t$) was computed based on a model $w^{t_i^{start}}$ (where $t_i^{start}$ is the server iteration when client $i$ obtained this model) and a fresh data sample $\xi_i$ drawn at the time of computation. Let $\mathcal{F}_{t_i^{start}}$ be the $\sigma$-algebra of information up to the point $w^{t_i^{start}}$ was determined. Then,*

$$\mathbb{E}[U_i^{cache} \mid \mathcal{F}_{t_i^{start}}] = \nabla F_i(w^{t_i^{start}}).$$

**Assumption a.4** (*Bounded Sampling Noise*). *The variance of the stochastic gradients used to form* $U_i^{cache}$ *is bounded:*

$$\mathbb{E}[\|U_i^{cache} - \nabla F_i(w^{t_i^{start}})\|_2^2 \mid \mathcal{F}_{t_i^{start}}] \leq \sigma^2.$$

**Assumption a.5** (*Bounded Algorithmic Delay for ACED*). *The algorithm-defined maximum delay threshold* $\tau_{algo}$ *is finite and* $\tau_{algo} \geq 1$. *For any client* $i \in A(t)$ *(the active set at server iteration* $t$*), the effective delay of its cached gradient* $U_i^{cache}$ *relative to the current server model* $w^t$ *is* $\delta_i(t) = t - t_i^{start}$, *satisfying* $0 \leq \delta_i(t) \leq \tau_{algo}$.

**Assumption a.6** (*Bounded Data Heterogeneity (BDH)*). *The dissimilarity between local true gradients and the (ideal) global true gradient is bounded:*

$$\|\nabla F_i(w) - \nabla F(w)\|_2^2 \leq \zeta^2$$

*for some constant* $\zeta^2 \geq 0$.

**Assumption a.7** (*Bounded Gradients*). *The expectation of the local gradients are uniformly bounded:* $\|\nabla F_i(w)\|^2 \leq G^2$ *for all* $i, w$. *Note that this assumption is* *NOT necessary*, *but for the simplicity of the notations in the proof.*

**Assumption a.8** (*Minimum Participation for ACED*). *The number of active clients* $n_t = |A(t)|$ *in any update step* $t$ *is lower bounded by* $n_{min} \geq 1$.

### D.3 Convergence Theorem for ACED

**Theorem a.6** (ACED Convergence). *Suppose Assumptions A1-A7 hold. If the step size satisfies* $\eta_t = \eta \leq \frac{1}{12L\tau_{algo}}$, *then for the ACED algorithm, after* $T$ *iterations:*

$$\frac{1}{T}\sum_{t=0}^{T-1}\mathbb{E}\|\nabla F(w^t)\|^2 \leq \frac{2\Delta}{T\eta} + \frac{12(\zeta^2 + G^2)}{T}\sum_{t:n_t<n}\left(1 - \frac{n_t}{n}\right)^2$$

$$+ \frac{6L\tau_{algo}\eta\sigma^2}{n_{min}} + \frac{6L^2\tau_{algo}^2\eta^2\sigma^2}{n_{min}}$$

*where* $n_{min}$ *is the lower bound of active clients. Substituting* $\eta \simeq \frac{1}{\sqrt{T}}$, *the RHS converges to 0 (plus the vanishing bias term) as* $T \to \infty$.

*Proof.* The proof starts with the Descent Lemma. For simplicity, we denote "bounded delay-aware" as BDA.

For an update $w^{t+1} = w^t - \eta_t u_{BDA}^t$, where $u_{BDA}^t = \frac{1}{n_t}\sum_{i \in A(t)} U_i^{cache}$, we have:

$$\mathbb{E}[F(w^{t+1})] \leq \mathbb{E}[F(w^t)] - \eta_t\mathbb{E}[\langle\nabla F(w^t), u_{BDA}^t\rangle] + \frac{L\eta_t^2}{2}\mathbb{E}\|u_{BDA}^t\|^2 \tag{a.17}$$

where $\overline{u}_{BDA}^t = \mathbb{E}_\xi[u_{BDA}^t|\mathcal{F}_t] = \frac{1}{n_t}\sum_{i \in A(t)}\nabla F_i(w^{t_i^{start}})$.

Rearranging a.17:

$$\eta_t\mathbb{E}[\langle\nabla F(w^t), u_{BDA}^t\rangle] \leq \mathbb{E}[F(w^t)] - \mathbb{E}[F(w^{t+1})] + \frac{L\eta_t^2}{2}\mathbb{E}\|u_{BDA}^t\|^2 \tag{a.18}$$

We analyze the two terms on the RHS separately to strictly handle the coefficients.

**Term 1: The Inner Product.** Using the property $\mathbb{E}_\xi[u_{BDA}^t] = \overline{u}_{BDA}^t$, we have $\mathbb{E}[\langle\nabla F(w^t), u_{BDA}^t\rangle] = \langle\nabla F(w^t), \overline{u}_{BDA}^t\rangle$. Using the identity $-\langle a, b\rangle = \frac{1}{2}\|a-b\|^2 - \frac{1}{2}\|a\|^2 - \frac{1}{2}\|b\|^2$:

$$-\eta_t\langle\nabla F(w^t), \overline{u}_{BDA}^t\rangle = -\frac{\eta_t}{2}\|\nabla F(w^t)\|^2 - \frac{\eta_t}{2}\|\overline{u}_{BDA}^t\|^2 + \frac{\eta_t}{2}\|\nabla F(w^t) - \overline{u}_{BDA}^t\|^2$$

**Term 2: The Smoothness Term.** Using the exact variance decomposition $\mathbb{E}\|u_{BDA}^t\|^2 = \|\overline{u}_{BDA}^t\|^2 + \mathbb{E}\|u_{BDA}^t - \overline{u}_{BDA}^t\|^2$:

$$\frac{L\eta_t^2}{2}\mathbb{E}\|u_{BDA}^t\|^2 = \frac{L\eta_t^2}{2}\|\overline{u}_{BDA}^t\|^2 + \frac{L\eta_t^2}{2}\mathbb{E}\|u_{BDA}^t - \overline{u}_{BDA}^t\|^2$$

**Combine Term 1 and Term 2:** Substituting these back into a.17:

$$\mathbb{E}[F(w^{t+1})] - \mathbb{E}[F(w^t)] \leq -\frac{\eta_t}{2}\|\nabla F(w^t)\|^2 + \left(\frac{L\eta_t^2}{2} - \frac{\eta_t}{2}\right)\|\overline{u}_{\text{BDA}}^t\|^2$$

$$+ \frac{\eta_t}{2}\|\nabla F(w^t) - \overline{u}_{\text{BDA}}^t\|^2 + \frac{L\eta_t^2}{2}\mathbb{E}\|u_{\text{BDA}}^t - \overline{u}_{\text{BDA}}^t\|^2 \qquad \text{(a.19)}$$

Now, we bound the key terms:

*A. Sampling Noise of $u_{BDA}^t$:*

Following the similar derivation in the proof of Theorem B.3,

$$\mathbb{E}\|u_{\text{BDA}}^t - \overline{u}_{\text{BDA}}^t\|_2^2 \leq \frac{\sigma^2}{n_t} \leq \frac{\sigma^2}{n_{\min}} \quad \text{(by Assumption a.4, a.8)} \qquad \text{(P1)}$$

*B. Squared Norm of $u_{BDA}^t$:*

Using the variance decomposition and P1:

$$\mathbb{E}\|u_{\text{BDA}}^t\|_2^2 = \mathbb{E}\|u_{\text{BDA}}^t - \overline{u}_{\text{BDA}}^t\|_2^2 + \|\overline{u}_{\text{BDA}}^t\|_2^2$$

$$\leq \frac{\sigma^2}{n_t} + \|\overline{u}_{\text{BDA}}^t\|_2^2 \qquad \text{(P2)}$$

$$\leq \frac{\sigma^2}{n_{\min}} + \|\overline{u}_{\text{BDA}}^t\|_2^2$$

*C. Model Drift $\mathbb{E}\|w^t - w^s\|_2^2$ for $s < t$:*

Let $s = t_i^{\text{start}}$, $\delta = t - s \leq \tau_{\text{algo}}$. The sum of the cross-iteration gradient error is zero::

$$\mathbb{E}\|w^t - w^{t_i^{\text{start}}}\|_2^2 = \mathbb{E}\|\sum_{k=s}^{t-1}(w^{k+1} - w^k)\|_2^2 = \eta_t^2\mathbb{E}\|\sum_{k=s}^{t-1} u_{\text{BDA}}^k\|_2^2$$

$$= \eta_t^2\mathbb{E}\|\sum_{k=s}^{t-1}(u_{\text{BDA}}^k - \overline{u}_{\text{BDA}}^k) + \sum_{k=s}^{t-1}\overline{u}_{\text{BDA}}^k\|_2^2 \quad \text{(Using Lemma a.3)}$$

$$= 2\eta_t^2\underbrace{\mathbb{E}\|\sum_{k=s}^{t-1}(u_{\text{BDA}}^k - \overline{u}_{\text{BDA}}^k)\|_2^2}_{\text{Using Lemma a.6 and P1}} + 2\eta_t^2\underbrace{\mathbb{E}\|\sum_{k=s}^{t-1}\overline{u}_{\text{BDA}}^k\|_2^2}_{\text{Using Lemma a.3}}$$

$$\leq 2\eta_t^2\tau_{\text{algo}}\left(\frac{\sigma^2}{n_t} + \sum_{k=s}^{t-1}\mathbb{E}\|\overline{u}_{\text{BDA}}^k\|_2^2\right) \qquad \text{(P3)}$$

$$\leq 2\eta_t^2\tau_{\text{algo}}\left(\frac{\sigma^2}{n_{\min}} + \sum_{k=s}^{t-1}\mathbb{E}\|\overline{u}_{\text{BDA}}^k\|_2^2\right)$$

*D. Gradient Error* $\mathbb{E}\|\nabla F(w^t) - \overline{u}_{BDA}^t\|^2 := \mathbb{E}\|\mathcal{E}_{BDA}^t\|^2$:

$$\mathcal{E}_{\mathrm{BDA}}^t = \overline{u}^t - \nabla F(w_{\mathrm{stale}}^t)$$

$$= \frac{1}{n_t} \sum_{i \in A(t)} \nabla F_i(w^s) - \frac{1}{n} \sum_{i=1}^n \nabla F_i(w^t)$$

$$= \underbrace{\left(\frac{1}{n_t} - \frac{1}{n}\right) \sum_{i \in A(t)} \nabla F_i(w^t)}_{\text{Part 1}} - \underbrace{\frac{1}{n} \sum_{i \notin A(t)} \nabla F_i(w^t)}_{\text{Part 2}}$$

$$+ \underbrace{\frac{1}{n_t} \sum_{i \in A(t)} \left(\nabla F_i(w^s) - \nabla F_i(w^t)\right)}_{\mathcal{E}_{\text{Delay}}}$$

And by Lemma a.3,
$$\mathbb{E}\|\mathcal{E}_{\mathrm{BDA}}^t\|^2 \le 3\mathbb{E}\|\text{Part 1}\|_2^2 + 3\mathbb{E}\|\text{Part 2}\|_2^2 + 3\mathbb{E}\|\mathcal{E}_{\text{Delay}}\|^2$$

Therefore, similar as the proof for Theorem B.4, for the *partial participation bias* Part 1 and 2, the client subset $\mathcal{S}$ to determine the sum is $A(t)$ or $[n]/A(t)$:

$$\mathbb{E}\|\sum_{i \in \mathcal{S}} \nabla F_i(w^t)\|_2^2 = \mathbb{E}\|\sum_{i \in \mathcal{S}} \nabla F_i(w^t) - \sum_{i \in \mathcal{S}} \nabla F(w^t) + \sum_{i \in \mathcal{S}} \nabla F(w^t)\|_2^2$$

$$\le \underbrace{2\mathbb{E}\|\sum_{i \in \mathcal{S}} \nabla F_i(w^t) - \sum_{i \in \mathcal{S}} \nabla F(w^t)\|_2^2}_{\text{Can be determined by BDH Assumption and Lemma a.3}} + 2\mathbb{E}\|\sum_{i \in \mathcal{S}} \nabla F(w^t)\|_2^2 \quad \text{(By Lemma a.3)}$$

$$\le 2|\mathcal{S}| \sum_{i \in \mathcal{S}} \zeta^2 + 2|\mathcal{S}| \sum_{i \in \mathcal{S}} \mathbb{E}\|\nabla F(w^t)\|_2^2$$

Given that $|A(t)| = n_t, |[n]/A(t)| = n - n_t$:

$$\mathbb{E}\|\text{Part 1}\|_2^2 \le 2\left(\frac{1}{n_t} - \frac{1}{n}\right)^2 \left(n_t^2 \zeta^2 + n_t \sum_{i \in A(t)} \mathbb{E}\|\nabla F(w^t)\|_2^2\right)$$

$$\mathbb{E}\|\text{Part 2}\|_2^2 \le \frac{2}{n^2} \left((n - n_t)^2 \zeta^2 + (n - n_t) \sum_{i \notin A(t)} \mathbb{E}\|\nabla F(w^t)\|_2^2\right)$$

Note that
$$2\left(\frac{1}{n_t} - \frac{1}{n}\right)^2 n_t^2 + \frac{2}{n^2}(n - n_t)^2 = 4\left(1 - \frac{n_t}{n}\right)^2,$$

And we can bound the expectation of the global gradient by Assumption a.7:

$$\mathbb{E}\|\nabla F(w^t)\|_2^2 = \mathbb{E}\|\frac{1}{n} \sum_{i=1}^n \nabla F_i(w^t)\|_2^2$$

$$\le \frac{1}{n^2} \cdot n \sum_{i=1}^n \|F_i(w^t)\|^2 \quad \text{(Lemma a.3)}$$

$$\le \frac{1}{n^2} \cdot n \sum_{i=1}^n G^2 = G, \quad \text{(Assumption a.7)}$$

Therefore,
$$\mathbb{E}\|\text{Part 1}\|_2^2 + \mathbb{E}\|\text{Part 2}\|_2^2 \le 4\left(1 - \frac{n_t}{n}\right)^2 (\zeta^2 + G^2)$$

$$\le 4\left(1 - \frac{n_{\min}}{n}\right)^2 (\zeta^2 + G^2)$$

The delay error $\mathcal{E}_{\text{Delay}}$ (using P3):

$$\mathbb{E}\|\mathcal{E}_{\text{Delay}}\|^2 = \|\frac{1}{n_t}\sum_{i\in A(t)}\left(\nabla F_i(w^s) - \nabla F_i(w^t)\right)\|^2$$

$$\leq \frac{L^2}{n_t}\sum_{i\in A(t)}\mathbb{E}\|w^{t_i^{\text{start}}} - w^t\|^2 \qquad \text{(Lemma a.3)}$$

$$\leq \frac{L^2}{n_t}\cdot n_t\cdot 2\eta_t^2\tau_{\text{algo}}\left(\frac{\sigma^2}{n_t} + \sum_{k=s}^{t-1}\mathbb{E}\|\overline{u}_{\text{BDA}}^k\|_2^2\right) \qquad \text{(Using P3)}$$

$$= 2L^2\eta_t^2\tau_{\text{algo}}\left(\frac{\sigma^2}{n_t} + \sum_{k=s}^{t-1}\mathbb{E}\|\overline{u}_{\text{BDA}}^k\|_2^2\right)$$

$$\leq 2L^2\eta_t^2\tau_{\text{algo}}\left(\frac{\sigma^2}{n_{\min}} + \sum_{k=s}^{t-1}\mathbb{E}\|\overline{u}_{\text{BDA}}^k\|_2^2\right)$$

Thus,

$$\mathbb{E}\|\mathcal{E}_{\text{BDA}}^t\|^2 = \mathbb{E}\|\nabla F(w^t) - \overline{u}_{\text{BDA}}^t\|^2$$

$$\leq 6\mathbb{E}\|\text{Part 1}\|_2^2 + 6\mathbb{E}\|\text{Part 2}\|_2^2 + 6\mathbb{E}\|\mathcal{E}_{\text{Delay}}\|^2$$

$$\leq 24\left(1 - \frac{n_t}{n}\right)^2(\zeta^2 + G^2)$$

$$+ 12L^2\eta_t^2\tau_{\text{algo}}\left(\frac{\sigma^2}{n_t} + \sum_{k=s}^{t-1}\mathbb{E}\|\overline{u}_{\text{BDA}}^k\|_2^2\right) \qquad \text{(P4)}$$

$$\leq 24\left(1 - \frac{n_{\min}}{n}\right)^2(\zeta^2 + G^2)$$

$$+ 12L^2\eta_t^2\tau_{\text{algo}}\left(\frac{\sigma^2}{n_{\min}} + \sum_{k=s}^{t-1}\mathbb{E}\|\overline{u}_{\text{BDA}}^k\|_2^2\right)$$

Substituting P1 and P4 into a.19:

$$\mathbb{E}[F(w^{t+1})] - \mathbb{E}[F(w^t)] \leq -\frac{\eta_t}{2}\mathbb{E}\|\nabla F(w^t)\|^2 + \left(\frac{L\eta_t^2}{2} - \frac{\eta_t}{2}\right)\|\overline{u}_{\text{BDA}}^t\|^2$$

$$+ \frac{\eta_t}{2}\mathbb{E}\|\mathcal{E}_{\text{BDA}}^t\|^2 + \frac{L\eta_t^2}{2}\mathbb{E}\|u_{\text{BDA}}^t - \overline{u}_{\text{BDA}}^t\|^2$$

$$\leq -\frac{\eta_t}{2}\mathbb{E}\|\nabla F(w^t)\|^2 + \frac{L\eta_t^2}{2}\frac{\sigma^2}{n_t}$$

$$+ \frac{\eta_t}{2}\left[24\left(1 - \frac{n_t}{n}\right)^2(\zeta^2 + G^2) + 12L^2\eta_t^2\tau_{\text{algo}}\left(\frac{\sigma^2}{n_t} + \sum_{k=s}^{t-1}\mathbb{E}\|\overline{u}_{\text{BDA}}^k\|_2^2\right)\right]$$

$$+ \left(\frac{L\eta_t^2}{2} - \frac{\eta_t}{2}\right)\mathbb{E}\|\overline{u}_{\text{BDA}}^t\|_2^2$$

$$\leq -\frac{\eta_t}{2}\mathbb{E}\|\nabla F(w^t)\|^2 + \frac{\eta_t}{2}\cdot 24\left(1 - \frac{n_t}{n}\right)^2(\zeta^2 + G^2)$$

$$+ \left(\frac{L\eta_t^2}{2} + 6L^2\eta_t^3\tau_{\text{algo}}\right)\frac{\sigma^2}{n_t}$$

$$+ \left(\frac{L\eta_t^2}{2} - \frac{\eta_t}{2} + 6L^2\eta_t^3\tau_{\text{algo}}^2\right)\max_t\mathbb{E}\|\overline{u}_{\text{BDA}}^t\|_2^2 \qquad \text{(BD)}$$

Let $f(\eta_t) = 6L^2\eta_t^2\tau_{\text{algo}}^2 + L\eta_t/2 - 1/2$. If $f(\eta_t) \leq 0$, then $\left(\frac{L\eta_t^2}{2} - \frac{\eta_t}{2} + 6L^2\eta_t^3\tau_{\text{algo}}^2\right)\max_t\mathbb{E}\|\overline{u}_{\text{BDA}}^t\|^2 \leq 0$. A loose condition to derive $f(\eta_t) \leq 0$ is to decompose $-1/2 = -1/4 - 1/4$ and assign them

separately:

$$
\begin{cases} L\eta_t/2 - 1/4 \le 0 \implies \eta_t \le \frac{1}{2L} \\ 6\eta_t^2 L^2 \tau_{\text{algo}}^2 - 1/4 \le 0 \implies \eta_t \le \frac{1}{2\sqrt{3}L\tau_{\text{algo}}} \end{cases} \implies \eta_t \le \frac{1}{2\sqrt{3}L\tau_{\text{algo}}} \text{ (Using tighter bound)}
$$

With appropriately selected learning rates for each server iteration $t$, BD further becomes:

$$
\frac{\eta_t}{2}\mathbb{E}[\|\nabla F(w^t)\|^2] \le \mathbb{E}[F(w^t)] - \mathbb{E}[F(w^{t+1})] + \frac{\eta_t}{2} \cdot 24\left(1 - \frac{n_t}{n}\right)^2 (\zeta^2 + G^2)
$$
$$
+ \left(\frac{L\eta_t^2}{2} + 6L^2\eta_t^3\tau_{\text{algo}}\right)\frac{\sigma^2}{n_t}
$$

Multiply $2/\eta_t$ on both sides:

$$
\mathbb{E}[\|\nabla F(w^t)\|^2] \le \frac{2}{\eta_t}(\mathbb{E}[F(w^t)] - \mathbb{E}[F(w^{t+1})]) + 24\left(1 - \frac{n_t}{n}\right)^2 (\zeta^2 + G^2)
$$
$$
+ (L\eta_t + 12L^2\eta_t^2\tau_{\text{algo}})\frac{\sigma^2}{n_{\min}}
$$

Let $\{t : n_t < n, t \in [0, T-1]\}$ be the set of iterations with partial client participation. The bias term (in P4) related to $(n - n_t)^2$ is non-zero only for $t \in \{t : n_t < n, t \in [0, T-1]\}$. Summing from $t = 0$ to $T - 1$ and dividing by $T$:

$$
\frac{1}{T}\sum_{t=0}^{T-1}\mathbb{E}[\|\nabla F(w^t)\|^2] \le \frac{1}{T}\sum_{t=0}^{T-1}\frac{2}{\eta_t}(\mathbb{E}[F(w^t)] - \mathbb{E}[F(w^{t+1})])
$$
$$
+ \frac{24(\zeta^2 + G^2)}{T}\sum_{t:n_t<n}(1 - \frac{n_t}{n})^2
$$
$$
+ \frac{1}{T}\sum_{t=0}^{T-1}(L\eta_t + 12L^2\eta_t^2\tau_{\text{algo}})\frac{\sigma^2}{n_t}
$$

Now, we set a fixed learning rate $\eta_t = c\sqrt{n/T}$ for some constant $c > 0$. Let's analyze each term on the RHS:

1. For the first term, given $\Delta = F(w^0) - \mathbb{E}[F(w^T)]$, we have:

$$
\frac{1}{T}\sum_{t=0}^{T-1}\frac{2}{\eta_t}(\mathbb{E}[F(w^t)] - \mathbb{E}[F(w^{t+1})]) = \frac{1}{T}\sum_{t=0}^{T-1}\frac{2}{c\sqrt{n/T}}(\mathbb{E}[F(w^t)] - \mathbb{E}[F(w^{t+1})])
$$
$$
= \frac{2}{c\sqrt{nT}}\sum_{t=0}^{T-1}(\mathbb{E}[F(w^t)] - \mathbb{E}[F(w^{t+1})])
$$
$$
= \frac{2\Delta_F}{c\sqrt{nT}},
$$

2. For the second term, it remains being a sum over only the partial participation iterations:

$$
\frac{24(\zeta^2 + G^2)}{T}\sum_{t:n_t<n}(1 - \frac{n_t}{n})^2
$$

3. For the third term, we substitute $\eta_t = c\sqrt{n/T}$:

$$\frac{1}{T}\sum_{t=0}^{T-1}(L\eta_t + 12L^2\eta_t^2\tau_{\text{algo}})\frac{\sigma^2}{n_t} = \frac{1}{T}\sum_{t=0}^{T-1}\left(Lc\sqrt{\frac{n}{T}} + 12L^2\frac{c^2n}{T}\tau_{\text{algo}}\right)\frac{\sigma^2}{n_t}$$

$$= \frac{1}{T}\sum_{t=0}^{T-1}\left(\frac{12L^2c^2\tau_{\text{algo}}\sigma^2}{T}n + \frac{Lc\sigma^2}{\sqrt{T}}\sqrt{n}\right)\frac{1}{n_t}$$

$$= \frac{12L^2c^2\tau_{\text{algo}}\sigma^2}{T}\cdot n\cdot\underbrace{\frac{1}{T}\sum_{t=0}^{T-1}\frac{1}{n_t}}_{:=\frac{1}{n_{\text{avg}}}} + \frac{Lc\sigma^2}{\sqrt{T}}\cdot\sqrt{n}\cdot\underbrace{\frac{1}{T}\sum_{t=0}^{T-1}\frac{1}{n_t}}_{:=\frac{1}{n_{\text{avg}}}}$$

$$\leq \frac{12L^2c^2\tau_{\text{algo}}\sigma^2}{T}\frac{n}{n_{\text{avg}}} + \frac{Lc\sigma^2}{\sqrt{T}}\frac{\sqrt{n}}{n_{\text{avg}}}$$

Since $\frac{1}{n_{\text{avg}}} = \frac{1}{T}\sum_{t=0}^{T-1}\frac{1}{n_t}$, it is worth noting that $n_{\min} \leq n_{\text{avg}} \leq n$.

Combining these terms, we obtain a bound for the convergence rate:

$$\frac{1}{T}\sum_{t=0}^{T-1}\mathbb{E}[||\nabla F(w^t)||^2] \leq \frac{2\Delta}{c\sqrt{nT}} + \frac{24(\zeta^2 + G^2)}{T}\sum_{t\in T_{\mathcal{P}}}(1 - \frac{n_t}{n})^2 + \frac{12L^2c^2\tau_{\text{algo}}\sigma^2}{T}\frac{n}{n_{\text{avg}}} + \frac{Lc\sigma^2}{\sqrt{T}}\frac{\sqrt{n}}{n_{\text{avg}}}$$

$$\lesssim \frac{\Delta}{\sqrt{nT}} + (\zeta^2 + G^2)\underbrace{\sum_{t:n_t<n}\frac{(n-n_t)^2}{T}}_{\text{Vanishes as } T \text{ increases}}$$

$$+ \frac{L^2\tau_{\text{algo}}\sigma^2}{T}\frac{n}{n_{\text{avg}}} + \frac{L\sigma^2}{\sqrt{T}}\frac{\sqrt{n}}{n_{\text{avg}}}$$

$\square$

### D.3.1 ALTERNATIVE CONVERGENCE RATE ANALYSIS FOR ACED

Similar to the alternative proof provided for the conceptual ACE algorithm, we present a supplementary convergence analysis for ACED that strictly avoids potential correlation issues without relying on the Law of Iterated Expectations.

This analysis anchors the error estimation to the reference model $w^{t-\tau_{\text{algo}}}$. Since the ACED mechanism explicitly enforces that all gradients contributing to the update $u_{\text{BDA}}^t$ are computed on models no older than $\tau_{\text{algo}}$ iterations (i.e., $t - t_i^{\text{start}} \leq \tau_{\text{algo}}$), anchoring to $w^{t-\tau_{\text{algo}}}$ guarantees that the data samples associated with these gradients were generated *after* the reference model was fixed. This secures explicit statistical independence between the reference point and the stochastic noise. While this technique treats the allowable delay as model drift - resulting in a looser upper bound with larger constant coefficients - it rigorously confirms that the convergence properties of ACED are robust and hold independently of the filtration assumptions used in the primary proof.

*Proof.* Let $s^t := \max(0, t - \tau_{\text{algo}})$ be the delayed time index. For any client $i \in A(t)$ utilized in ACED, the delay is bounded by $t - t_i^{\text{start}} \leq \tau_{\text{algo}}$, and the decoupling lag is $t - s^t \leq \tau_{\text{algo}}$.

By the $L$-smoothness of $F$ and the update rule $w^{t+1} = w^t - \eta u_{\text{BDA}}^t$:

$$\mathbb{E}[F(w^{t+1})] - \mathbb{E}[F(w^t)] \leq -\eta\mathbb{E}\langle\nabla F(w^t), u_{\text{BDA}}^t\rangle + \frac{L\eta^2}{2}\mathbb{E}\|u_{\text{BDA}}^t\|^2 \tag{a.20}$$

**Step 1: Rigorous Decomposition of Inner Product.** To handle the statistical dependency between $w^t$ and the historical gradients in $u_{\text{BDA}}^t$, we decompose the inner product using the independent anchor $w^{s^t}$. Since all gradients in $u_{\text{BDA}}^t$ started computation at times $\geq t - \tau_{\text{algo}} \geq s^t$, the stochastic noise in $u_{\text{BDA}}^t$ is independent of $w^{s^t}$ (conditioned on $\mathcal{F}_{s^t}$). We split the inner product into a "Decoupled Term" and a "Coupling Error":

$$-\eta\mathbb{E}\langle\nabla F(w^t), u_{\text{BDA}}^t\rangle = -\eta\mathbb{E}\langle\nabla F(w^{s^t}), u_{\text{BDA}}^t\rangle - \eta\mathbb{E}\langle\nabla F(w^t) - \nabla F(w^{s^t}), u_{\text{BDA}}^t\rangle$$

For the first term, we apply the conditional expectation $\mathbb{E}[u_{\text{BDA}}^t | \mathcal{F}_{s^t}] = \bar{u}_{\text{BDA}}^t$, where $\bar{u}_{\text{BDA}}^t = \frac{1}{n_t} \sum_{i \in A(t)} \nabla F_i(w^{t_i^{\text{start}}})$. Substituting this back:

$$-\eta \mathbb{E}\langle \nabla F(w^{s^t}), \bar{u}_{\text{BDA}}^t \rangle = -\eta \mathbb{E}\langle \nabla F(w^t), \bar{u}_{\text{BDA}}^t \rangle + \eta \mathbb{E}\langle \nabla F(w^t) - \nabla F(w^{s^t}), \bar{u}_{\text{BDA}}^t \rangle$$

Combining these, we isolate the coupling error $\mathcal{E}_{couple}$:

$$-\eta \mathbb{E}\langle \nabla F(w^t), u_{\text{BDA}}^t \rangle = -\eta \mathbb{E}\langle \nabla F(w^t), \bar{u}_{\text{BDA}}^t \rangle + \underbrace{\eta \mathbb{E}\langle \nabla F(w^t) - \nabla F(w^{s^t}), \bar{u}_{\text{BDA}}^t - u_{\text{BDA}}^t \rangle}_{\mathcal{E}_{couple}}$$

Let $\delta^t := u_{\text{BDA}}^t - \bar{u}_{\text{BDA}}^t$ be the zero-mean noise vector. We bound $\mathcal{E}_{couple}$ using the Cauchy-Schwarz inequality, $L$-smoothness, and the update rule $w^t - w^{s^t} = -\sum_{j=1}^{t-s^t} \eta u_{\text{BDA}}^{t-j}$:

$$\mathcal{E}_{couple} \le \eta \mathbb{E}\left[ \|\nabla F(w^t) - \nabla F(w^{s^t})\| \|\delta^t\| \right] \le \eta L \mathbb{E}\left[ \|w^t - w^{s^t}\| \|\delta^t\| \right]$$

$$= \eta L \mathbb{E}\left[ \left\| \sum_{j=1}^{t-s^t} \eta u_{\text{BDA}}^{t-j} \right\| \|\delta^t\| \right] \le \eta^2 L \sum_{j=1}^{\tau_{\text{algo}}} \mathbb{E}[\|u_{\text{BDA}}^{t-j}\| \|\delta^t\|]$$

Using Young's Inequality ($xy \le \frac{1}{2}x^2 + \frac{1}{2}y^2$) on each term in the sum:

$$\mathcal{E}_{couple} \le \frac{\eta^2 L}{2} \sum_{j=1}^{\tau_{\text{algo}}} \left( \mathbb{E}\|u_{\text{BDA}}^{t-j}\|^2 + \mathbb{E}\|\delta^t\|^2 \right) = \frac{\eta^2 L}{2} \sum_{j=1}^{\tau_{\text{algo}}} \mathbb{E}\|u_{\text{BDA}}^{t-j}\|^2 + \frac{\eta^2 L \tau_{\text{algo}}}{2} \mathbb{E}\|\delta^t\|^2 \qquad (a.21)$$

For the main descent term, we use the identity $-\langle a, b \rangle = \frac{1}{2}\|a - b\|^2 - \frac{1}{2}\|a\|^2 - \frac{1}{2}\|b\|^2$:

$$-\eta \mathbb{E}\langle \nabla F(w^t), \bar{u}_{\text{BDA}}^t \rangle = -\frac{\eta}{2} \mathbb{E}\|\nabla F(w^t)\|^2 - \frac{\eta}{2} \mathbb{E}\|\bar{u}_{\text{BDA}}^t\|^2 + \frac{\eta}{2} \mathbb{E}\|\nabla F(w^t) - \bar{u}_{\text{BDA}}^t\|^2 \qquad (a.22)$$

**Step 2: Combining Terms with Quadratic Bound.** For the quadratic term in (a.20), we use Young's Inequality: $\mathbb{E}\|u_{\text{BDA}}^t\|^2 = \mathbb{E}\|\bar{u}_{\text{BDA}}^t + \delta^t\|^2 \le 2\mathbb{E}\|\bar{u}_{\text{BDA}}^t\|^2 + 2\mathbb{E}\|\delta^t\|^2$. Substituting (a.21) and (a.22) into (a.20):

$$\mathbb{E}[F(w^{t+1})] \le \mathbb{E}[F(w^t)] - \frac{\eta}{2} \mathbb{E}\|\nabla F(w^t)\|^2 + \left( L\eta^2 - \frac{\eta}{2} \right) \mathbb{E}\|\bar{u}_{\text{BDA}}^t\|^2$$

$$+ \frac{\eta}{2} \underbrace{\mathbb{E}\|\nabla F(w^t) - \bar{u}_{\text{BDA}}^t\|^2}_{\text{Gradient Error}}$$

$$+ \underbrace{\left( L\eta^2 + \frac{L\tau_{\text{algo}}\eta^2}{2} \right) \mathbb{E}\|\delta^t\|^2}_{\text{Noise Terms}} + \underbrace{\frac{L\eta^2}{2} \sum_{j=1}^{\tau_{\text{algo}}} \mathbb{E}\|u_{\text{BDA}}^{t-j}\|^2}_{\text{Coupling Drift}} \qquad (a.23)$$

**Step 3: Three-Part Decomposition of Gradient Error.** We rigorously decompose the gradient error $\mathcal{E}_{grad} = \bar{u}_{\text{BDA}}^t - \nabla F(w^t)$ into three parts: Participation Bias (Scaling + Missing) and Delay Drift.

$$\mathcal{E}_{grad} = \underbrace{\left( \frac{1}{n_t} - \frac{1}{n} \right) \sum_{i \in A(t)} \nabla F_i(w^t)}_{P1:\text{Scaling}} - \underbrace{\frac{1}{n} \sum_{i \notin A(t)} \nabla F_i(w^t)}_{P2:\text{Missing}} + \underbrace{\frac{1}{n_t} \sum_{i \in A(t)} (\nabla F_i(w^{t_i^{\text{start}}}) - \nabla F_i(w^t))}_{P3:\text{Delay}}$$

Using the inequality $\|a + b + c\|^2 \le 3\|a\|^2 + 3\|b\|^2 + 3\|c\|^2$:

1. **P1 (Scaling Bias):** Using Assumption a.7 (Bounded Gradients $\|\nabla F_i\|^2 \le G^2$) and Jensen's inequality:

$$\mathbb{E}\|P1\|^2 \le \left( \frac{n - n_t}{n n_t} \right)^2 n_t \sum_{i \in A(t)} \mathbb{E}\|\nabla F_i(w^t)\|^2$$

2. **P2 (Missing Data Bias):** Similarly:

$$\mathbb{E}\|P2\|^2 \le \frac{1}{n^2}(n-n_t)\sum_{i\notin A(t)}\mathbb{E}\|\nabla F_i(w^t)\|^2$$

Therefore, similar as the proof for Theorem B.4, for the *partial participation bias* Part 1 and 2, the client subset $\mathcal{S}$ to determine the sum is $A(t)$ or $[n]/A(t)$:

$$\mathbb{E}\|\sum_{i\in\mathcal{S}}\nabla F_i(w^t)\|_2^2 = \mathbb{E}\|\sum_{i\in\mathcal{S}}\nabla F_i(w^t) - \sum_{i\in\mathcal{S}}\nabla F(w^t) + \sum_{i\in\mathcal{S}}\nabla F(w^t)\|_2^2$$

$$\le \quad 2\mathbb{E}\underbrace{\|\sum_{i\in\mathcal{S}}\nabla F_i(w^t) - \sum_{i\in\mathcal{S}}\nabla F(w^t)\|_2^2}_{\text{Can be determined by BDH Assumption and Lemma a.3}} + 2\mathbb{E}\|\sum_{i\in\mathcal{S}}\nabla F(w^t)\|_2^2 \quad \text{(By Lemma a.3)}$$

$$\le 2|\mathcal{S}|\sum_{i\in\mathcal{S}}\zeta^2 + 2|\mathcal{S}|\sum_{i\in\mathcal{S}}\mathbb{E}\|\nabla F(w^t)\|_2^2$$

Given that $|A(t)| = n_t, |[n]/A(t)| = n - n_t$:

$$\mathbb{E}\|\text{P1}\|_2^2 \le 2\left(\frac{1}{n_t} - \frac{1}{n}\right)^2\left(n_t^2\zeta^2 + n_t\sum_{i\in A(t)}\mathbb{E}\|\nabla F(w^t)\|_2^2\right)$$

$$\mathbb{E}\|\text{P2}\|_2^2 \le \frac{2}{n^2}\left((n-n_t)^2\zeta^2 + (n-n_t)\sum_{i\notin A(t)}\mathbb{E}\|\nabla F(w^t)\|_2^2\right)$$

Note that

$$2\left(\frac{1}{n_t} - \frac{1}{n}\right)^2 n_t^2 + \frac{2}{n^2}(n-n_t)^2 = 4\left(1 - \frac{n_t}{n}\right)^2,$$

And we can bound the expectation of the global gradient by Assumption a.7:

$$\mathbb{E}\|\nabla F(w^t)\|_2^2 = \mathbb{E}\|\frac{1}{n}\sum_{i=1}^n\nabla F_i(w^t)\|_2^2$$

$$\le \frac{1}{n^2}\cdot n\sum_{i=1}^n\|F_i(w^t)\|^2 \qquad \text{(Lemma a.3)}$$

$$\le \frac{1}{n^2}\cdot n\sum_{i=1}^n G^2 = G, \qquad \text{(Assumption a.7)}$$

Therefore,

$$\mathbb{E}\|\text{Part 1}\|_2^2 + \mathbb{E}\|\text{Part 2}\|_2^2 \le 4\left(1 - \frac{n_t}{n}\right)^2(\zeta^2 + G^2)$$

$$\le 4\left(1 - \frac{n_{\min}}{n}\right)^2(\zeta^2 + G^2)$$

3. **P3 (Delay Drift):** Using $L$-smoothness and Jensen's inequality:

$$\mathbb{E}\|P3\|^2 \le \frac{1}{n_t}\sum_{i\in A(t)}L^2\mathbb{E}\|w^{t_i^{\text{start}}} - w^t\|^2$$

$$\le \frac{L^2}{n_t}\sum_{i\in A(t)}\tau_{\text{algo}}\eta^2\sum_{j=1}^{\tau_{\text{algo}}}\mathbb{E}\|u_{\text{BDA}}^{t-j}\|^2 = L^2\tau_{\text{algo}}\eta^2\sum_{j=1}^{\tau_{\text{algo}}}\mathbb{E}\|u_{\text{BDA}}^{t-j}\|^2$$

Substituting back into the gradient error term in (a.23):

$$\frac{\eta}{2}\mathbb{E}\|\mathcal{E}_{grad}\|^2 \le 3\|P1\|^2 + 3\|P2\|^2 + 3\|P3\|^2 \tag{a.24}$$

$$\le \frac{\eta}{2}\left[12\left(1-\frac{n_t}{n}\right)^2(\zeta^2+G^2) + 3L^2\tau_{\text{algo}}\eta^2\sum_{j=1}^{\tau_{\text{algo}}}\mathbb{E}\|u_{\text{BDA}}^{t-j}\|^2\right]$$

$$= 6\eta\left(1-\frac{n_t}{n}\right)^2(\zeta^2+G^2) + \frac{3}{2}L^2\tau_{\text{algo}}\eta^3\sum_{j=1}^{\tau_{\text{algo}}}\mathbb{E}\|u_{\text{BDA}}^{t-j}\|^2 \tag{a.25}$$

**Step 4: Global Summation and Coefficient Analysis.** Summing (a.23) from $t = 0$ to $T - 1$ and inserting (a.24):

$$\Delta \ge \frac{\eta}{2}\sum_{t=0}^{T-1}\mathbb{E}\|\nabla F(w^t)\|^2 - \sum_{t=0}^{T-1}\left(L\eta^2 - \frac{\eta}{2}\right)\mathbb{E}\|\bar{u}_{\text{BDA}}^t\|^2$$

$$- \sum_{t=0}^{T-1}6\eta\left(1-\frac{n_t}{n}\right)^2(\zeta^2+G^2)$$

$$- \sum_{t=0}^{T-1}\left(L\eta^2 + \frac{L\tau_{\text{algo}}\eta^2}{2}\right)\mathbb{E}\|\delta^t\|^2$$

$$- \underbrace{\left(\frac{L\eta^2}{2} + \frac{3}{2}L^2\tau_{\text{algo}}\eta^3\right)}_{C_{\text{drift}}}\sum_{t=0}^{T-1}\sum_{j=1}^{\tau_{\text{algo}}}\mathbb{E}\|u_{\text{BDA}}^{t-j}\|^2$$

We regroup the historical update terms. Note that $\sum_{t=0}^{T-1}\sum_{j=1}^{\tau_{\text{algo}}}\mathbb{E}\|u^{t-j}\|^2 \le \tau_{\text{algo}}\sum_{t=0}^{T-1}\mathbb{E}\|u^t\|^2$. Expanding $\mathbb{E}\|u^t\|^2 \le 2\mathbb{E}\|\bar{u}^t\|^2 + 2\mathbb{E}\|\delta^t\|^2$:

$$\sum_{t=0}^{T-1}\sum_{j=1}^{\tau_{\text{algo}}}\mathbb{E}\|u^{t-j}\|^2 \le \tau_{\text{algo}}\sum_{t=0}^{T-1}(2\mathbb{E}\|\bar{u}_{\text{BDA}}^t\|^2 + 2\mathbb{E}\|\delta^t\|^2)$$

We now calculate the total coefficient $C_{\bar{u}}$ for $\sum_{t=0}^{T-1}\mathbb{E}\|\bar{u}_{\text{BDA}}^t\|^2$:

$$C_{\bar{u}} = \left(L\eta^2 - \frac{\eta}{2}\right) + 2\tau_{\text{algo}}C_{\text{drift}}$$

$$= L\eta^2 - \frac{\eta}{2} + 2\tau_{\text{algo}}\left(\frac{L\eta^2}{2} + \frac{3}{2}L^2\tau_{\text{algo}}\eta^3\right)$$

$$= -\frac{\eta}{2}\left(1 - 2L\eta - 2L\tau_{\text{algo}}\eta - 6L^2\tau_{\text{algo}}^2\eta^2\right)$$

We require $C_{\bar{u}} \le 0$. By choosing $\eta \le \frac{1}{12L\tau_{\text{algo}}}$ (and assuming $\tau_{\text{algo}} \ge 1$):

- $2L\eta \le \frac{1}{6} \approx 0.16$

- $2L\tau_{\text{algo}}\eta \le \frac{1}{6} \approx 0.16$

- $6L^2\tau_{\text{algo}}^2\eta^2 \le 6\cdot\frac{1}{144} \approx 0.04$

Sum is $0.36 < 1$. Thus, the term in parenthesis is positive, so $C_{\bar{u}} \le 0$. We can safely drop the $\mathbb{E}\|\bar{u}_{\text{BDA}}^t\|^2$ terms.

**Step 5: Final Rate.** We collect all remaining terms involving $\mathbb{E}\|\delta^t\|^2$. Recall $\mathbb{E}\|\delta^t\|^2 \le \frac{\sigma^2}{n_{min}}$ (upper bound).

$$
\text{Total Noise Coeff } C_\delta = \sum_{t=0}^{T-1} \left[ \left( L\eta^2 + \frac{L\tau_{\text{algo}}\eta^2}{2} \right) + 2\tau_{\text{algo}}C_{\text{drift}} \right]
$$

$$
= \sum_{t=0}^{T-1} \eta^2 \left[ L + \frac{L\tau_{\text{algo}}}{2} + L\tau_{\text{algo}} + 3L^2\tau_{\text{algo}}^2\eta \right]
$$

$$
\le T\eta^2 \left[ 3L\tau_{\text{algo}} + 3L^2\tau_{\text{algo}}^2\eta \right] \quad (\text{using } \tau_{\text{algo}} \ge 1)
$$

Rearranging the main inequality:

$$
\frac{\eta}{2} \sum_{t=0}^{T-1} \mathbb{E}\|\nabla F(w^t)\|^2 \le \Delta + \sum_{t=0}^{T-1} 6\eta \left( 1 - \frac{n_t}{n} \right)^2 (\zeta^2 + G^2) + C_\delta \frac{\sigma^2}{n_{\min}}
$$

Dividing by $T\eta/2$:

$$
\frac{1}{T} \sum_{t=0}^{T-1} \mathbb{E}\|\nabla F(w^t)\|^2 \le \frac{2\Delta}{T\eta} + \frac{12(\zeta^2 + G^2)}{T} \sum_{t:n_t < n} \left( 1 - \frac{n_t}{n} \right)^2
$$

$$
+ \frac{2\sigma^2}{n_{\min}T\eta} \cdot T\eta^2 (3L\tau_{\text{algo}} + 3L^2\tau_{\text{algo}}^2\eta)
$$

$$
\le \frac{2\Delta}{T\eta} + \frac{12(\zeta^2 + G^2)}{T} \sum_{t\in\mathcal{P}} \left( 1 - \frac{n_t}{n} \right)^2 + \frac{6L\tau_{\text{algo}}\eta\sigma^2}{n_{\min}} + \frac{6L^2\tau_{\text{algo}}^2\eta^2\sigma^2}{n_{\min}}
$$

This confirms the rate. $\qquad\qquad\qquad\qquad\qquad\qquad\qquad\qquad\qquad\qquad\qquad\qquad$ $\square$

## D.4 DISCUSSIONS ON ACED

### D.4.1 ALGORITHM BEHAVIOR WITH DROPPED CLIENTS

Let $S_{\text{drop}}$ be the set of $N_{\text{drop}}$ clients that permanently stop sending updates after contributing a final gradient, say $G_j^{\text{last}}$ for client $j \in S_{\text{drop}}$. Let $S_{\text{active}}$ be the set of $N_{\text{active}} = n - N_{\text{drop}}$ clients that continue to participate. For iterations $t$ occurring significantly after the dropouts, the aggregated gradient $u^t$ effectively becomes:

$$
g^t = \frac{1}{n} \left( \sum_{i\in S_{\text{active}}} G_i^{\text{latest},t} + \sum_{j\in S_{\text{drop}}} G_j^{\text{last}} \right)
$$

where $G_i^{\text{latest},t} = \nabla f_i(w^{t-\tau_i^t}; \xi_i^{\kappa_i})$ is the latest (stochastic) gradient from an active client $i$, computed on a (potentially stale) model $w^{t-\tau_i^t}$. The crucial part is that $G_j^{\text{last}}$ for $j \in S_{\text{drop}}$ are fixed, unchanging gradient values based on very old (and increasingly stale) model parameters.

The core problem introduced by permanent dropouts is a persistent bias in the aggregated gradient. Let $\overline{u}^t = \mathbb{E}[u^t | \mathcal{F}_{t'}]$ for some appropriately chosen history $\mathcal{F}_{t'}$ (e.g., $t' = t - \tau_{max}$ for active clients). Taking the expectation over the stochasticity of fresh samples from active clients:

$$
\overline{u}^t \approx \frac{1}{n} \sum_{i\in S_{\text{active}}} \mathbb{E}[G_i^{\text{latest},t} | \mathcal{F}_{t'}] + \frac{1}{n} \sum_{j\in S_{\text{drop}}} G_j^{\text{last}}
$$

Assuming $G_i^{\text{latest},t}$ are unbiased estimates for $\nabla F_i(w^{t-\tau_i^t})$:

$$
\overline{u}^t \approx \frac{1}{n} \sum_{i\in S_{\text{active}}} \nabla F_i(w^{t-\tau_i^t}) + \underbrace{\frac{1}{n} \sum_{j\in S_{\text{drop}}} G_j^{\text{last}}}_{:=B_{\text{drop}}}
$$

The term $B_{\text{drop}}$ represents a constant vector that acts as a persistent bias. This bias does not depend on the current model $w^t$ in the same way active client gradients do.

The bias of the expected update $\overline{u}^t$ relative to the true current gradient $\nabla F(w^t)$ is:

$$\mathcal{E}_t = \overline{u}^t - \nabla F(w^t)$$

$$\mathcal{E}_t = \underbrace{\frac{1}{n}\sum_{i \in S_{\text{active}}} \left(\nabla F_i(w^{t-\tau_i^t}) - \nabla F_i(w^t)\right)}_{\text{Delay error from active clients}} + \underbrace{B_{\text{drop}} - \frac{1}{n}\sum_{j \in S_{\text{drop}}} \nabla F_j(w^t)}_{\text{Non-vanishing bias from dropped clients}}$$

The critical component is $B_{\text{drop}} - \frac{1}{n}\sum_{j \in S_{\text{drop}}} \nabla F_j(w^t)$, which is a non-vanishing bias term. Even if active clients' models $w^{t-\tau_i^t}$ were perfectly up-to-date ($w^t$), and even if $w^t$ were to converge to some $w^*$, the term $B_{\text{drop}} - \frac{1}{n}\sum_{j \in S_{\text{drop}}} \nabla F_j(w^*)$ would remain, unless $B_{\text{drop}}$ coincidentally matches $\frac{1}{n}\sum_{j \in S_{\text{drop}}} \nabla F_j(w^*)$.

### D.4.2    DISCUSSION OF THE ASSUMPTIONS

**Assumption a.6: Managing the Diversity-Staleness Trade-off with the BDH Assumption**    The $\tau_{\text{algo}}$ parameter in ACED provides a direct mechanism to manage the trade-off between client diversity and update staleness, a challenge central to practical AFL. Its primary role is to eliminate the non-vanishing bias ($B_{\text{drop}}$) that arises from permanently dropped or extremely delayed clients, which would otherwise contribute fixed, outdated gradients ($G_j^{\text{last}}$). The convergence analysis quantifies the consequence of this filtering: when the active client set $n_t$ is less than the total $n$, a manageable participation bias emerges, captured by terms related to $(n - n_t)^2 \zeta^2$. The Bounded Data Heterogeneity (BDH) assumption, where $\|\nabla F_i(w) - \nabla F(w)\|_2^2 \leq \zeta^2$, is used in the analysis to bound the participation imbalance bias that occurs when the server update is not formed from all clients ($n_t < n$).

This theoretical insight is validated by experimental results. An excessively small $\tau_{\text{algo}}$ (e.g., $\tau_{\text{algo}} = 1$) leads to a small $n_t$ and significant participation bias, causing ACED's performance to degrade towards that of Vanilla ASGD. Conversely, the experiments show that a moderate $\tau_{\text{algo}}$ (e.g., twice the average client delay) maintains robust performance. This demonstrates that $\tau_{\text{algo}}$ is not a limitation but a tool: it allows the system to be configured to mitigate the more harmful non-vanishing bias from stragglers while controlling the manageable participation bias to maximize performance, thereby ensuring high participation ($n_t \approx n$) in typical scenarios.

**Assumption a.7: The Removability of the Bounded Gradients Assumption**    As explicitly stated, the Bounded Gradients assumption (Assumption a.7) for the ACED convergence analysis can indeed be removed, as it is **not necessary** and serves only **for the simplicity of the notations in the proof**. The assumption's sole purpose is to simplify the bound for the partial participation bias term (when $n_t < n$) during the derivation in Appendix D. Specifically, in the steps leading to **Part 1** and **Part 2** of the bias decomposition, this assumption allows the gradient norm term $\mathbb{E}\|\nabla F(w^t)\|_2^2$ to be bounded by a constant $G^2$, resulting in a concise bias upper bound proportional to $(\zeta^2 + G^2)$. Without this assumption, the bias term would retain its dependency on $(\zeta^2 + \mathbb{E}\|\nabla F(w^t)\|_2^2)$. This modified term would then be carried through to the gradient error bound and subsequently into the main single-step convergence inequality. At that stage, the inequality would contain the term $\mathbb{E}\|\nabla F(w^t)\|_2^2$ on both its left and right sides. By applying a simple algebraic rearrangement to collect all instances of $\mathbb{E}\|\nabla F(w^t)\|_2^2$ onto the left-hand side, one can proceed with the subsequent summation and analysis to derive a valid, although more complex, convergence rate. This confirms that the assumption is a matter of notation convenience rather than a theoretical necessity.

## E RATE COMPARISON WITH OTHER AFL ALGORITHMS

Table a.1: We present the key assumptions of the baseline algorithms, their corresponding convergence rates in the $\mathcal{O}$-sense, and the number of *client-server communication(s) per server iteration*.

| Algorithm | Convergence Rate $\frac{1}{T}\sum_{t=1}^{T}\mathbb{E}[\|\nabla F(w^t)\|^2]$ | Key Assumptions | Comms. per Server Iteration |
|---|---|---|---|
| Vanilla ASGD (Mishchenko et al., 2022) | $\sqrt{\dfrac{\sigma^2}{T}+\dfrac{n}{T}+\zeta^2}$ 
 (with non-vanishing error $\zeta^2$ as $T$ increases) | Bounded Sampling Noise $(\sigma^2)$, Bounded Data Heterogeneity $(\zeta^2)$. | 1 |
| FedBuff (Nguyen et al., 2022) | $\sqrt{\dfrac{\sigma^2+K\zeta^2}{mKT}+\dfrac{K\tau_{\text{avg}}\tau_{\max}\zeta^2+\tau_{\max}\sigma^2}{T}}$ 
 (with heterogeneity amplification $\tau\zeta^2$) | Bounded Sampling Noise $(\sigma^2)$, Bounded Data Heterogeneity $(\zeta^2)$, 

 Bounded Delay $(\tau_{\max}, \tau_{\text{avg}})$. | M |
| Delay-Adaptive ASGD (Koloskova et al., 2022) | $\sqrt{\dfrac{\sigma^2+\zeta^2}{T}}+\dfrac{\sqrt[3]{\tau_{\text{avg}}\frac{1}{n}\sum_{i=1}^{n}\tau_{\text{avg}}^i\zeta_i^2}}{T^{2/3}}$ 
 (with heterogeneity amplification $\tau\zeta^2$) | Bounded Sampling Noise $(\sigma^2)$, Bounded Data Heterogeneity (Global $\zeta^2$, local $\zeta_i^2$), 

 Bounded Delay $(\tau_{\max}, \tau_{\text{avg}})$. | 1 |
| CA²FL (Wang et al., 2024b) | $\dfrac{\Delta+\sigma^2}{\sqrt{TKM}}+\dfrac{\sigma^2+K\zeta^2}{TK}+\dfrac{(\tau_{\max}+\rho_{\max})\sigma^2}{T}$ 
 (No direct $\tau\zeta^2$ term due to calibration) | Bounded Sampling Noise $(\sigma^2)$, Bounded Data Heterogeneity $(\zeta^2)$, 

 Bounded Delay $(\tau_{\max}, \rho_{\max})$. | M |
| ACE (Ours, Theorem 1) | $\dfrac{\Delta}{\sqrt{nT}}+\dfrac{L\sigma^2}{\sqrt{nT}}+\dfrac{L^2\tau_{\max}\sigma^2}{T}$ 
 **(No heterogeneity amplification)** | Bounded Sampling Noise $(\sigma^2)$, Bounded Delay $(\tau_{\max})$. | 1 |
| ACED (Ours) 

 (Theorem D.3) | $\dfrac{\Delta}{\sqrt{nT}}+\dfrac{L\sigma^2}{\frac{n_{\text{avg}}}{\sqrt{n}}\sqrt{T}}+\dfrac{L^2\tau_{\text{algo}}\sigma^2}{T}\dfrac{n}{n_{\text{avg}}}$ 

 $+(\zeta^2+G^2)\displaystyle\sum_{t:n_t<n}\dfrac{(n-n_t)^2}{T}$ | Bounded Sampling Noise $(\sigma^2)$, Minimum Participation $(n_{\min}=\min_t|\{i\in[n]\mid t-t_i^{\text{start}}\le\tau_{\text{algo}}\}|)$, 

 Bounded Gradient $(G)$. | 1 |

Based on the convergence rates and communication costs presented in Table a.1:

- **Shortcomings of Buffered Methods:** Buffered algorithms like FedBuff and CA$^2$FL present two main drawbacks:

  - **High Communication Cost per Update and Slower Convergence:** These methods require the server to collect updates from $M$ clients to fill a buffer before performing a single global model update. This results in a communication cost per server iteration that is $M$ times higher than for non-buffered approaches. A fair metric for comparing convergence is the total number of client communications, $C_{\text{total}}$.
    * For buffered methods like CA$^2$FL, the convergence rate is dominated by the leading term $\mathcal{O}(\frac{1}{\sqrt{MKT}})$, where $T$ is the number of server iterations. Achieving $T$ iterations requires $C_{\text{total}}=M\cdot T$ communications. Substituting $T=C_{\text{total}}/M$, the rate with respect to total communications becomes $\mathcal{O}(\frac{1}{\sqrt{MK(C_{\text{total}}/M)}})=\mathcal{O}(\frac{1}{\sqrt{K\cdot C_{\text{total}}}})$.
    * In contrast, for ACE, each communication triggers a server update, so $T=C_{\text{total}}$. Its convergence in terms of total communications, is $\mathcal{O}(\frac{1}{\sqrt{n\cdot C_{\text{total}}}})$.
    * This shows that for the same communication budget, ACE's theoretical convergence is faster by a factor of $\sqrt{n/K}$. Given that experiments are conducted with $K=1$ for a fair comparison of aggregation strategies, the speedup factor is $\sqrt{n}$.

  - **Reliance on Bounded Data Heterogeneity:** Both algorithms' convergence guarantees depend on the Bounded Data Heterogeneity (BDH) assumption. For FedBuff, this is due to its *partial participation* mechanism $(M<n)$. For CA$^2$FL, it originates from an *imbalanced update scaling* that gives new updates from the buffer a larger weight than older, cached updates. In both cases, this imbalance requires the BDH assumption $(\zeta^2)$ to bound the resulting bias, making their performance theoretically vulnerable in settings with high data heterogeneity. See Appendix F.1 for a more detailed discussion.

- **Limitations of Partial Participation Methods:** Non-buffered, partial participation algorithms (e.g., Vanilla ASGD, Delay-Adaptive ASGD) are communication-efficient (1 communication per iteration) but can suffer from heterogeneity amplification. This is often indicated by terms coupling delay and heterogeneity ($\tau\zeta^2$) in their convergence rates. Furthermore, some of these methods exhibit a fixed error floor; for instance, the rate for Vanilla ASGD includes a non-vanishing $\zeta^2$ term.

- **ACE's Advantage:** ACE is also communication-efficient, requiring only **one communication per iteration**. Its all-client aggregation design eliminates the reliance on the Bounded Data Heterogeneity assumption entirely, thereby mitigating heterogeneity amplification while maintaining maximal communication efficiency.

- **Trade-off in ACED:** The convergence rate of ACED reveals a trade-off between client diversity (participation bias) and update staleness (delay error) in AFL systems. Observing the convergence rate expression for ACED (Theorem D.3, Table a.1):

$$\cdots + \underbrace{\frac{L^2\tau_{\text{algo}}\sigma^2}{T}\frac{n}{n_{\text{avg}}}}_{\text{Delay Error}} + \underbrace{(\zeta^2 + G^2)\sum_{t:n_t<n}\frac{(n-n_t)^2}{T}}_{\text{Participation Bias}}$$

In an AFL system with both high delay (implying some clients may drop out or their local models become very stale) and high heterogeneity (making it difficult to estimate the global gradient from a subset of clients, see the explanation for the BDH assumption in Section 3), a trade-off emerges:

  - Discarding updates from clients with extreme delays (by setting a *smaller* $\tau_{\text{algo}}$) introduces participation bias, quantified by the $(\zeta^2 + G^2)\sum\frac{(n-n_t)^2}{T}$ term.
  - Including these updates (by setting a *larger* $\tau_{\text{algo}}$) introduces significant delay error due to their stale models, which is captured by the $\frac{L^2\tau_{\text{algo}}\sigma^2}{T}\frac{n}{n_{\text{avg}}}$ term.

This dynamic illustrates that these two sources of error cannot be simultaneously eliminated in practical AFL systems. For typical AFL systems, the design of ACED allows clients to rejoin the active set once their delay returns to an acceptable level defined by $\tau_{\text{algo}}$. Provided that extreme delays are reasonably handled, setting a moderate $\tau_{\text{algo}}$ (as shown in Figure 3 in Section 5) to include as many clients as possible is generally more beneficial for improving algorithm performance. This strategy better addresses the common challenge of data heterogeneity (participation bias) in FL and is consistent with the core principle of ACE, which leverages updates from the maximum number of clients to refine the global model.

- **Equivalence of ACED and ACE under a Sufficiently Large Delay Threshold:** The ACED algorithm becomes functionally identical to the conceptual ACE algorithm under a specific condition. This occurs when the delay threshold, $\tau_{\text{algo}}$, is set to a value greater than or equal to the maximum possible system delay, $\tau_{\max}$. In this scenario, the condition for a client's inclusion in the active set, $t - t_i^{\text{start}} \le \tau_{\text{algo}}$, is always satisfied for all clients at every iteration. Consequently, the active set $A(t)$ consistently includes all $n$ clients, making $n_t = n = n_{\text{avg}}$ for all $t$. The update rule for ACED then simplifies to that of ACE. This equivalence extends to their theoretical guarantees. The participation bias term in the convergence rate of ACED, $(\zeta^2 + G^2)\sum_{t:n_t<n}\frac{(n-n_t)^2}{T}$, vanishes as $n_t$ is always equal to $n$. The remaining terms in the ACED rate then simplify to precisely match the convergence rate of ACE in Theorem 1 and Table a.1.

# F  ADDITIONAL EXPERIMENTAL DETAILS

## F.1  DETAILED DISCUSSION ON BASELINE METHODS

This section provides an overview of selected asynchronous federated learning algorithms, detailing their design philosophies and presenting their pseudocode. We focus on FedBuff(Nguyen et al., 2022), CA$^2$FL(Wang et al., 2024b) (Cache-Aided Asynchronous Federated Learning), and Delay-Adaptive Asynchronous SGD(Koloskova et al., 2022) (ASGD). Vanilla ASGD (Mishchenko et al., 2022) can be regarded as a special case of FedBuff when $M = 1$.

### F.1.1  FEDBUFF (FEDERATED LEARNING WITH BUFFERED ASYNCHRONOUS AGGREGATION)

**Design Idea** FedBuff (Nguyen et al., 2022) is designed to improve the efficiency and scalability of federated learning by allowing clients to send their model updates to the server asynchronously. Instead of waiting for all clients in a round to complete their local training (as in synchronous methods like FedAvg), the server in FedBuff accumulates updates from clients as they arrive. The global model is updated only after a certain number of client updates (defined by a buffer size, $M$) have been received. This approach helps to mitigate the straggler problem, where slow clients can delay the entire training process. Upon receiving an update from a client, the server can immediately assign a new task to an available client, thus maintaining a consistent level of client activity (concurrency, $M_c$).

---

**Algorithm a.2** FedBuff (without Differential Privacy)

---

**Require:** Local step size $\eta_l$, global step size $\eta$, server concurrency $M_c$, buffer size $M$, total number of clients $N$.
1: **Initialize:** Global model update accumulator $\Delta_1 \leftarrow 0$, update count $m \leftarrow 0$.
2: Sample an initial set of $M_c$ active clients to run local SGD updates.
3: **repeat**
4:    **if** a client update $\Delta_t^i$ is received from client $i$ **then**
5:       Server accumulates update: $\Delta_t \leftarrow \Delta_t + \Delta_t^i$.
6:       $m \leftarrow m + 1$.
7:       Sample another client $j$ from available clients.
8:       Broadcast the current global model $w_t$ to client $j$.
9:       Client $j$ runs local SGD updates.
10:    **end if**
11:    **if** $m = M$ **then**
12:       Update global model: $w_{t+1} \leftarrow w_t + \eta \cdot (\Delta_t/M)$.
13:       Reset for next aggregation: $m \leftarrow 0, \Delta_{t+1} \leftarrow 0, t \leftarrow t + 1$.
14:    **end if**
15: **until** Convergence

---

**Partial Participation ($M < N$):** This is the standard operational mode for FedBuff (Nguyen et al., 2022). The server waits to fill a buffer of size $M$ before updating the global model. As our theoretical analysis in Section 4 shows, this design inherently introduces **partial participation bias**, which is the root cause of heterogeneity amplification when client data is non-IID. Vanilla ASGD (Mishchenko et al., 2022) represents the extreme case where $M = 1$, maximizing this bias and the variance of the global updates.

**Full Participation ($M = N$):** In this hypothetical scenario, FedBuff would be forced to wait for updates from all $N$ clients before performing a single update. This transforms the algorithm into a **synchronous** protocol, similar to FedAvg (Li et al., 2020), thereby losing the primary advantage of AFL in overcoming straggler issues.

**Update Frequency and Communication Cost:** A critical consequence of FedBuff's buffered design is the decoupling of client communication from global model updates. To perform a single server iteration (one global update), the server must wait for and process $M$ individual client communications. This introduces a synchronization-like bottleneck, reducing the overall frequency

of model evolution. This means that the communication cost per learning step is $M$ times higher than for a non-buffered approach, a crucial factor in evaluating overall system efficiency.

### F.1.2 CA²FL (CACHE-AIDED ASYNCHRONOUS FEDERATED LEARNING)

**Design Idea** CA$^2$FL (Wang et al., 2024b) uses a buffering mechanism similar to FedBuff but adds a calibration step using a server-side cache of historical updates from all clients. Its behavior changes drastically depending on the buffer size $M$. The core idea is for the server to maintain a cache of the latest model update (or difference) received from each client. These cached updates are then used to calibrate the global model update. When a client sends its new update $\Delta_t^i$, the server calculates the difference between this new update and the client's previously cached update $h_t^i$. This calibrated difference, $\Delta_t^i - h_t^i$, is then accumulated. The global update $v_t$ incorporates the average of these calibrated differences along with a global cached variable $h_t$ (which is the average of all clients' currently cached updates). This mechanism aims to make the aggregated update more consistent with the current global model state, especially when dealing with stale updates from delayed clients and diverse data distributions across clients. CA$^2$FL is designed to achieve these improvements without imposing additional communication or computation overhead on the clients.

---

**Algorithm a.3** CA$^2$FL (Cache-Aided Asynchronous FL)

---

**Require:** Local step size $\eta_l$, global step size $\eta$, server concurrency $M_c$, buffer size $M$, total number of clients $N$.

1: **Initialize:** Global model update accumulator $\Delta_1 \leftarrow 0$, Cached update for each client $i \in [N]$, $h_1^i \leftarrow 0$, Global cached variable $h_1 \leftarrow \frac{1}{N}\sum_{i=1}^N h_1^i$, Update count $m \leftarrow 0$, set of clients updated in current buffer $\mathcal{S}_t \leftarrow \emptyset$.
2: Sample an initial set of $M_c$ active clients to run local SGD updates.
3: **repeat**
4:     **if** a client update $\Delta_t^i$ is received from client $i$ **then**
5:         Server accumulates calibrated update: $\Delta_t \leftarrow \Delta_t + (\Delta_t^i - h_t^i)$.
6:         Server updates client's cached variable: $h_{t+1}^i \leftarrow \Delta_t^i$.
7:         $m \leftarrow m + 1$.
8:         $\mathcal{S}_t \leftarrow \mathcal{S}_t \cup \{i\}$.
9:         Sample another client $j$ from available clients.
10:        Broadcast the current global model $w_t$ to client $j$.
11:        Client $j$ runs local SGD updates.
12:     **end if**
13:     **if** $m = M$ **then**
14:         **for all** clients $j \notin \mathcal{S}_t$ **do**
15:            Server maintains their cached variable: $h_{t+1}^j \leftarrow h_t^j$.
16:         **end for**
17:         Calculate calibrated global update: $v_t \leftarrow h_t + \frac{1}{|\mathcal{S}_t|}\Delta_t$.
18:         Update global model: $w_{t+1} \leftarrow w_t + \eta \cdot v_t$.
19:         Initialize global cached variable for next round: $h_{t+1} \leftarrow \frac{1}{N}\sum_{i=1}^N h_{t+1}^i$.
20:         Reset for next aggregation: $m \leftarrow 0, \Delta_{t+1} \leftarrow 0, \mathcal{S}_{t+1} \leftarrow \emptyset, t \leftarrow t + 1$.
21:     **end if**
22: **until** Convergence

---

**The $M = N$ Limit: A Synchronous Algorithm** A critical distinction is that setting the buffer size $M = N$ in CA$^2$FL does not make it equivalent to ACE; it makes it **synchronous**. The server's workflow requires waiting until all $N$ client updates are received to perform a **single** global update. During this waiting period, the global model $w^t$ remains static. Consequently, all $N$ clients compute their updates based on the exact same model version and receive the same new model $w^{t+1}$ for the next round. In this synchronous workflow, information staleness, becomes trivially zero for all clients ($\tau_i^t = 0$), which is fundamentally different from any asynchronous protocol.

**The $M = 1$ Limit: Imbalanced Update Weighting** Even in the $M = 1$ case, where both CA$^2$FL and ACE update upon every client's arrival, their mathematical update rules are fundamentally

different. Let $h_t = \frac{1}{N} \sum_{k=1}^{N} h_k^{\text{old}}$ be the average of all cached updates before a new update $\Delta_j^{\text{new}}$ arrives from client $j$.

- The **CA²FL** update rule becomes:

$$v_t = h_t + (\Delta_j^{\text{new}} - h_j^{\text{old}}) \tag{a.26}$$

The global update applies the **full, unscaled** change from the reporting client to the global average. This retains a form of partial participation bias, and its convergence rate consequently depends on the data heterogeneity bound $\zeta^2$, as shown in Table a.1.

- The **ACE** incremental update rule (derived in Section 3.4) is:

$$u^t = u^{t-1} + \frac{1}{N}(\Delta_j^{\text{new}} - h_j^{\text{old}}) \tag{a.27}$$

Here, the change from the reporting client is **scaled by** $1/N$. This scaling is crucial as it ensures all clients, whether their information is new or old, **contribute equally** to the final average. This design choice is what eliminates the dependency on the BDH assumption and removes the $\zeta^2$ term from ACE's convergence bound.

**Update Frequency and Communication Cost:** Despite its advanced calibration mechanism, CA²FL's reliance on a buffer of size M means it shares the same fundamental limitation as FedBuff regarding update frequency. A single global model update requires the server to wait for $M$ clients. This design choice inherently trades higher model evolution frequency for its calibration benefits, resulting in a communication cost of $M$ client uploads for every server iteration.

### F.1.3 DELAY-ADAPTIVE ASYNCHRONOUS SGD (ASGD)

**Design Idea** Standard Asynchronous SGD (ASGD) allows workers to compute and send gradients at their own pace without synchronization. This can lead to the server applying "stale" gradients, which are gradients computed based on older versions of the global model. The convergence rates of such algorithms often depend on the maximum gradient delay ($\tau_{max}$), a metric that can be overly pessimistic if significant delays (stragglers) are rare. Delay-Adaptive ASGD (Koloskova et al., 2022) directly targets the adverse effect of staleness by dynamically adjusting the learning rate $\eta_t$ based on the delay $\tau_t$ of each incoming gradient. The core idea is that gradients computed on older models (i.e., with a large $\tau_t$) are less reliable and should have a smaller impact on the global model update.

---

**Algorithm a.4** Delay-Adaptive Asynchronous SGD

---

**Require:** Initial model $w^{(0)}$, base learning rate parameter $\eta \leq 1/(4L)$ (where $L$ is the smoothness constant of the objective function), total iterations $T$.
 1: **Initialize:** Server selects an initial set of active workers $\mathcal{C}_0$ and sends them $w^{(0)}$.
 2: **for** $t = 0, \dots, T-1$ **do**
 3:     Active workers $\mathcal{C}_t$ compute stochastic gradients $g = \nabla F(w_{\text{model}}, \xi)$ in parallel, based on the model version $w_{\text{model}}$ they were assigned.
 4:     Once a worker $j_t$ finishes computation (gradient $g_t = \nabla F(w^{(t-\tau_t)}, \xi_t)$ for model $w^{(t-\tau_t)}$ with delay $\tau_t$), it sends $g_t$ to the server.
 5:     Server determines delay-adaptive step size $\eta_t$:
 6:     **if** $\tau_t \leq \tau_C$ **then**                           $\triangleright \tau_C$ is concurrency or average concurrency
 7:         $\eta_t \leftarrow \eta$.
 8:     **else**
 9:         Choose $\eta_t$ such that $0 \leq \eta_t < \min\{\eta, 1/(4L\tau_t)\}$.    $\triangleright$ e.g., drop ($\eta_t = 0$) or scale down
10:     **end if**
11:     Server updates global model: $w_{t+1} \leftarrow w_t - \eta_t \cdot g_t$.
12:     Server selects a subset $\mathcal{A}_t$ of inactive workers (can include $j_t$) and sends them the latest model $w^{t+1}$.
13:     Update active worker set: $\mathcal{C}_{t+1} \leftarrow (\mathcal{C}_t \setminus \{j_t\}) \cup \mathcal{A}_t$.
14: **end for**

---

**Connection to Delay Error:** Our theoretical framework in Section 4 identifies that the **Delay Error** (Term C) is amplified by the model drift experienced by a client. This drift is influenced by a factor proportional to $\eta^2 \tau_i^t$. The inequality below, derived from our analysis of the per-iteration delay error, highlights this dependency on different algorithm design choices:

$$
\mathbb{E}||\text{Delay Error}||^2 \lesssim \underbrace{\eta^2 \tau_i^t}_{\text{Learning rate}} \left\{ \underbrace{\frac{\sigma^2}{m}}_{\text{Noise}} + \underbrace{\sum_{s=t-\tau_i^t}^{t-1}}_{\text{Number of server iterations}} \underbrace{((N-m)^2 K^2 \zeta^2 + \cdots)}_{\text{Local steps}} \right\}
$$

A large delay $\tau_i^t$ can cause this error term to dominate, especially when combined with the bias from partial participation. Delay-Adaptive ASGD mitigates this by ensuring the product $\eta_t^2 \tau_t$ does not grow uncontrollably. By setting $\eta_t$ to be inversely proportional to $\tau_t$ for large delays (e.g., $\eta_t \propto 1/\tau_t$), the algorithm effectively down-weights the contribution of highly stale gradients, thus suppressing their negative impact and reducing the magnitude of the overall Delay Error.

**Limitations:** While this adaptive learning rate strategy effectively reduces the error component related to *staleness*, it does not address the **Bias Error** (Term B) that arises from its single-client update mechanism ($m = 1$). The global model is still updated based on the perspective of a single, potentially unrepresentative client at each step. Therefore, it only partially mitigates the heterogeneity amplification effect, whereas ACE is designed to eliminate the partial participation bias at its source.

F.1.4 ACE AND ACED: ASYNCHRONOUS FULL AND DYNAMIC PARTICIPATION

---

**Algorithm a.5** ACE Implementation (Incremental Update), in addition to Algorithm 1

---

1: **System Initialization:**
2:     Server initializes global model $w^0$.
3:     For each client $i \in [n]$:
4:         Client computes initial gradient $g_i^0 \leftarrow \nabla f_i(w^0; \xi_i^0)$ and sends it to the server.
5:         Client stores its gradient locally: $g_i^{\text{prev}} \leftarrow g_i^0$.
6:     Server computes initial aggregate update: $u \leftarrow \frac{1}{n} \sum_{i=1}^n g_i^0$.         $\triangleright \mathcal{O}(d)$ server storage cost
7:     Server updates model: $w^1 \leftarrow w^0 - \eta u$.
8:     Server makes $w^1$ available to clients.
9: **Server Loop:** For $t = 1, \ldots, T-1$:
10:     Wait to receive a gradient difference $(g_i^{\text{new}} - g_i^{\text{prev}})$ from some client $j$.
11:     Incrementally update the aggregate: $u \leftarrow u + (g_j^{\text{new}} - g_j^{\text{prev}})/n$.
12:     Update global model: $w^{t+1} \leftarrow w^t - \eta u$.
13:     Server makes $w^{t+1}$ available to client $j$.
14: **Client $i$ Operation (after initialization):**
15:     $w_{\text{local}} \leftarrow$ latest model version received from server.
16:     Compute new gradient $g_i^{\text{new}} \leftarrow \nabla f_i(w_{\text{local}}; \xi_i^{\text{new}})$.
17:     Send gradient difference $(g_i^{\text{new}} - g_i^{\text{prev}})$ to server.
18:     Update local state for next round: $g_i^{\text{prev}} \leftarrow g_i^{\text{new}}$.         $\triangleright \mathcal{O}(d)$ client storage cost

---

**ACE:** By design, ACE is an asynchronous algorithm that always leverages information from $m = n$ clients. However, unlike the synchronous $M = N$ case of CA$^2$FL, it performs a global update **immediately** upon the arrival of *any single* client's gradient. **It averages this freshly arrived gradient with the stale gradients from other clients.** This results in a high frequency of model updates, where the global model is constantly evolving. This dynamic is the essence of its asynchronous nature and is precisely what gives rise to the non-trivial staleness values ($\tau_i^t > 0$) that our framework analyzes.

**ACED:** This variant introduces a dynamic participation model where $m$ becomes a variable, $n_t$, determined by system dynamics and the hyperparameter $\tau_{\text{algo}}$. It explicitly navigates the trade-off discussed in this paper: when $n_t < n$, it accepts a controllable level of partial participation bias in exchange for robustness against the extreme staleness introduced by stragglers or dropped-out clients.

**Update Frequency and Communication Efficiency:** A core design principle of ACE is its non-buffered, immediate update mechanism. This establishes a **1-to-1 relationship** between a client's arrival and a global model update (one server iteration). Consequently, for a given budget of total client communications (e.g., 1000 uploads), ACE performs 1000 global updates, whereas a buffered method with $M = 10$ would only perform 100. This makes ACE more communication-efficient, allowing for faster model evolution under the same communication constraints.

## F.2 EXTENDED CONVERGENCE ANALYSIS AND STABILITY VISUALIZATION FOR SECTION 5

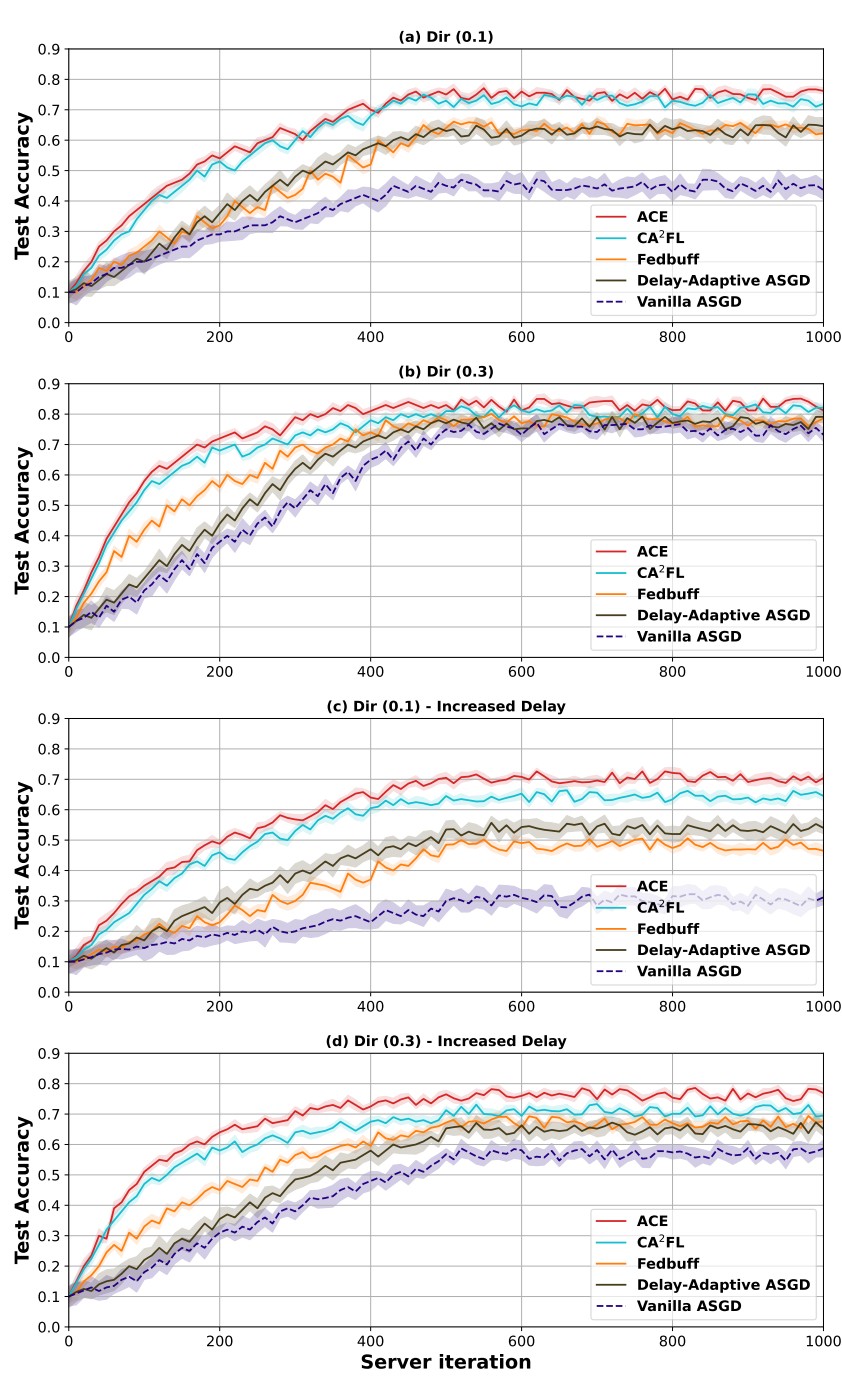

Figure a.1: Extended performance comparison of AFL algorithms on CIFAR-10 up to 1000 server iterations, including stability analysis via error bars. The four subplots correspond to the scenarios detailed in Section 5: (a) Dir (0.1), (b) Dir (0.3), (c) Dir (0.1) with increased delay, and (d) Dir (0.3) with increased delay. Shaded regions represent the standard deviation ($\pm\sigma$) of accuracy. The error bands clearly show that single-client update methods (Vanilla ASGD, Delay-Adaptive ASGD) exhibit higher variance, while multi-client aggregation methods (FedBuff, CA$^2$FL, and ACE) converge more stably.

To provide a more comprehensive view of the algorithms' long-term behavior, we extend the primary experiments in Section 5 to 1000 server iterations, with the results presented in Figure a.1. This extended analysis serves two main purposes. First, it demonstrates that the primary convergence dynamics and final performance rankings of all algorithms are well-established within the first 450-500 iterations. The subsequent iterations show that the learning curves have reached their plateaus, validating our choice of $T = 500$ in the main paper as a sufficient duration for a conclusive comparison.

Second, the larger format of this appendix figure allows for the inclusion of error bars (visualized as shaded regions representing one standard deviation, $\pm\sigma$), which were omitted from the smaller figures in the main text due to space constraints that would compromise visual clarity. The insights from these error bars provide strong empirical support for our theoretical framework:

- **Stability Correlates with Participation:** A clear trend emerges from the visualization: an algorithm's stability is directly correlated with the number of clients participating in each global update.

- **High Variance in Single-Client Methods:** The single-client update methods, Vanilla ASGD and Delay-Adaptive ASGD, consistently exhibit the widest and most volatile error bands. This empirically demonstrates their high update variance, as each step is guided by a single client's potentially noisy and biased gradient, leading to a more erratic convergence path.

- **Variance Reduction via Aggregation:** In contrast, methods that aggregate updates from multiple clients (FedBuff, CA$^2$FL, and our proposed ACE) show narrower and more stable error bands. This confirms that aggregating information across a diverse client set effectively reduces the variance of the global updates, resulting in a more reliable and predictable training process. Notably, ACE, which leverages information from all clients at every step, maintains one of the most stable profiles throughout the training, reinforcing the benefits of its all-client engagement design.

In summary, this extended analysis provides a clear visual confirmation that increased client participation is crucial not only for final accuracy but also for achieving a more stable training process.

### F.3 ADDITIONAL EXPERIMENTS

To further validate the robustness and effectiveness of our proposed ACE algorithm, we conduct additional experiments across a variety of datasets and task types. These experiments are designed to assess ACE's performance under different data distributions, model architectures, and against specific challenges inherent in federated learning.

#### F.3.1 RESULTS ON CIFAR-100 DATASET

We simulate an Asynchronous Federated Learning (AFL) environment to evaluate the performance of various algorithms on the CIFAR-100 (Krizhevsky, 2009) image classification dataset with ResNet-18(He et al., 2016) models. We deploy $n = 100$ clients, each holding a non-identically distributed (non-IID) subset of the data. The non-IID nature is modeled using a *Dirichlet distribution*, where the concentration parameter $\alpha$ controls the degree of data heterogeneity across clients. Lower $\alpha$ values indicate higher heterogeneity (clients' data distributions are more dissimilar), while higher $\alpha$ values represent more IID-like data distributions. The $\alpha$ values explored are $\alpha \in \{0.1, 0.3, 1.0, 10.0\}$.

The delays in AFL are simulated using an exponential distribution with a mean parameter $\beta$. Higher $\beta$ values signify longer average delays and a greater likelihood of extreme delays (stragglers) in the system. The $\beta$ values investigated are $\beta \in \{1, 5, 20, 30\}$.

All algorithms are trained for $T = 500$ server iterations and results are reported as an average of 5 runs. The primary evaluation metric is the *test accuracy* achieved by the global model on the CIFAR-100 test set. The test set remains identical across different levels of heterogeneity across clients and the extent of delays. The experiments aim to understand how different AFL algorithms perform under varying data heterogeneity and delay profile, particularly focusing on the phenomenon of "heterogeneity amplification" where faster clients with specific data distributions can disproportionately influence the global model in asynchronous settings. The baseline algorithms compared include FedBuff (Nguyen et al., 2022), CA$^2$FL (Wang et al., 2024b), Delay-adaptive ASGD (Koloskova et al., 2022), Vanilla

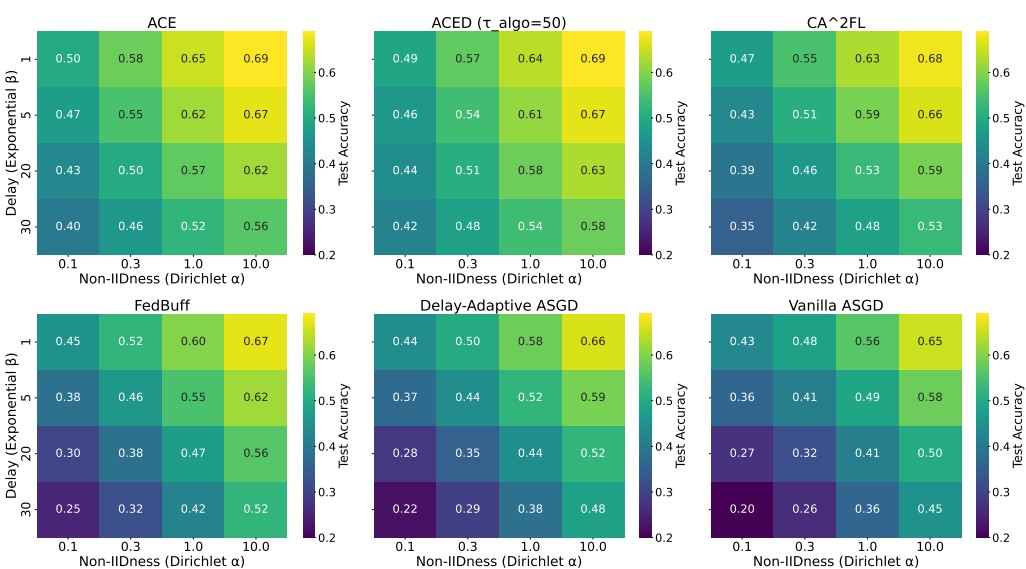

Figure a.2: **Comparative Performance of Asynchronous Federated Learning Algorithms on CIFAR-100 under Varying Data Heterogeneity and System Delays.** The heatmaps illustrate the final test accuracy of six AFL algorithms: (a) ACE, (b) ACED ($\tau_{\text{algo}} = 50$), (c) CA$^2$FL, (d) FedBuff, (e) Delay-Adaptive ASGD, and (f) Vanilla ASGD. The x-axis represents the Dirichlet distribution parameter $\alpha$ controlling client data non-IIDness (lower $\alpha$ indicates higher heterogeneity). The y-axis represents the mean $\beta$ of an exponential distribution modeling client delays (higher $\beta$ indicates greater system delay and straggler presence). Accuracy values are normalized across all heatmaps using a common color scale to facilitate direct comparison. Algorithms like ACE and ACED demonstrate strong performance and robustness, particularly maintaining higher accuracies under combined high heterogeneity and high delay conditions. In contrast, algorithms such as FedBuff, Delay-Adaptive ASGD, and Vanilla ASGD show a more pronounced degradation, illustrating the impact of heterogeneity amplification. ACED's performance at high delay (e.g., $\beta = 30$) relative to ACE highlights its design for mitigating the impact of extreme stragglers.

ASGD (Mishchenko et al., 2022), alongside the proposed ACE and its practical variant ACED (with $\tau_{\text{algo}} = 50$). The goal is to observe how design choices such as full client gradient aggregation (ACE) or bounded-delay aggregation (ACED) impact robustness and final performance under these challenging AFL conditions.

### F.3.2 RESULTS ON 20NEWSGROUP TEXT CLASSIFICATION FOR BERT MODELS

**20Newsgroup Dataset** The 20Newsgroup dataset is a widely used collection of approximately 20k newsgroup documents, partitioned (nearly) evenly across 20 different newsgroups (Lang, 1995). Some examples of these newsgroups include topics like computers (e.g., `comp.graphics`), science (e.g., `sci.med`, `sci.space`), politics (e.g., `talk.politics.misc`), and religion (e.g., `soc.religion.christian`). The paper uses this dataset for **text classification tasks** because its larger output space (20 labels) is important for studying label-distribution shift scenarios. (Lang, 1995) specifies the training and test set sizes for 20Newsgroup as 11.3k training examples and 7.5k test examples. To simulate non-IID data, particularly label distribution shift, we partition the dataset among clients using a Dirichlet distribution $Dir(\alpha)$. We distribute the datasets across $n = 100$ clients. For the experiments presented in Table a.2, client delays are simulated using an exponential distribution with a mean parameter $\beta = 5$.

**Models: DistilBERT and BERT** Our experiments primarily utilize Transformer-based architectures.

- **BERT** (Bidirectional Encoder Representations from Transformers) is a language representation model pre-trained on a large corpus of text, which can be fine-tuned for a wide range

of NLP tasks (Devlin et al., 2019). The paper uses BERT-base for comparison, which has around 110 million parameters.

- **DistilBERT** is a distilled version of BERT, designed to be smaller, faster, cheaper, and lighter while retaining a significant portion of BERT's performance (Sanh et al., 2019). It achieves this through knowledge distillation during the pre-training phase. DistilBERT has approximately 67.0 million tunable parameters.

Table a.2: Test accuracy (mean $\pm$ 2×std. error over 5 runs, shown as percentages) of AFL algorithms on 20Newsgroup with DistilBERT and BERT-base under label distribution shift ($\alpha$) and low system delay ($\beta = 5$).

| Algorithm | DistilBERT | | | BERT-base | | |
|---|---|---|---|---|---|---|
| | $\alpha = 0.1$ | $\alpha = 1.0$ | $\alpha = 10$ | $\alpha = 0.1$ | $\alpha = 1.0$ | $\alpha = 10$ |
| Vanilla ASGD (Mishchenko et al., 2022) | $49.3 \pm 2.8\%$ | $59.1 \pm 2.2\%$ | $62.2 \pm 1.8\%$ | $54.2 \pm 2.9\%$ | $64.3 \pm 2.3\%$ | $67.1 \pm 1.9\%$ |
| Delay-Adaptive ASGD(Koloskova et al., 2022) | $52.4 \pm 2.6\%$ | $61.8 \pm 2.0\%$ | $65.3 \pm 1.6\%$ | $57.3 \pm 2.7\%$ | $66.8 \pm 2.1\%$ | $70.2 \pm 1.7\%$ |
| FedBuff (Nguyen et al., 2022) | $55.7 \pm 2.4\%$ | $65.2 \pm 1.8\%$ | $68.1 \pm 1.5\%$ | $60.4 \pm 2.5\%$ | $70.3 \pm 1.9\%$ | $73.1 \pm 1.6\%$ |
| CA$^2$FL (Wang et al., 2024b) | $61.6 \pm 2.0\%$ | $69.3 \pm 1.5\%$ | $71.2 \pm 1.2\%$ | $66.2 \pm 2.1\%$ | $74.1 \pm 1.6\%$ | $76.3 \pm 1.3\%$ |
| ACED (Ours, $\tau_{algo} = 50$) | $63.1 \pm 1.8\%$ | $70.7 \pm 1.3\%$ | $72.6 \pm 1.1\%$ | $68.3 \pm 1.9\%$ | $75.7 \pm 1.4\%$ | $77.6 \pm 1.1\%$ |
| ACE (Ours) | $\mathbf{63.7 \pm 1.7\%}$ | $\mathbf{71.4 \pm 1.2\%}$ | $\mathbf{73.1 \pm 1.0\%}$ | $\mathbf{68.7 \pm 1.8\%}$ | $\mathbf{76.6 \pm 1.3\%}$ | $\mathbf{78.2 \pm 1.0\%}$ |

**Performance on 20Newsgroup** The Table a.2 summarizes the accuracy of different AFL algorithms on the 20Newsgroup dataset using DistilBERT and BERT-base, under varying degrees of label distribution shift controlled by $\alpha$, and with a fixed low system delay ($\beta = 5$). The accuracies presented reflect performance after 500 server iterations. Reference accuracies for these models under a hypothetical synchronous, no-delay federated setup would generally be slightly higher than the values reported here for $\beta = 5$.

### F.3.3 REDUCING THE MEMORY OVERHEAD OF ACE BY COMPRESSION

A key insight from our theoretical and empirical analysis is the positive correlation between an algorithm's performance under client heterogeneity and the memory overhead required to manage all-client state. Algorithms that operate with minimal overhead, such as Vanilla ASGD (Mishchenko et al., 2022), inherently suffer from heterogeneity amplification because they lack the necessary information to correct for participation imbalance.

Conversely, state-of-the-art methods that effectively combat this issue, including **both CA$^2$FL (Wang et al., 2024b) and our proposed ACE**, rely on caching information from all $n$ clients. CA$^2$FL requires an $\mathcal{O}(nd)$ server-side cache for historical updates ($h_i^t$) to perform its calibration, while ACE requires an $\mathcal{O}(nd)$ cache for the latest gradients to perform its full aggregation.

Therefore, the $\mathcal{O}(nd)$ overhead should be viewed as a **necessary cost** for achieving top-tier performance and robustness in challenging AFL environments. The comparison in Table a.3 should be interpreted through this lens: the increased overhead of ACE and CA$^2$FL directly corresponds to their superior ability to handle the core challenges of ACE.

Table a.3: Comparison of storage overheads and convergence rates for various AFL algorithms. The table highlights a fundamental trade-off between memory efficiency and robustness to client heterogeneity. Algorithms with lower storage overhead, such as Vanilla ASGD, Delay-Adaptive ASGD, and FedBuff, are susceptible to heterogeneity amplification, as indicated by the presence of heterogeneity-dependent terms (non-vanishing $\zeta^2$ or $\tau\zeta^2$ interaction) in their convergence rates. Conversely, methods like CA$^2$FL and our proposed ACE/ACED achieve superior convergence by eliminating this amplification effect, but at the cost of a higher total system overhead of $\mathcal{O}(nd)$. This higher cost is necessary to cache state information from all clients, which is used to correct the participation imbalance bias. Notably, ACE offers implementation flexibility, allowing this $\mathcal{O}(nd)$ overhead to be concentrated on the server (Direct Aggregation) or distributed among the clients (Incremental Update).

| Algorithm | Client-Side Overhead | Server-Side Overhead | Total Cost | Convergence Rate $\mathcal{O}(\cdot)$ | Notes |
|---|---|---|---|---|---|
| Vanilla ASGD (Mishchenko et al., 2022) | $\mathcal{O}(1)$ | $\mathcal{O}(1)$ | $\mathcal{O}(n)$ | $\sqrt{\frac{\sigma^2}{T} + \frac{n}{T}} + \zeta^2$ (with non-vanishing error $\zeta^2$ as $T$ increases) | The client and server are stateless, leading to low overhead but susceptibility to bias. |
| Delay-Adaptive ASGD (Koloskova et al., 2022) | $\mathcal{O}(1)$ | $\mathcal{O}(1)$ | $\mathcal{O}(n)$ | $\sqrt{\frac{\sigma^2+\zeta^2}{T}} + \frac{\sqrt[3]{\tau_{\mathrm{avg}}\frac{1}{n}\sum_{i=1}^{n}\tau_{\mathrm{avg}}^i\zeta_i^2}}{T^{2/3}}$ (with heterogeneity amplification $\tau\zeta^2$) | The client and server are stateless, leading to low overhead but susceptibility to bias. |
| FedBuff (Nguyen et al., 2022) | $\mathcal{O}(1)$ | $\mathcal{O}(Md)$ | $\mathcal{O}(n+Md)$ | $\sqrt{\frac{\sigma^2+K\zeta^2}{mKT}} + \frac{K\tau_{\mathrm{avg}}\tau_{\max}\zeta^2+\tau_{\max}\sigma^2}{T}$ (with heterogeneity amplification $\tau\zeta^2$) | The server buffers $M$ updates; performance is limited by the $\tau\zeta^2$ term. |
| CA$^2$FL (Wang et al., 2024b) | $\mathcal{O}(1)$ | $\mathcal{O}(nd)$ | $\mathcal{O}(nd)$ | $\frac{\Delta+\sigma^2}{\sqrt{TKM}} + \frac{\sigma^2+K\zeta^2}{TK} + \frac{(\tau_{\max}+\rho_{\max})\sigma^2}{T}$ (**No heterogeneity amplification** $\tau\zeta^2$ term due to calibration) | Server caches state for all $n$ clients to calibrate updates, mitigating direct amplification. |
| ACE (Direct Aggregation) | $\mathcal{O}(1)$ | $\mathcal{O}(nd)$ | $\mathcal{O}(nd)$ | $\frac{\Delta}{\sqrt{nT}} + \frac{L\sigma^2}{\sqrt{nT}} + \frac{L^2\tau_{\max}\sigma^2}{T}$ (**No heterogeneity amplification** $\tau\zeta^2$) | Server caches the latest gradient from all $n$ clients, eliminating the $\zeta^2$ term from the rate. |
| ACE (Incremental Update) | $\mathcal{O}(d)$ | $\mathcal{O}(d)$ | $\mathcal{O}(nd)$ | $\frac{\Delta}{\sqrt{nT}} + \frac{L\sigma^2}{\sqrt{nT}} + \frac{L^2\tau_{\max}\sigma^2}{T}$ (**No heterogeneity amplification** $\tau\zeta^2$) | Reallocates the total $\mathcal{O}(nd)$ system cost, shifting storage burden from server to clients. |
| ACED (Ours) | $\mathcal{O}(1)$ | $\mathcal{O}(nd)$ | $\mathcal{O}(nd)$ | $\ldots+\frac{L^2\tau_{\mathrm{algo}}\sigma^2}{T}\frac{n}{n_{\mathrm{avg}}}+(\zeta^2+G^2)\sum\frac{(n-n_t)^2}{T}$ (**No heterogeneity amplification** $\tau\zeta^2$, a vanishing error term $\frac{\zeta^2}{T}$ as $T$ increases) | The client is stateless. The server still needs to cache the latest gradients from all $n$ clients to dynamically select the aggregation subset based on the delay threshold. |

As demonstrated in Table a.3, the total system overhead of ACE is comparable to that of CA$^2$FL. The choice between our Direct Aggregation (server-heavy) and Incremental Update (client-heavy) implementations allows for flexibility in deploying this all-client principle, depending on where the system's resource capacity lies. In other words, for the two implementations of ACE, the total system overhead remains the same (Direct Aggregation: client $n \cdot \mathcal{O}(1)$ + server $\mathcal{O}(nd)$; Incremental Update: client $n \cdot \mathcal{O}(d)$ + server $\mathcal{O}(d)$, for a total system state of $\mathcal{O}(nd)$). The incremental approach merely reallocates the storage burden between clients and the server, rather than reducing it. Given that this overhead is a fundamental requirement for high performance, we argue that practical optimization efforts should focus on reducing the size of individual gradient vectors. To this end, we investigate

the use of 8-bit quantization as a promising direction to significantly lower the memory overhead while preserving the performance benefits of our approach.

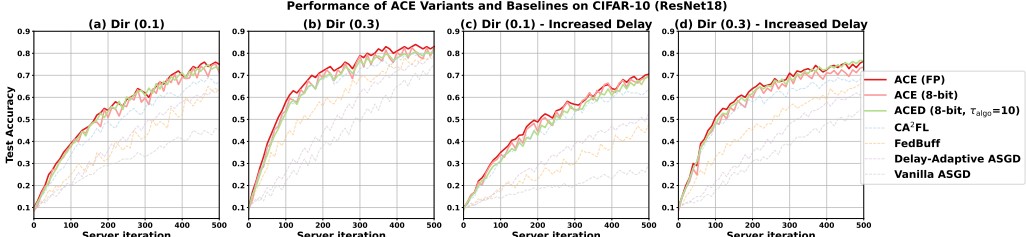

Figure a.3: Impact of 8-bit server-side gradient quantization on the test accuracy of ACE and ACED on CIFAR-10 with ResNet18. The 8-bit variations achieve comparable final performance to the full-precision implementation.

A practical consideration for **ACE** and its variant **ACED** is the server-side memory required to store the latest gradients from all clients for the full aggregation step, especially when dealing with large-scale models possessing a massive number of trainable parameters. This section is motivated by the need to address this potential limitation and explore memory-efficient implementations. We investigate the impact of applying 8-bit quantization to the gradients cached at the server before they are aggregated. The goal is to determine if a significant reduction in memory overhead can be achieved while largely preserving the convergence speed and final performance benefits demonstrated by the full-precision versions of ACE and ACED.

To achieve this, we introduce **ACE-8bit** and **ACED-8bit**. The core modification lies in how the server handles the incoming gradients from clients. Specifically:

- In both ACE-8bit and ACED-8bit, clients compute and transmit their gradients, $\nabla f_i(w^{t-\tau_i^t}; \xi_i)$, using full precision as in the original algorithms.

- Upon receiving a gradient from client $i$, say $U_i^t = \nabla f_i(w^{t-\tau_i^t}; \xi_i)$, the server quantizes this gradient to an **8-bit representation**, denoted as $Q(U_i^t)$. This can be achieved using standard unbiased quantization techniques.

- The server then stores this quantized gradient $Q(U_i^t)$ in its cache for client $i$.

- For the global model update, ACE-8bit computes $u^t = \frac{1}{n} \sum_{i=1}^n Q(U_i^t)$, utilizing the latest available quantized gradient from all $n$ clients. Similarly, ACED-8bit computes its update $u_{\text{BDA}}^t = \frac{1}{n_t} \sum_{i \in A(t)} Q(U_i^{\text{cache}})$, using the quantized gradients from the set $A(t)$ of active clients whose information meets the delay threshold $\tau_{\text{algo}}$.

This approach directly reduces the memory overhead on the server for storing the gradient components from each client. In addition, this approach also illustrates the compatibility of ACE/ACED and the model compression algorithm, providing the possibility for the practical use of ACE/ACED in a large-scale federated learning system.

## F.4 Hyper-parameter Configurations

Table a.4: Hyper-parameters for CIFAR-10 Experiments (Section 5, Appendix F.3.3). Total clients $n = 100$. $T = 500$ server iterations.

| Hyper-parameter | ACE | FedBuff | CA$^2$FL | Vanilla ASGD |
|---|---|---|---|---|
| Model | | ResNet-18 | | |
| Global Learning Rate ($\eta$) | $0.2\sqrt{n/T}$ | $0.2\sqrt{n/T}$ | $0.2\sqrt{n/T}$ | $0.2\sqrt{n/T}$ |
| Local Learning Rate ($\eta_l$) | N/A ($K = 1$) | $5 \times 10^{-2}$ | $5 \times 10^{-2}$ | N/A ($K = 1$) |
| Optimizer (Local step) | SGD (momentum 0.9) | SGD (momentum 0.9) | SGD (momentum 0.9) | SGD (momentum 0.9) |
| Batch Size | 50 | 50 | 50 | 50 |
| $\alpha$ (Dirichlet) | {0.1, 0.3} | {0.1, 0.3} | {0.1, 0.3} | {0.1, 0.3} |
| $\beta$ (Mean Exp. Delay Param.) | {5, 30} | {5, 30} | {5, 30} | {5, 30} |
| Buffer Size ($M$) | N/A | 10 | 10 | N/A |
| Concurrency ($M_c$) | N (all clients) | 20 | 20 | 1 (sequential) |

Table a.5: Hyper-parameters for CIFAR-100 Experiments (Appendix F.3.1). Total clients $n = 100$. $T = 500$ server iterations.

| Hyper-parameter | ACE | FedBuff | CA$^2$FL | Vanilla ASGD |
|---|---|---|---|---|
| Model | | ResNet-18 | | |
| Global Learning Rate ($\eta$) | $0.2\sqrt{n/T}$ | $0.2\sqrt{n/T}$ | $0.2\sqrt{n/T}$ | $0.2\sqrt{n/T}$ |
| Local Learning Rate ($\eta_l$) | N/A ($K = 1$) | $5 \times 10^{-2}$ | $5 \times 10^{-2}$ | N/A ($K = 1$) |
| Optimizer (Local step) | SGD (momentum 0.9) | SGD (momentum 0.9) | SGD (momentum 0.9) | SGD (momentum 0.9) |
| Batch Size | 50 | 50 | 50 | 50 |
| $\alpha$ (Dirichlet) | {0.1, 0.3, 1.0, 10.0} | {0.1, 0.3, 1.0, 10.0} | {0.1, 0.3, 1.0, 10.0} | {0.1, 0.3, 1.0, 10.0} |
| $\beta$ (Mean Exp. Delay Param.) | {1, 5, 20, 30} | {1, 5, 20, 30} | {1, 5, 20, 30} | {1, 5, 20, 30} |
| Buffer Size ($M$) | N/A | 10 | 10 | N/A |
| Concurrency ($M_c$) | N (all clients) | 20 | 20 | 1 (sequential) |

Table a.6: Hyper-parameters for 20Newsgroup (BERT fine-tuning) Experiments (Appendix F.3.2). Total clients $n = 20$. $T = 100$ server iterations.

| Hyper-parameter | ACE | FedBuff | CA$^2$FL | Vanilla ASGD |
|---|---|---|---|---|
| Model | | DistilBERT / BERT-base | | |
| Global Learning Rate ($\eta$) | $0.2\sqrt{n/T}$ | $0.2\sqrt{n/T}$ | $0.2\sqrt{n/T}$ | $0.2\sqrt{n/T}$ |
| Local Learning Rate ($\eta_l$) | N/A ($K = 1$) | $5 \times 10^{-4}$ | $5 \times 10^{-4}$ | N/A ($K = 1$) |
| Optimizer (Local step) | AdamW | AdamW | AdamW | AdamW |
| Batch Size | 32 | 32 | 32 | 32 |
| $\alpha$ (Dirichlet) | {0.1, 1.0, 10.0} | {0.1, 1.0, 10.0} | {0.1, 1.0, 10.0} | {0.1, 1.0, 10.0} |
| $\beta$ (Mean Exp. Delay Param.) | 5 | 5 | 5 | 5 |
| Buffer Size ($M$) | N/A | 10 | 10 | N/A |
| Concurrency ($M_c$) | N (all clients) | 10 | 10 | 1 (sequential) |

**General Setup** To ensure a theoretically consistent comparison that directly aligns with our analytical framework, the local computational workload for every client across **all compared algorithms** was standardized to a **single gradient descent step** ($K = 1$) per communication round. This approach prioritizes a direct test of the different aggregation strategies by eliminating the confounding effects of local client drift. Specifically:

- For algorithms theoretically based on a single gradient update, such as ACE, Vanilla ASGD, and Delay-Adaptive ASGD, each client computes a stochastic gradient on *one mini-batch* of its local data using the unmodified global model it received. This single gradient is then sent to the server.

- For algorithms designed to support multiple local steps, namely FedBuff and CA$^2$FL, we explicitly set their local step parameter to $K = 1$. This ensures they also perform only a single mini-batch update before communication, making their update mechanism directly comparable to the other methods under our theoretical lens.

This setup provides a clear evaluation of how each aggregation method handles staleness and participation bias, which is the central focus of our paper. For CIFAR datasets, this single step was performed using SGD with momentum 0.9. For 20Newsgroup (BERT) experiments, the AdamW optimizer was used for the local step.

Data heterogeneity across clients is configured using a Dirichlet distribution controlled by parameter $\alpha$. For CIFAR-10, $\alpha \in \{0.1, 0.3\}$. For CIFAR-100, $\alpha \in \{0.1, 0.3, 1.0, 10.0\}$. For 20Newsgroup, $\alpha \in \{0.1, 1.0, 10.0\}$.

Client update delays are generated using an exponential distribution governed by a mean parameter $\beta$. For CIFAR-10, $\beta \in \{5, 30\}$. For CIFAR-100, $\beta \in \{1, 5, 20, 30\}$. For the 20Newsgroup experiments detailed in Table a.2, a fixed $\beta = 5$ was used. All resulting delays are inherently bounded.

The tables summarize key hyper-parameters. Global learning rates are tuned based on scaling $\sqrt{n/T}$ with parameter $c \in \{10, 5, 2, 1, 0.5, 0.2, 0.1\}$, and local learning rates are tuned based on grid search.

For the ACED variant, the additional hyper-parameter $\tau_{\text{algo}}$ is specified depending on the experimental case/setting.

