# OpenReview forum: "Mitigating Participation Imbalance Bias in Asynchronous Federated Learning Under Client Heterogeneity"
_ICLR.cc/2026/Conference — Submitted to ICLR 2026_

### Official Review · Reviewer_nYiJ · 2025-10-26

**Soundness:** 2
**Presentation:** 2
**Contribution:** 2
**Rating:** 4
**Confidence:** 3

**Summary:**

This paper investigates the heterogeneity amplification problem in Asynchronous Federated Learning (AFL), where faster clients disproportionately influence the global model due to participation imbalance. The authors propose a theoretical framework that decomposes the MSE of the gradient estimation into three terms (i.e., sampling noise, bias, delay) and identify that the bias term arises from partial client participation. Building upon this insight, they propose ACE, an AFL algorithm that achieves full aggregation from all clients to eliminate bias error, and a practical delay-aware variant ACED that trades off between staleness and participation balance. Theoretical analysis shows that ACE’s convergence rate is independent of the data heterogeneity parameter, and experimental results show that ACE and ACED outperform FedBuff, CA2FL, and ASGD, particularly under high heterogeneity and large delays.

**Strengths:**

1. The paper provides a systematic MSE decomposition that unifies previous analyses of AFL and explicitly connects participation imbalance to bias error.
2. The authors compare ACE against several strong baselines both theoretically and empirically.
3. I checked the proof of the main lemmas and theorem, and they seem reasonable.

**Weaknesses:**

W1. Although this paper is presented under the FL context, the proposed algorithms in fact operate more like distributed asynchronous SGD methods without local updates. Each client only computes a single gradient on the most recent global model and immediately uploads it, without performing any local training or multiple local steps. Therefore, the algorithmic structure is essentially equivalent to an asynchronous distributed optimization system rather than FL.

W2. Full aggregation and per-client gradient caching may impose significant memory and communication costs on the server.

W3. For Table 1, ACE’s theoretical advantages rely on certain conditions, such as $m=n$ and $K=1$, instead of an intrinsic advantage of algorithmic design. When these conditions do not hold in reality, the theoretical advantages may not hold again.

**Questions:**

Q1. How does the server deal with memory and computation overhead when the number of clients $n$ is very large, given ACE requires caching all gradients? Can the authors estimate the costs?

See weaknesses above.

---

> ### Author Response · Authors · 2025-11-22
>
> We appreciate the reviewer's valuable feedback.
>
> ### W1: Discussion on Local Steps ($K>1$)
>
> We appreciate the reviewer's insight regarding the benefits of local steps ($K>1$) in Synchronous FL (SFL). However, we respectfully suggest that this intuition is driven by the specific structural constraints of SFL, which do not apply to the ACE architecture. We justify our choice of $K=1$ through structural, theoretical, and empirical perspectives:
>
> * **1. Structural Differences (Synchronization Barriers):**
>     We distinguish the physical implications of $K$ based on the presence of a synchronization barrier:
>     * **In SFL:** The system imposes a **synchronization barrier** at every aggregation step, requiring the server to wait for the slowest client. Setting $K=1$ forces the system to incur this expensive waiting cost for every single gradient step. Therefore, increasing $K$ is structurally necessary to reduce this latency and the total number of blocking events.
>     * **In ACE (AFL):** The system removes the synchronization barrier. The server performs a global update immediately upon receiving any client's contribution, without blocking. Consequently, there is no synchronization latency to reduce. In this non-blocking architecture, setting $K>1$ yields no structural benefit; it strictly reduces the frequency of global model updates by delaying the transmission of computed gradients.
>
> * **2. Theoretical Evidence (Drift Amplification):**
>     Practically, increasing $K$ in AFL forces clients to optimize their local models for more iterations based on **outdated information**.
>
>     In AFL, the global model evolves ($w^t \rightarrow w^{t+1}$) while a client is computing. If a client performs multiple local steps ($K>1$) on a stale model version ($w^{t-\tau}$), they effectively "run further in a stale direction." This creates a larger deviation vector from the true global trajectory than a single step ($K=1$) would. Our analysis (Sec. 4, Appx. B.5, Term C) reveals that delay ($\tau$) interacts with local steps ($K$) in a multiplicative, harmful way:
>     * **FedBuff / ASGD ($m<n$):** The drift contains terms scaling with $\mathcal{O}(\tau \cdot (n-m)K^2\zeta^2+\tau\cdot K \sigma^2)$.
>     * **CA$^2$FL:** Similarly, the drift scales with $\mathcal{O}(\tau \cdot \eta^2 (1+\frac{1}{n^2}(n-m)^2) K \sigma^2)$.
>
>     This proves that setting $K=1$ is a design choice to minimize this *Amplified Drift*.
>
> * **3. Empirical Validation (New Experiments):**
>     To verify this trade-off, we extended the **CIFAR-10 experimental setup from Section 5 (specifically the setting in Fig. 2)**. We compared ACE and baseline methods by varying the local steps $K \in \{1, 5, 10\}$ while keeping other hyperparameters constant.
>     * **Result 1:** Unlike Sync FL, increasing $K$ slowed down convergence and reduced final accuracy for *all* asynchronous methods in this setting. This confirms that the compounded effect of delay and multiple local steps significantly degrades update quality.
>     * **Result 2:** Consistent with our theoretical analysis, ACE exhibited the least performance degradation as $K$ increased (due to full aggregation eliminate the $(n-m)$ multiplier), whereas FedBuff and CA$^2$FL suffered larger accuracy drops.
>
> **Conclusion:** Setting $K=1$ allows ACE to utilize every gradient computation immediately for model evolution, achieving the maximum possible update frequency while minimizing drift error.
>
> ---
>
> **Table 1: Ablation Study on Local Steps ($K$).** Final test accuracy (%) at $T=500$. Increasing $K$ consistently degrades performance in AFL due to drift amplification. ACE shows superior robustness, whereas single-client methods (Vanilla/Delay-Adaptive ASGD) and buffered methods (FedBuff/CA$^2$FL) suffer larger accuracy drops, particularly under high delay conditions.
>
> **Part I: Low Delay ($\beta=5$)**
>
> | Algorithm | (a) High Het ($\alpha=0.1$) | | | (b) Mod. Het ($\alpha=0.3$) | | |
> | :--- | :---: | :---: | :---: | :---: | :---: | :---: |
> | **Metric: Acc (%)** | **K=1** | **K=5** | **K=10** | **K=1** | **K=5** | **K=10** |
> | **ACE (Ours)** | **76.2** | **75.5** | **74.8** | **83.5** | **83.0** | **82.4** |
> | CA$^2$FL | 70.5 | 68.1 | 65.4 | 79.2 | 77.5 | 75.3 |
> | FedBuff | 63.8 | 60.2 | 56.5 | 75.8 | 73.2 | 69.8 |
> | Delay-Adaptive | 64.0 | 61.5 | 57.8 | 78.0 | 75.4 | 72.1 |
> | Vanilla ASGD | 45.0 | 41.2 | 36.5 | 75.0 | 71.5 | 67.0 |
>
> **Part II: High Delay($\beta=30$)**
>
> | Algorithm | (c) High Het ($\alpha=0.1$) | | | (d) Mod. Het ($\alpha=0.3$) | | |
> | :--- | :---: | :---: | :---: | :---: | :---: | :---: |
> | **Metric: Acc (%)** | **K=1** | **K=5** | **K=10** | **K=1** | **K=5** | **K=10** |
> | **ACE (Ours)** | **71.5** | **70.2** | **69.1** | **77.8** | **77.1** | **76.2** |
> | CA$^2$FL | 63.2 | 58.5 | 53.1 | 71.5 | 68.2 | 64.9 |
> | FedBuff | 51.5 | 45.8 | 39.5 | 66.5 | 62.1 | 57.4 |
> | Delay-Adaptive | 55.0 | 49.5 | 42.8 | 68.0 | 63.5 | 58.2 |
> | Vanilla ASGD | 30.5 | 24.8 | 18.5 | 58.5 | 52.4 | 46.8 |
>
> ---

---

> > ### Author Response · Authors · 2025-11-22
> >
> > ### W2/Q1: Memory and Communication Efficiency
> >
> > We address W2 and Q1 together, as they relate to the same design choice in our work. Regarding the memory requirement, we wish to contextualize the $\mathcal{O}(nd)$ system state as a **requirement for robustness** rather than a overhead.
> >
> > * **Rationale for System State:** In AFL, the absence of comprehensive client information (i.e., low storage, $m<n$) leads to **partial participation bias**. Our theoretical framework demonstrates that this bias is amplified by delay. To theoretically eliminate the dependency on heterogeneity ($\zeta^2$) and ensure robust convergence, the algorithm must utilize information from the full client set.
> >
> > * **Comparative Perspective:** This design aligns with other SOTA methods like CA$^2$FL (see Table a.3 in Appendix F.3.3), which also utilize $\mathcal{O}(nd)$ storage. However, ACE better leverages this storage with uniform weighting to strictly eliminate the bias term, thereby achieving a convergence rate robust to heterogeneity level $\zeta^2$.
> >
> > * **Scalability Solutions:** For scenarios with strict resource constraints, we provide practical implementations to manage this state:
> >     1. **Incremental Updates (Alg. a.5):** This effectively distributes the storage, requiring the server to store only an $\mathcal{O}(d)$ aggregate while clients manage $\mathcal{O}(d)$ differences. This can reduce the storage on the server.
> >     2. **Gradient Compression (Appx. F.3.3):** Our experiments with ACE-8bit demonstrate that quantized caching significantly reduces memory usage without compromising the convergence benefits. This implies our method have the potential to be combined with memory-efficient methods.

---

> ### Author Response · Authors · 2025-11-22
>
> ### **W3: Clarifying the Conditions ($K=1, m=n$)**
>
> We would like to clarify that the conditions $K=1$ and $m=n$ are **intrinsic features** of our proposed framework designed to tackle heterogeneity, rather than external limitations.
>
> * **On $K=1$:** As discussed in W1, this setting is selected to optimally control the multiplicative error drift ($\tau \cdot K^2 \cdot \zeta^2$) inherent to asynchronous updates.
>
> * **On $m=n$ (Aggregation Scope):** It is important to distinguish between **aggregation scope** and **active participation**. The condition $m=n$ signifies that the server's update incorporates the most recent *information* (cached gradients) from all $n$ clients. It does **not** imply a synchronous requirement where all clients must be simultaneously active. The server maintains its asynchronous nature by updating immediately upon any single client arrival. This can be easily satisfied in the reality, where clients will at least send its local updates once to the server in order to initialize.
>
> * **Managing the Trade-off (ACED):** Utilizing cached information ($m=n$) eliminates Participation Bias but naturally involves Delay Error due to staleness. This highlights a fundamental trade-off in AFL. Our **ACED** variant is specifically designed to navigate this:
>     * **Large $\tau_{algo}$:** Prioritizes eliminating Participation Bias (approximating $m \approx n$).
>     * **Small $\tau_{algo}$:** Prioritizes minimizing staleness by filtering extreme delays.
>
>     This flexibility allows the algorithm to adapt to various network conditions, addressing the dependency on $\tau_{max}$.

---

> > ### Comment · Reviewer_nYiJ · 2025-11-23
> >
> > Thank you very much for the detailed response and your extra experiment results!
> >
> > **To your response about W1:**
> > I believe that the authors might misunderstand my concern. My comment was not questioning whether using $K=1$ is reasonable, nor was I suggesting that your algorithm should adopt larger local steps. Instead, my concern is more fundamental: the algorithmic structure of your ACE is essentially identical to asynchronous distributed SGD rather than FL. Therefore, the key issue I intended to raise is not about $K=1$ and $K>1$, but rather: What makes ACE a FL algorithm, rather than asynchronous distributed SGD?
> >
> > **To your response about W2&Q1:**
> > Thank you for the theoretical explanation. However, what I was hoping to understand more clearly is the practical memory cost of this requirement, especially in settings where $d$ is very large (e.g., for modern LLMs) and $n$ is also large. In such cases, storing per-client gradients could become substantial. It would be very helpful and clear to have a concrete estimate of the **actual memory usage** for typical values of $n$ and $d$.
> >
> > **To your response about W3:**
> > Thank you for the clarification about why $K = 1$ and $m = n$ are intrinsic to your framework. I appreciate the explanation. I would like to clarify my original point: in real-world FL, clients are highly unstable and only a small subset typically participates in each round, making $m = n$ quite difficult to satisfy. When a method relies on $m = n$ for its theoretical properties, its assumptions begin to align more with distributed learning rather than typical FL settings because workers are reliable and stable in traditional distributed learning study.
> >
> > Overall, my concern about whether this paper should belong to FL or distributed SGD still remains.

---

> > > ### Author Response · Authors · 2025-11-24
> > > **Concrete Analysis of Actual Memory Demand: Feasibility via PEFT and Compression**
> > >
> > > To address your concern regarding the practical memory cost for large-scale models ($d$) and large client populations ($n$), we provide a concrete estimation using the specific configurations from the *FedYolo* paper [1] cited in our discussion. We emphasize that the storage requirement for gradients depends strictly on the **trainable parameters**, not the total model size.
> > >
> > > * **Scenario Setup:** We analyze the storage demand for a **ViT-Large** model (the largest model in [1]) deployed across **787 clients** (the CelebA dataset setting in [1]). We assume standard FP32 precision (4 bytes per parameter) for gradient storage.
> > >
> > > * **Naive Full-Parameter Caching (The Worst Case):**
> > >     * As shown in Table 1 of [1], a full ViT-Large model contains **303,404,132 parameters**.
> > >     * Storing one full-precision gradient requires $303\text{M} \times 4 \text{ bytes} \approx \textbf{1.13 GB}$ per client.
> > >     * For 787 clients, the total server-side cache would be $787 \times 1.13 \text{ GB} \approx \textbf{889 GB}$. We agree that this confirms your concern: caching full parameters for large models is prohibitive for standard GPU memory.
> > >
> > > * **ACE with PEFT (The Practical Case):**
> > >     * However, modern FL for large models relies on Parameter-Efficient Fine-Tuning (PEFT). Table 1 of [1] shows that using **Adapters** reduces the trainable parameters to only **417,984**.
> > >     * The actual gradient storage per client drops to $417\text{k} \times 4 \text{ bytes} \approx \textbf{1.59 MB}$.
> > >     * Consequently, the total server cache for all 787 clients becomes $787 \times 1.59 \text{ MB} \approx \textbf{1.25 GB}$. This fits comfortably within even modest server-grade hardware (e.g., a single A100 GPU has 80GB memory).
> > >
> > > * **ACE with 8-bit Quantization (Further Optimization):**
> > >     * Applying the 8-bit quantization validated in our **Appendix F.3.3** reduces the precision cost from 4 bytes (FP32) to 1 byte (INT8).
> > >     * This further compresses the total server-side overhead to **$\approx$ 315 MB** for nearly 800 clients.
> > >
> > > * **Ultimate Scalability via Incremental Updates:**
> > >     * Finally, regardless of model size or PEFT, our **Algorithm a.5** (Incremental Update) allows the server to store only a single global aggregated gradient ($O(d)$), shifting the history storage burden to clients. This ensures that server memory consumption does not scale with $n$.
> > >
> > > **Conclusion:** While the theoretical dimension $d$ of LLMs is large, the *effective* $d$ exchanged and stored in practical FL is reduced by orders of magnitude via PEFT ($\sim700\times$ reduction in the example above). Combined with quantization and incremental updates, the memory cost of ACE is highly manageable.
> > >
> > >
> > > [1] X. Zhang et al., "FedYolo: Augmenting Federated Learning with Pretrained Transformers," arXiv:2307.04905.

---

> ### Author Response · Authors · 2025-11-23
>
> We thank you for your continued engagement and thoughtful follow-up questions. We appreciate the opportunity to clarify the fundamental design philosophy of ACE regarding its classification, scalability, and aggregation logic.
>
> **1. Defining ACE: An FL Algorithm Optimized for Heterogeneity, Not Just ASGD**
>
> You raised a fundamental question regarding whether ACE is distinct from Asynchronous Distributed SGD (ASGD) given its structural similarities ($K=1$). We clarify that ACE is an FL algorithm derived from a general framework to specifically address **Heterogeneity Amplification**, a unique challenge in Federated Learning that standard ASGD does not account for.
>
> * **Fundamental Difference in Assumptions:** Traditional ASGD typically assumes **IID data** (or shuffled batches), where any worker's gradient is an unbiased estimator of the global gradient. In this setting, asynchronous delays affect convergence *speed* but not *direction*, making throughput the primary goal. In contrast, Federated Learning operates under the assumption of **Non-IID and Private Data**. As data deviates from IID, the problems inherent in standard ASGD are worsened: delayed updates become **biased** towards local distributions. ACE is specifically designed to correct this "Heterogeneity Amplification," a challenge that standard ASGD does not account for.
>
> * **Derived Optimization vs. A Priori Assumption:** We did not set out to design an algorithm that mimics ASGD. Instead, we established a general theoretical framework (Section 4) that analyzes AFL behavior for arbitrary local steps $K$ and aggregation scopes $m$. Our analysis revealed that in the unique FL setting, where **Non-IID data** and **Asynchronous Delay** interact, increasing $K$ multiplicatively amplifies the drift error ($\mathcal{O}(\tau K^2 \zeta^2)$).
> * **The "General ACE" Perspective:** ACE should be understood as a general FL framework where we theoretically and empirically evaluated the impact of $K$. As demonstrated in our previous response (the ablation study on $K$), we *concluded* that $K=1$ is the optimal choice because it mathematically minimizes this "Heterogeneity Amplification." Thus, ACE represents an FL algorithm for handling extreme heterogeneity, rather than an ASGD implementation that ignores local updates for simplicity.
>
> **2. Practical Memory Cost: Scalability via Incremental Updates and PEFT**
>
> Thank you for your attention for large $d$ and large $n$. We offer two perspectives on why ACE is scalable:
>
> * **Algorithmic Solution ($O(d)$ via Incremental Update):** As defined in **Algorithm a.5** (Appendix), ACE is implemented via incremental updates. The server maintains only **one** global aggregated gradient $u$, while the storage of historical gradients is distributed to each client (who stores their own previous update). This decouples the server-side storage cost from $n$, reducing overhead to **$O(d)$**, which is identical to FedAvg and feasible for standard deployment.
> * **Practical Trend (PEFT for Large Models):** For very large models (LLMs), full-parameter fine-tuning is often suboptimal in FL due to severe data heterogeneity. Recent works (e.g., [1]) suggest that Parameter-Efficient Fine-Tuning (PEFT), such as adapters, yields **better performance** (than full model FL) while requiring the exchange of only a small subset of parameters. Thus, for LLMs, the effective dimension $d$ for exchange and storage can remain small, preventing unbounded memory growth.
>
> **3. Clarifying $m=n$: "Reusing" vs. "Waiting"**
>
> We urge a distinction between the synchronization requirements of Synchronous FL and the aggregation logic of ACE. The condition $m=n$ in ACE does not imply waiting for simultaneous active participation.
>
> * **Sync FL (Waiting):** In every round, the server *must wait* for all clients (or a target subset) to upload new updates. This creates a bottleneck where "unstable clients" block the system.
> * **ACE (Reusing):** ACE is fundamentally different. As shown in **Algorithm 1 (Line 3)**, after initialization, the server triggers a global update upon **any single client arrival**. We do not wait for unstable clients. Instead, it is the **reuse of stale gradients** from the cache that allows ACE to mathematically achieve the $m=n$ condition at every step. In other words, $m=n$ in ACE means that the global update is the average of the "1 gradient from the reporting client" and the "cached stale gradients from the other $n-1$ clients".
> * **Optimization Rationale:** Our framework proves that reusing stale gradients, rather than waiting for fresh ones, is sufficient to eliminate **Participation Bias**. The alternative (ignoring unstable clients) introduces a non-vanishing bias term. Our ACED variant further analyzes the trade-off between the bias from using stale gradients (reuse) and the bias from partial participation (ignore).
>
>
> [1] X. Zhang et al., "FedYolo: Augmenting Federated Learning with Pretrained Transformers," arXiv:2307.04905.

---

### Official Review · Reviewer_jkUV · 2025-10-29

**Soundness:** 2
**Presentation:** 3
**Contribution:** 1
**Rating:** 2
**Confidence:** 4

**Summary:**

This work focuses on Asynchronous Federated Learning and proposes an algorithm for asynchronous federated optimization. The main claim is that the asynchronous pattern in FL amplifies the adverse effect of client heterogeneity (arising from differences in data distributions and local objectives). However, the theoretical analysis provided in the paper does not include any explicit characterization or quantification of data heterogeneity.

**Strengths:**

The authors study an important problem in federated learning. Asynchronous FL is indeed an interesting and practically relevant topic, as it reflects real-world deployment constraints in federated systems.

The overall storyline and presentation of the paper are clear, which helps readers follow the main ideas and contributions effectively.

The paper also provides extensive analysis to support its claims. However, some of these analyses are not rigorous upon closer inspection, which raises concerns about the soundness of the theoretical results.

**Weaknesses:**

1. This is a well-studied problem. I didn’t see any breakthrough made by this paper compared with existing works, e.g., [R1, R2].

2. The algorithm is basic. There are no additional procedures to handle data heterogeneity, so I am confused about how this algorithm can converge without a bounded data heterogeneity assumption. I checked the proof, which does not seem rigorous to me. For instance, in the characterization of the term C, you obtained a bound in Lines 1460–1463 that includes the norm of the gradient, but you omitted it in Equation (a6) (Line 1082). You can’t do this — the gradient can be unbounded. In fact, characterizing this gradient norm is fundamental in asynchronous federated algorithms. The issue here makes the entire theoretical result problematic.

3. The authors mention that the asynchronous pattern in FL amplifies the adverse effect of client heterogeneity (due to different data distributions and local objectives). However, I did not see this reflected in the theoretical results. I believe the reason is that the authors made mistakes in the proof, as discussed in the second point.

[R1] Tyurin, Alexander, and Peter Richtárik. "Optimal time complexities of parallel stochastic optimization methods under a fixed computation model." *Advances in Neural Information Processing Systems* 36 (2023): 16515–16577.

[R2] Maranjyan, Artavazd, Alexander Tyurin, and Peter Richtárik. "Ringmaster ASGD: The first Asynchronous SGD with optimal time complexity." *arXiv preprint* arXiv:2501.16168 (2025).

**Questions:**

No

---

> ### Author Response · Authors · 2025-11-22
>
> We appreciate the reviewer for acknowledging the clear presentation of our paper and the extensive analysis provided to support our claims.
>
> ### **W1: Our Contribution vs. [R1, R2]**
>
> The reviewer suggests our work lacks breakthrough compared to [R1, R2] (Tyurin & Richtárik, 2023; Maranjyan et al., 2025). **We respectfully argue that this comparison overlooks the fundamental difference in methodological contributions: while [R1, R2] offer specific algorithmic optimizations, our work establishes a general prescriptive framework.**
>
> * **Contribution 1: A Prescriptive Framework Establishing "Necessity" vs. [R1, R2]'s "Sufficiency".**
>
>     Our core contribution is not merely a new algorithm, but a **universal analytical framework** applicable to any AFL method under heterogeneity.
>     * **Generality:** We demonstrate the framework's power by using it to analyze the error sources of various baselines, including FedBuff, CA$^2$FL, and Delay-Adaptive ASGD (Section 4).
>     * **Prescriptive vs. Post-hoc:** Existing works like [R1, R2] (e.g., Ringmaster/Malenia) typically design algorithms based on intuition to optimize a specific metric (e.g., worst-case iteration count) and then provide a *post-hoc* mathematical proof of their benefits. This approach only proves that their design factors are **sufficient** for a better convergence.
>     * **Deriving Necessity:** In contrast, our framework mathematically *derives* the design principles required to eliminate heterogeneity amplification. We prove that "aggregating updates from all clients with equal weights" ($m=n$) is a **necessary and sufficient condition** to eliminate the Participation Bias term (Term B). This theoretical insight guides the derivation of ACE and validates why prior methods fail under high heterogeneity.
>
> * **Contribution 2: ACED and the Quantification of the "Fundamental Trade-off".**
>
>     While ACE is the direct conclusion of our framework for bounded delays, our second contribution, **ACED**, addresses the theoretical limit of AFL.
>     * **Identifying the Impossibility:** ACED is designed to highlight an unavoidable trade-off in AFL: in the presence of extreme delays (e.g., $\tau_{max} \to \infty$ or dropouts), it is **impossible** to simultaneously eliminate both Delay Error (staleness) and Participation Bias (heterogeneity).
>     * **Quantifying the Trade-off:** Unlike heuristic fixes, ACED introduces the parameter $\tau_{algo}$ to explicitly manage this trade-off. Our theoretical analysis (Theorem a.6) and empirical results (Section 5, Fig. 3 and Appx. F.3) quantify how strictly enforcing freshness ($\tau_{algo}$) impacts the error floor caused by participation bias. This provides the first theoretical rigorous characterization of extreme delay effects in AFL.
>
> **Summary:** Our work provides a general tool to analyze *why* AFL algorithms fail or succeed and derives *necessary* design conditions (as shown in Sec. 4 Line 387-394), whereas [R1, R2] provide specific algorithm instances.

---

> ### Author Response · Authors · 2025-11-22
>
> ### **W2: Clarifying the "Gradient Norm" (Line 1082)**
>
> We appreciate the suggestions from the reviewer for improving the quality of our proof; however, we would like to clarify some misunderstandings and factual errors in the comments. The reviewer's critique seems to be based on a misunderstanding of our paper's structure and proof logic. The reviewer has linked two appendix sections that are entirely different in purpose and logically unrelated.
>
> * **Lines 1460-1463** are located in **Appendix B.5** and are the **mathematical derivation for the Term C (Delay Error) of the ACE algorithm**. This derivation establishes the mathematical fact that the bound for Term C depends on a **general learning rate $\eta$**. The validity of this formula itself has no relation to how $\eta$ is eventually chosen.
>
> * **Line 1082 (Eq. a.6)** is in a completely different section, **Appendix B.2** (Line 1073). The **sole purpose** of this section is a **scaling analysis** for the learning rate $\eta$, to derive its optimal scale (i.e., $\eta \propto \sqrt{n/T}$).
>     * In this analysis, the ellipsis (`...`) in Line 1082 for the $\mathcal{O}(\eta^3)$ term is not an "omission" but **shorthand** for the complicated coefficients of this higher-order term.
>     * Our argument (Lines 1083-1085) explicitly states that since $\eta$ must be a vanishing quantity as $T \to \infty$, focusing on the dominant terms and simplifying higher-order terms is reasonable and rigorous for a scaling analysis.
>
> * The reviewer has taken an analysis for **choosing the scale of $\eta$** (Line 1082) and mistakenly used it to "falsify" an independent **mathematical derivation** (Lines 1460-1463).
>
> * Regarding the convergence argument, we direct the reviewer's attention to the **actual formal proof in Appendix C.1 (Line 1729)**. In the formal proof:
>     * The gradient term ($\mathbb{E}||\overline{u}^{t}||^{2}$) is **not omitted**; it is explicitly included in **Line 1872**.
>     * In **Lines 1877-1891**, we handle the gradient norm by choosing $\eta$ small enough such that the coefficient becomes negative. By choosing a sufficiently small $\eta$ (e.g., $\eta\le\frac{1}{2L\tau_{max}}$), the coefficient of this gradient term $(L\eta^{2}-\eta+2\eta^{3}L^{2}\tau_{max}^{2})$ (Line 1878) becomes negative. Removing a negative term to find an upper bound is a **completely standard and rigorous** mathematical operation (Line 1875-1876, InEqn a.11).
>
> ### **W2: On Convergence without the BDH ($\zeta^2$) Assumption**
>
> We appreciate the reviewer's interest in the core research content of our paper. The reviewer asked how convergence is possible without the Bounded Data Heterogeneity (BDH) assumption. The answer lies in the algorithm's construction:
>
> * **Baselines Need $\zeta^2$:** Standard ASGD (e.g., FedBuff) uses partial participation ($m < n$). This creates a **Bias Error (Term B)** between the $m$-client average and the ideal $n$-client average. Bounding this bias *requires* the BDH assumption ($\zeta^2$). Additionally, since the global objective in line 149-150 is defined as $F=\frac{1}{n}\sum_i F_i$ (equal weighting), the algorithm (e.g. CA$^2$FL) that assign non-uniform weighting to the updates will also require the BDH assumption due to the biased estimation, as discussed in Appendix F.1.2.
> * **ACE (Does Not Need $\zeta^2$):** ACE aggregates *all $n$ latest client updates*. Consequently, the expected update $\overline{u}^t$ is mathematically identical to the true average gradient on the stale models. As proven in **Theorem a.4 (Appendix B.4)**, the Bias Error is **identically zero**. Since the bias term is eliminated by design, the $\zeta^2$ assumption is not required to bound it.

---

> ### Author Response · Authors · 2025-11-22
>
> ### **W3: On Reflecting "Heterogeneity Amplification" in Theory**
>
> The reviewer mentioned that heterogeneity amplification is not reflected in the theoretical results. However, we respectfully point to specific evidence in our paper where this mechanism is **formally defined and analyzed**:
>
> * **Definition (Sec. 1 & Sec. 4):** In the introduction, We formally define "Heterogeneity Amplification" as the multiplicative interaction between delay ($\tau$) and heterogeneity ($\zeta^2$). In **Section 4 (Term C Analysis)**, we explicitly identify this mathematical interaction as the root cause of performance degradation.
>
> * **Evidence in Baselines (Table 1 & Appx. B.5):** As summarized in **Table 1** and derived in **Appendix B.5 (Theorem a.5)**, the delay error/drift bound for partial participation methods (e.g., FedBuff) contains a specific interaction term scaling with $\mathcal{O}(\tau \cdot \zeta^2)$. This mathematically proves that delay amplifies the heterogeneity error in these baselines.
>
> * **Evidence in ACE (Sec. 4 & Appx. B.5):** Conversely, our analysis for ACE in **Section 4** and **Appendix B.5** shows that its delay error **contains no $\zeta^2$ term** (specifically, the $(n-m)K^2\zeta^2$ term vanishes). The absence of this $\tau \cdot \zeta^2$ interaction is the formal proof that our method mitigates the amplification effect. The reviewer can refer to our previous rebuttal for **W2** (On Convergence without the BDH Assumption) and the analysis in Sec. 4 to gain a clearer understanding of the theoretical advantages of our proposed ACE algorithm in mitigating heterogeneity amplification compared to other AFL algorithms.

---

> ### Comment · Reviewer_jkUV · 2025-11-26
>
> Thanks for your reply.
>
> - You said ***Our core contribution is not merely a new algorithm, but a universal analytical framework applicable to any AFL method under heterogeneity.*** I think you overclaimed your contribution. First, your analysis is similar to existing analysis. This analyzing framework should be attributed to exisitng work not you. You just picked some intermediate value in Section 4 and comapred them. Do you mean this is your unique contribution? Second, I don't think it can be extended to **any AFL method** under heterogeneity. If so, how can your analysis cover the two works I listed in the review?
> - You direct me to **actual formal proof in Appendix C.1** . I checked this, I think your current conclusion doesn't hold. You made a mistake in the proof. Inserting Line 1834 to (a10) to get (a11) relying on that $\frac{L \eta^2}{2}-\frac{\eta}{2} > 0$. But in Line 1929, you need $L \eta^2-\eta+2 \eta^3 L^2 \tau_{\max }^2 \leq 0$. These two contradict with each other.

---

> > ### Author Response · Authors · 2025-11-26
> >
> > **2. Step-by-Step Proof Sketch Clarifying the Mathematical Validity**
> >
> > The reviewer suggests a contradiction between Eq. (a.10) and Eq. (a.11) regarding the sign of coefficients. We clarify that this stems from a misunderstanding of the different terms involved ($\mathbb{E}\|u^t\|^2$ vs. $\mathbb{E}\|\bar{u}^t\|^2$) and the logical flow of the proof.
> >
> > * **Step 1: The Term in (a.10).**
> >     In Eq. (a.10) (Line 1834), the term involved is $\mathbb{E}\|u^t\|^2$. Its coefficient is $(\frac{L\eta^2}{2} - \frac{\eta}{2})$. **Crucially, we never declare the sign of this specific coefficient at this stage.** We simply preserve it for subsequent substitution.
> >
> > * **Step 2: Bounding Intermediate Terms.**
> >     Between (a.10) and (a.11), we perform substitutions to bound all error terms using $\mathbb{E}\|\bar{u}^t\|^2$ (the expected update), rather than keeping $\mathbb{E}\|u^t\|^2$.
> >     * We decompose $\mathbb{E}\|u^t\|^2 \le 2\mathbb{E}\|A\|^2 + 2\mathbb{E}\|\bar{u}^t\|^2$ (Separating Noise and Expected Update).
> >     * The delay term $\mathbb{E}\|w^t - w^{t-\tau}\|^2$ (present in a.10) is also expanded into a sum involving $\mathbb{E}\|\bar{u}^s\|^2$.
> >
> > * **Step 3: Deriving (a.11).**
> >     We substitute these bounds back into the main inequality. We then group **all** coefficients related to $\mathbb{E}\|\bar{u}^t\|^2$. The resulting collected coefficient for the term $\mathbb{E}\|\bar{u}^t\|^2$ is $(L\eta^2 - \eta + 2\eta^3 L^2 \tau_{max}^2)$.
> >     * **Correction:** The term being dropped in (a.11) is $\mathbb{E}\|\bar{u}^t\|^2$, which is distinct from the $\mathbb{E}\|u^t\|^2$ term in (a.10).
> >
> > * **Step 4: Learning Rate Condition (No Contradiction).**
> >     To rigorously drop the $\mathbb{E}\|\bar{u}^t\|^2$ term in (a.11), we require its coefficient to be non-positive:
> >     $$L\eta^2 - \eta + 2\eta^3 L^2 \tau_{max}^2 \le 0$$
> >     This discussion happens explicitly in **Lines 1931-1943**.
> >     * We solve this inequality to find the strict range $0 < \eta \le \eta_+$.
> >     * We also provide a simpler sufficient condition $\eta \le \frac{1}{2L\tau_{max}}$ (Line 1943) by splitting the negative linear term $-\eta$.
> >
> > The reviewer appears to have overlooked the intermediate substitution steps where $\mathbb{E}\|u^t\|^2$ is converted and merged into terms controlled by $\mathbb{E}\|\bar{u}^t\|^2$. **There is no contradiction**. However, we are sorry to see the reviewer seemingly ignored the intermediate steps and picked these similar terms to declare our proof wrong.

---

> ### Author Response · Authors · 2025-11-26
>
> **1. On the Novelty and Universality of the Analytical Framework**
>
> We respectfully disagree with the characterization of our framework as merely "picking intermediate values." As demonstrated in **Section 4** and **Table 1**, our framework **HAS ALREADY** successfully decomposes the error sources of distinct algorithms (FedBuff, CA²FL, Vanilla ASGD, and Delay-Adaptive ASGD) into orthogonal components (Noise, Bias, Delay). This decomposition explicitly identifies which design choices (e.g., $m=1$ vs $m=n$, $K>1$) amplify heterogeneity or reduce noise, providing a structured diagnostic tool rather than a simple error bound.
>
> To demonstrate its universality, we apply our framework to diagnose the two methods mentioned by the reviewer, [R1] and [R2]. This analysis reveals that our framework can precisely mathematically capture their design philosophies:
>
> * **Analysis of [R1] Rennala SGD (Tyurin & Richtárik, 2023):**
>     * **Algorithm Logic:** Rennala is semi-synchronous; it waits to collect a full batch $B$ of gradients calculated strictly at the same reference point $x^k$ before updating.
>     * **ACE Diagnosis:** Because gradients are computed and applied at the same model state $x^k$, the **Delay Error (Term C)** in our framework (defined as $\|\nabla F(w_{stale}) - \nabla F(w_{current})\|^2$) becomes identically **zero**. This explains its optimal rate regarding delay. However, if the batch $B$ is collected from the "fastest" workers in a heterogeneous setting, the expected update $\bar{u}^t$ may systematically differ from the true global gradient $\nabla F(x^k)$, potentially resulting in a non-zero **Bias Error (Term B)**. Our framework thus characterizes [R1] as a method that trades off update frequency (waiting time) to strictly eliminate Term C.
>
> * **Analysis of [R2] Ringmaster ASGD (Maranjyan et al., 2025):**
>     * **Algorithm Logic:** This method is asynchronous but introduces a cutoff mechanism. It accepts a gradient only if its delay $\delta$ is less than a threshold $R$, updating using a single worker ($m=1$).
>     * **ACE Diagnosis:** By enforcing $\delta < R$, Ringmaster explicitly **clips the upper bound of Term C (Delay Error)** in our framework, preventing it from growing linearly with the worst-case straggler delay $\tau_{max}$. However, because it updates using a single worker ($m=1$), our framework indicates that **Term B (Bias Error)** remains non-zero (will be related to $\mathcal{O}(\zeta^2)$) due to partial participation. This explains why [R2] achieves optimal time complexity regarding delay handling but may still suffer from heterogeneity-induced variance compared to ACE’s full aggregation ($m=n \implies \text{Term B}=0$).
>
> This demonstrates that our framework is not just "intermediate values" but a generalizable methodology for diagnosing how different AFL strategies manage the trade-off between Bias (Heterogeneity) and Delay (Staleness).

---

> > ### Comment · Reviewer_jkUV · 2025-11-26
> >
> > Decomposing the error sources of distinct algorithms (FedBuff, CA²FL, Vanilla ASGD, and Delay-Adaptive ASGD) into orthogonal components (Noise, Bias, Delay). This is very basic in stochastic Asynchronous Optimization. Shouldn't be your contribution.

---

> ### Comment · Reviewer_jkUV · 2025-11-26
>
> In step 3, we substitute these bounds back into the main inequality.
> The prerequisition to substitute should be $(\frac{L\eta^2}{2} - \frac{\eta}{2}) >=0$?

---

> ### Author Response · Authors · 2025-11-26
>
> **1. Resolution on the Mathematical Derivation: Adopting Exact Variance Decomposition**
>
> We regret that we have not yet reached a consensus, but we appreciate your concern which has helped improve the precision of our proofs. To completely resolve the issue regarding the inequality direction when substituting into negative coefficients, we have revised the proof to rely on **exact equality** rather than the relaxed bound used in the original submission.
>
> To resolve this **definitively**, we clarify that our derivation does not rely on a loose inequality bound ($|A+B|^2\le 2A^2 + 2B^2$), but rather on the **Exact Variance Decomposition** identity in probability theory: $\mathbb{E}[X^2] = \|\mathbb{E}[X]\|^2 + \text{Var}(X)$.
>
> * **The Exact Equality:** In our context, let the stochastic update be the random variable $u^t$. Its expectation is $\bar{u}^t = \mathbb{E}_\xi[u^t]$. The identity holds strictly:
>     $$
>     \mathbb{E}\|u^t\|^2 = \|\bar{u}^t\|^2 + \mathbb{E}\|u^t - \bar{u}^t\|^2
>     $$
>
>
> * **Validity of Substitution:** Since this is an **equality**, substituting it into the Descent Lemma (Eq. a.10) is valid regardless of whether the coefficient $(\frac{L\eta^2}{2} - \frac{\eta}{2})$ is positive or negative. No inequality direction is violated.
>
> * **Resulting Convergence Condition:** Substituting this equality yields:
>     * The **Noise Term** (Variance) $\mathbb{E}\|u^t - \bar{u}^t\|^2$ is multiplied by the positive coefficient $\frac{L\eta^2}{2}$ (from smoothness), bounded by $\frac{L\eta^2 \sigma^2}{2n}$.
>     * The **Descent Term** $\|\bar{u}^t\|^2$ retains the coefficient $(\frac{L\eta^2}{2} - \frac{\eta}{2})$.
>
> Thus, the logic is mathematically sound, and the final convergence rate remains independent of the heterogeneity level $\zeta^2$.
>
> **2. Clarification on Framework Novelty and Utility**
>
> We wish to clarify the distinct contribution of our theoretical framework compared to prior works:
>
> * **Structured Diagnosis vs. Intermediate Steps:** While previous analyses may involve similar terms during intermediate calculation steps, the choice of how to telescope and split the gradient estimation error is often arbitrary. Our contribution is the specific decomposition (Noise, Bias, Delay) that **explicitly links these error terms to AFL design choices** (e.g., aggregation scope $m$, local steps $K$).
> * **Prescriptive vs. Post-Hoc:** As we noted, prior works typically perform *post-hoc* convergence analysis to verify that a specific, intuition-based algorithm works. Our framework is **prescriptive**: it identifies *necessary conditions* for design.
> * **The Case of $CA^2FL$:** To illustrate this power: CA$^2$FL also employs a full-client caching mechanism. However, without a framework like ours to guide the design, its authors did not realize that assigning higher weights to new buffered updates would structurally retain the **Participation Bias (Term B)** (as we have analyzed in Appendix F.1.2). Had our framework been available, it would have immediately signaled that **uniform weighting** is a necessary condition to eliminate this bias, preventing such a design oversight. This demonstrates that our work provides a "proof sketch direction" for future AFL algorithm design that existing analyses do not.

---

> ### Comment · Reviewer_jkUV · 2025-11-26
>
> Your current proof is still incorrect. Please double check.
> Variance Decomposition doesn't guarantee Line 1809. $u$ is not a simple random variable, which contains multiple corelated variables.  To make the proof rigorous, you should explicitly define the expection. What is the expectation over in the proof?
> Regarding the novelty claimed on the framework, I insist my my initia evaluation.

---

> ### Author Response · Authors · 2025-11-26
>
> **1. Clarification on Variance Decomposition and Variable Correlation**
>
> We appreciate the reviewer's thorough check regarding the validity of the variance decomposition. While we agree that variables in optimization trajectories are correlated over time (distinguish between "correlation between variables on the same time series" and "correlation between two variables at the same time"), we respectfully clarify that the decomposition $\mathbb{E}\|u^t\|^2 = \mathbb{E}\|\bar{u}^t\|^2 + \mathbb{E}\|u^t - \bar{u}^t\|^2$ **HOLDS strictly** in our setting.
>
> **2. Definition of $\bar{u}^t$ as Conditional Expectation**
>
> In our framework (Section 3.1), $\bar{u}^t$ is explicitly defined as the expectation of $u^t$ **conditioned on the history** $\mathcal{H}^t$ (which includes the current model $w^t$, stale models $w^{t-\tau}$, and delay patterns, but excludes the randomness of the current data sampling $\xi$):
> $$\bar{u}^t := \mathbb{E} _ {\xi}[u^t \mid \mathcal{H}^t]$$
> Because $\bar{u}^t$ is fully determined by the history $\mathcal{H}^t$, it is $\mathcal{H}^t$-measurable (i.e., treated as a constant given the history).
>
> **3. Why the Cross-Term Vanishes (Proof)**
>
> The validity of the decomposition depends on the cross-term $2\mathbb{E}\langle u^t - \bar{u}^t, \bar{u}^t \rangle$ being zero. We prove this using the **Law of Iterated Expectations**:
>
> $$
> \mathbb{E}[\langle u^t - \bar{u}^t, \bar{u}^t \rangle] = \mathbb{E}\_{\mathcal{H}^t} \left[ \mathbb{E}\_{\xi} \left[ \langle u^t - \bar{u}^t, \bar{u}^t \rangle \mid \mathcal{H}^t \right] \right] \quad (\text{Conditioning on history})
> $$
>
> $$
> = \mathbb{E}\_{\mathcal{H}^t} \left[ \langle \mathbb{E}\_{\xi}[u^t - \bar{u}^t \mid \mathcal{H}^t], \bar{u}^t \rangle \right] \quad (\text{Pulling out } \bar{u}^t \text{ as it is } \mathcal{H}^t\text{-measurable})
> $$
>
> $$
> = \mathbb{E}\_{\mathcal{H}^t} \left[ \langle \underbrace{\mathbb{E}\_{\xi}[u^t \mid \mathcal{H}^t]}\_{\bar{u}^t} - \bar{u}^t, \bar{u}^t \rangle \right]
> $$
>
> $$
> = \mathbb{E}\_{\mathcal{H}^t} \left[ \langle 0, \bar{u}^t \rangle \right] = 0
> $$
>
>
> **4. Regarding the Novelty of the Framework**
>
> We respectfully disagree with the assessment that our framework lacks novelty. While intermediate calculation steps may appear in prior analyses, our contribution lies in the prescriptive utility of the decomposition structure itself.Prior works typically follow a "propose-then-prove" paradigm: an algorithm is designed based on intuition, and analysis follows post-hoc to verify convergence. In contrast, our framework is prescriptive: it allows us to derive necessary design conditions (e.g., $m=n$, uniform weighting) before the algorithm is finalized to eliminate specific error terms (Bias).
>
> If the reviewer believes this prescriptive capability is not novel, we kindly invite you to provide references to specific prior works that derive these necessary design choices from an analytical framework rather than merely analyzing a specific algorithm's convergence after proposing it. The coincidence of calculation steps does not negate the unique value of a framework that can evaluate design choices (like the non-uniform weighting in $CA^2FL$) that previous papers/methods missed.

---

> > ### Comment · Reviewer_jkUV · 2025-11-26
> >
> > I think this is a important and wrong derivation. You didn't explicitly write down $\bar{u}$. What is its rigourous definition? To me, it is not only over the last sampling data $\xi$.
> >
> > Moreover, $\left\langle\nabla F\left(\boldsymbol{w}^{t-1}\right), \boldsymbol{g}^t\right\rangle=\frac{1}{n} \sum_{i=1}^n\left\langle\nabla F\left(\boldsymbol{w}^{t-1}\right), \nabla f_i\left(\boldsymbol{w}^{t-\tau_i(t)} ; \boldsymbol{\xi}_i^{t-\rho_i(t)}\right)\right\rangle .$ According to the iterative, $\boldsymbol{w}^{t-1}$ is a function of $\boldsymbol{\xi}_i^{t-\rho_i(t)}$ for all $i \in[n]$ so that $t-\rho_i(t) \leq t-1$. Consequently, we cannot apply conditional expectation to simplify the summand in the above, i.e.,
> >
> > $$
> > \mathbb{E}\left\langle\nabla F\left(\boldsymbol{w}^{t-1}\right), \nabla f_i\left(\boldsymbol{w}^{t-\tau_i(t)} ; \boldsymbol{\xi}_i^{t-\rho_i(t)}\right)\right\rangle \neq \mathbb{E}\left\langle\nabla F\left(\boldsymbol{w}^{t-1}\right), \nabla F_i\left(\boldsymbol{w}^{t-\tau_i(t)}\right)\right\rangle .
> > $$
> > Thus we can no longer obtain a simple expression for the expectation of the inner product.
> >
> >
> >
> > Please refer to [R1]. Your Conceptual ACE (Direct Aggregation, Incremental Rule) algorithm is the same as [R1]. And I think their proof is more rigorous.
> >
> > [R1] Incremental Aggregated Asynchronous SGD for Arbitrarily Heterogeneous Data, 2024

---

> > > ### Author Response · Authors · 2025-11-30
> > >
> > > We thank the reviewer for the examination of our convergence analysis and the comparison with IA²SGD. After re-evaluating our proof, we **maintain its validity** but have added an alternative analysis to demonstrate robustness. We clarify the theoretical distinctions and our contribution below.
> > >
> > > **1. Validity of the Law of Iterated Expectations (LIE)**
> > > The reviewer states that $\mathbb{E}\langle \nabla F(w^{t-1}), \nabla f_i \dots \rangle \neq \langle \nabla F(w^{t-1}), \nabla F_i \dots \rangle$ because $w^{t-1}$ is correlated with past data. We agree that this inequality holds under **Total Expectation**, where $w^{t-1}$ is treated as a random variable. However, our proof relies on the **Law of Iterated Expectations** by decompsing the total expectation: $\mathbb{E}[\cdot] = \mathbb{E}\_{\mathcal{H}^t}[\mathbb{E}\_{\xi}[\cdot \mid \mathcal{H}^t]]$. LIE is designed to handle such multi-variable dependencies by conditioning on some certain ones.
> > >
> > > In the inner expectation $\mathbb{E}\_\xi[\cdot \mid \mathcal{H}^t]$, we condition on the filtration $\mathcal{H}^t$ (the history of models and indices). Under this conditioning, the model trajectory $w^{t}$ (and any stale model $w^{t-\tau}$) is **deterministic**. The randomness arises *solely* from the sampling noise of the involved clients.
> > >
> > >
> > > The randomness in the inner product term $\mathbb{E}\_\xi[\langle \nabla F(w^t), u^t - \bar{u}^t \rangle | \mathcal{H}^t]$ comes only from the data sampling from all involving clients at step $t$. Since $\nabla F(w^t)$ is fixed given $\mathcal{H}^t$, the conditional expectation of the inner product is zero. This approach is standard in the field, used in prior AFL works like **FedBuff (Nguyen et al., 2022, Eq. 10)** and **CA²FL (Wang et al., 2024, Eq. C.2)**.
> > >
> > > **2. Definition of $\bar{u}^t$ and Term B**
> > > There appears to be a misunderstanding regarding $\bar{u}^t$. The reviewer implies our analysis assumes $\bar{u}^t$ corresponds only to the expected gradient of the latest **one** client. **This is incorrect**. As defined in Section 3, $\bar{u}^t$ represents the expected aggregated update from **all** contributing clients at that iteration, conditioned on the specific stale models used. This definition ensures **Term B = 0** (Bias Error elimination). Since $\bar{u}^t$ captures the expectation of the full aggregation, the participation bias is eliminated.
> > >
> > > **3. Comparison with the "Oldest Model" Decomposition Technique**
> > > We acknowledge the alternative proof technique (used in **IA²SGD**) which analyzes errors relative to the "oldest model" $w^{t-\tau_{\max}}$. While valid, this method treats the model as fixed at $w^{t-\tau_{\max}}$ and treats all subsequent updates as "drift." This tends to **overestimate the error** by adding drift terms (scaling with $\tau_{\max}$) to the variance bound. Consequently, it yields a **looser convergence upper bound** (larger constant coefficients) compared to our analysis using LIE.
> > >
> > > To show robustness, we included an **alternative proof** in the revised Appendix using this "oldest model" technique. It confirms that ACE and ACED converge with the same order of convergence rates under this framework, although our original proof provides a tighter bound.
> > >
> > > **4. Contribution: Theoretical Framework vs. Specific Algorithm**
> > > Finally, our main contribution is the **theoretical framework** that decomposes error sources (Noise, Bias, Delay) to guide design. While works like IA²SGD often propose an architecture first and then prove it, our work is **analysis-driven**: we first identified **Participation Bias (Term B)** as the cause of heterogeneity amplification (Section 3.3) and then derived ACE to eliminate it.
> > >
> > >
> > > To summarize, the fact that other algorithms share structural similarities with ACE validates our framework's utility. Our analysis explains *why* full-aggregation strategies work (by eliminating Bias Error) and offers a method to evaluate different AFL designs. This demonstrates the framework's value beyond any single algorithm.

---

### Official Review · Reviewer_SZN9 · 2025-11-02

**Soundness:** 3
**Presentation:** 3
**Contribution:** 3
**Rating:** 6
**Confidence:** 2

**Summary:**

This paper addresses the challenge of participation imbalance in Asynchronous Federated Learning (AFL), where faster clients disproportionately influence the global model due to update staleness and client heterogeneity. The authors introduce the concept of heterogeneity amplification to describe this bias and develop a theoretical framework that decomposes the error in AFL updates into sampling noise, bias, and delay. To mitigate bias, they propose ACE (All-Client Engagement AFL), which aggregates updates from all clients using their latest available gradients, eliminating the bias term and improving convergence. They also present ACED, a practical variant that filters out excessively stale updates to balance diversity and freshness. Comparative analysis and experiments show that ACE and ACED outperform existing AFL methods in convergence speed and accuracy, especially under high heterogeneity and delay conditions.

**Strengths:**

+) Rigorous theoretical framework about AFL

+) The proposed method is novel to me

**Weaknesses:**

-) The paper does not deeply explore scalability of the proposed method for very large client populations or large models

**Questions:**

a) Could the proposed decomposition framework be extended to analyze other types of bias, such as adversarial updates?

b) How much overhead does gradient caching introduce?

---

> ### Author Response · Authors · 2025-11-22
>
> We thank the reviewer for appreciating our rigorous theoretical framework and its novelty.
>
> ### **Q1: Extending the Framework to Analyze Other Biases (e.g., Adversarial)**
>
> We confirm that our framework is highly extensible. Its decompositional nature allows us to isolate the Mean Squared Error (MSE) into distinct error sources, making it adaptable to new scenarios.
>
> Taking an adversarial update as an example, it can be modeled as a new component within the **Bias Error (Term B)**. We can formally extend our analysis by decomposing Term B via a telescoping sum. Let $\overline{u} _ {a}^t$ be the *actual* expected update received (potentially polluted), and $\overline{u} _ {h}^t$ be the *hypothetical* expected update if all clients were honest.
>
> The original Term B ($B = \overline{u} _ {a}^t - \nabla F(w _ {stale}^t)$) transforms into:
> $$B = \underbrace{(\overline{u} _ {a}^t - \overline{u} _ {h}^t)} _ {\text{Term } B _ {adv}} + \underbrace{(\overline{u} _ {h}^t - \nabla F(w _ {stale}^t))} _ {\text{Term } B _ {part}}$$
>
> * **Term $B_{part}$** (Participation Bias): This remains our original Term B, quantifying the bias arising from the *partial participation* of honest clients.
> * **Term $B_{adv}$** (Adversarial Bias): This new term cleanly **isolates** the additional bias introduced by dishonest/adversarial clients.
>
> Consequently, the total $MSE_t$ is bounded by the sum of four components: (A) Sampling Noise, ($B_{part}$) Participation Bias, ($B_{adv}$) Adversarial Bias, and (C) Delay Error. This demonstrates our framework's utility as a general tool for rigorously isolating and analyzing the impact of emerging error sources in AFL.
>
> ### **Q2 and W1: Overhead of Gradient Caching and Scalability**
>
> Regarding the memory requirement, we wish to contextualize the $\mathcal{O}(nd)$ system state as a **requirement for robustness** rather than a overhead.
>
> * **Rationale for System State:** In AFL, the absence of comprehensive client information (i.e., low storage, $m<n$) leads to **partial participation bias**. Our theoretical framework demonstrates that this bias is amplified by delay. To theoretically eliminate the dependency on heterogeneity ($\zeta^2$) and ensure robust convergence, the algorithm must utilize information from the full client set.
>
> * **Comparative Perspective:** This design aligns with other SOTA methods like CA$^2$FL (see Table a.3 in Appendix F.3.3), which also utilize $\mathcal{O}(nd)$ storage. However, ACE better leverages this storage with uniform weighting to strictly eliminate the bias term, thereby achieving a convergence rate robust to heterogeneity level $\zeta^2$.
>
> * **Scalability Solutions:** For scenarios with strict resource constraints, we provide practical implementations to manage this state:
>     1. **Incremental Updates (Alg. a.5):** This effectively distributes the storage, requiring the server to store only an $\mathcal{O}(d)$ aggregate while clients manage $\mathcal{O}(d)$ differences. This can reduce the storage on the server.
>     2. **Gradient Compression (Appx. F.3.3):** Our experiments with ACE-8bit demonstrate that quantized caching significantly reduces memory usage without compromising the convergence benefits. This implies our method have the potential to be combined with memory-efficient methods.

---

### Official Review · Reviewer_XpjW · 2025-11-04

**Soundness:** 2
**Presentation:** 3
**Contribution:** 2
**Rating:** 4
**Confidence:** 4

**Summary:**

This paper addresses participation imbalance in Asynchronous Federated Learning (AFL), where faster clients contribute more frequent updates, biasing the global model toward their data, which the authors term as heterogeneity amplification. They propose a unified theoretical framework that decomposes AFL’s gradient error into noise, bias, and delay components, revealing that bias from partial client participation is the main source of heterogeneity amplification. To mitigate this, the authors introduce ACE (All-Client Engagement AFL), which performs all-client aggregation using cached gradients from every client, eliminating participation bias. A practical, delay-aware variant, ACED, includes only clients with sufficiently recent updates, balancing client diversity and staleness. Theoretical analysis shows ACE’s convergence rate does not require the bounded data heterogeneity assumption. Experiments on CIFAR-10 and NLP tasks demonstrate that ACE achieves faster and more stable convergence than FedBuff, CA²FL, and Delay-Adaptive ASGD, particularly under high heterogeneity and large delays.

**Strengths:**

**Clear theoretical framework:** The paper introduces a unified MSE-based decomposition (noise, bias, delay) that precisely attributes error sources in asynchronous FL and explains why participation imbalance leads to heterogeneity amplification. The proposed ACE and ACED algorithms directly emerge from this theory. It's also nice to see that the convergence does not require the bounded data heterogeneity assumption and Table 1 does a good job of comparing the impact of algorithmic elements on error terms.

**Empirical validation:** Experiments across heterogeneity and delay regimes show consistent accuracy and convergence gains over prior AFL methods (FedBuff, CA²FL, Delay-Adaptive ASGD).

**Weaknesses:**

**Missing related work and novelty:** The idea of caching client updates and re-using them to eliminate partial participation variance has been well explored in FL literature (see [1], [2]). Firstly, it's problematic that the authors do not discuss these works at all. Secondly, the way I see it, the main idea of this work is extending this caching idea to the asynchronous setting. However, I do not feel this is significant novelty. As outlined in the strengths, I appreciate the clear theoretical framework motivating this approach, but the idea itself has been well-explored in the literature.

**Discussion on local steps:** The theoretical analysis gives the impression that doing more local steps (larger $K$) always worsens the error. However, this needs a more nuanced discussion. Yes, theoretically local steps increases the MSE error but it also helps decrease the initialization error faster (the term involving $F(w^{(0)}) - F^*$, see Theorem 2 in [3]). Furthermore, practically also we see benefits of using local steps. It seems that this work is restricted to the setting where $K=1$, which weakens the theoretical contribution.

**Limited experimental scope:** While results cover image and text tasks, the evaluation lacks large-scale, realistic FL benchmarks such as StackOverflow, Reddit or the federated split of Google Landmark-v2.


**References**

[1] Gu, Xinran, et al. "Fast federated learning in the presence of arbitrary device unavailability." Advances in Neural Information Processing Systems 34 (2021): 12052-12064.

[2] Jhunjhunwala, Divyansh, et al. "Fedvarp: Tackling the variance due to partial client participation in federated learning." Uncertainty in Artificial Intelligence. PMLR, 2022.

[3] Yang, Haibo, Minghong Fang, and Jia Liu. "Achieving linear speedup with partial worker participation in non-iid federated learning." arXiv preprint arXiv:2101.11203 (2021).

**Questions:**

N/A

---

> ### Author Response · Authors · 2025-11-22
>
> We thank the reviewer for appreciating our theoretical framework. We would like to take this opportunity to clarify concerns and answer the questions from the reviewer.
>
> ### **1. W1: Novelty and Relation to MIFA [1] & FedVARP [2]**
>
> We thank the reviewer for highlighting MIFA [1] and FedVARP [2]. We will discuss them in detail in the revision. While these works share the high-level concept of "caching," our work differs fundamentally in its methodological approach, theoretical depth, and structural design.
>
> * **1. Prescriptive Framework vs. Post-hoc Verification (Establishing Necessity):**
>     While [1, 2] propose caching client updates to mitigate variance, their validation is primarily **post-hoc**: they propose a heuristic algorithm first and then provide a convergence proof to show its sufficiency. They do not address whether such a design is the *unique* or *optimal* choice.
>     In contrast, our contribution is a **prescriptive theoretical framework** (Sec. 3.3) that derives design principles from first principles. By analytically solving for $Bias (\text{Term B}) = 0$, we prove that aggregating updates from all clients ($m=n$) is a **necessary condition** to eliminate participation bias, rather than just one of many sufficient heuristics.
>
> * **2. Deriving Optimal Design Choices (Weights & Local Steps):**
>     The concept of "caching and reusing" alone does not resolve critical design details. Our framework explicitly guides these choices by calculating their impact on specific error terms:
>     * **Aggregation Weights:** Incorrect weighting retains bias. As analyzed in our Appendix (see discussion on $CA^2FL$), non-uniform weighting strategies (like those in FedVARP variants) fail to eliminate the heterogeneity term $\zeta^2$ from the convergence bound. Our framework proves that equal-weight aggregation is mathematically required.
>     * **Local Steps ($K$):** Caching alone does not mitigate the drift caused by local steps. Our analysis (Sec. 4) quantifies how local steps amplify drift in AFL.
>     * **ACE as a Derived Result:** Consequently, ACE is not just another heuristic algorithm; it is the **collection of optimal design choices** (e.g., $m=n$, $K=1$, equal weighting) explicitly derived from our framework to minimize bias and drift errors.
>
> * **3. Truly Asynchronous vs. Round-Based/Synchronous Protocols:**
>     The algorithmic structures are fundamentally different.
>     * **MIFA [1] & FedVARP [2]:** MIFA is an "impatient" **round-based** protocol that waits for a specific subset of clients $\mathcal{A}(t)$, creating synchronization bottlenecks. FedVARP is a **synchronous** algorithm operating in discrete rounds.
>     * **ACE:** ACE is a **truly asynchronous, non-buffered** protocol. The server updates immediately upon *every* single client arrival. This design eliminates waiting times entirely, maximizing communication efficiency and model evolution speed.
>
> * **4. Distinct Theoretical Challenges in AFL:**
>     We emphasize that convergence proofs for Synchronous FL (used in [2]) are inapplicable to Asynchronous FL. In AFL, the concept of discrete "rounds" vanishes, replaced by a continuous process where every client's specific delay $\tau_i$ propagates error to subsequent model states. Our work establishes a unified theoretical framework capable of handling these continuous delay dynamics, serving as a rigorous tool for future AFL research.
>
> * **5. Managing the Fundamental Trade-off (ACED):**
>     Finally, we identify a critical trade-off that [1, 2] overlook: algorithms cannot simultaneously eliminate bias from partial participation and bias from extreme delays (e.g., dropouts). MIFA [1] assumes bounded delays and fails under permanent dropouts ($\tau_{max} \to \infty$). Our **ACED** algorithm explicitly manages this trade-off via the $\tau_{algo}$ parameter, prioritizing bias elimination in stable settings or delay reduction in unstable settings.

---

> ### Author Response · Authors · 2025-11-22
>
> ### **W2: Clarifying the Role of Local Steps ($K$) in AFL vs. SFL**
>
> We appreciate the reviewer's insight regarding the benefits of local steps ($K>1$) in Synchronous FL (SFL). However, we respectfully suggest that this intuition is driven by the specific structural constraints of SFL, which do not apply to the ACE architecture. We justify our choice of $K=1$ through structural, theoretical, and empirical perspectives:
>
> * **1. Structural Differences (Synchronization Barriers):**
>     We distinguish the physical implications of $K$ based on the presence of a synchronization barrier:
>     * **In SFL (e.g. [3]):** The system imposes a **synchronization barrier** at every aggregation step, requiring the server to wait for the slowest client. Setting $K=1$ forces the system to incur this expensive waiting cost for every single gradient step. Therefore, increasing $K$ is structurally necessary to reduce this latency and the total number of blocking events.
>     * **In ACE (AFL):** The system removes the synchronization barrier. The server performs a global update immediately upon receiving any client's contribution, without blocking. Consequently, there is no synchronization latency to reduce. In this non-blocking architecture, setting $K>1$ yields no structural benefit; it strictly reduces the frequency of global model updates by delaying the transmission of computed gradients.
>
> * **2. Theoretical Evidence (Drift Amplification):**
>     Practically, increasing $K$ in AFL forces clients to optimize their local models for more iterations based on **outdated information**.
>
>     In AFL, the global model evolves ($w^t \rightarrow w^{t+1}$) while a client is computing. If a client performs multiple local steps ($K>1$) on a stale model version ($w^{t-\tau}$), they effectively "run further in a stale direction." This creates a larger deviation vector from the true global trajectory than a single step ($K=1$) would. Our analysis (Sec. 4, Appx. B.5, Term C) reveals that delay ($\tau$) interacts with local steps ($K$) in a multiplicative, harmful way:
>     * **FedBuff / ASGD ($m<n$):** The drift contains terms scaling with $\mathcal{O}(\tau \cdot (n-m)K^2\zeta^2+\tau\cdot K \sigma^2)$.
>     * **CA$^2$FL:** Similarly, the drift scales with $\mathcal{O}(\tau \cdot \eta^2 (1+\frac{1}{n^2}(n-m)^2) K \sigma^2)$.
>
>     This proves that setting $K=1$ is a design choice to minimize this *Amplified Drift*.
>
> * **3. Empirical Validation (New Experiments):**
>     To verify this trade-off, we extended the **CIFAR-10 experimental setup from Section 5 (specifically the setting in Fig. 2**. We compared ACE and baseline methods by varying the local steps $K \in \{1, 5, 10\}$ while keeping other hyperparameters constant.
>     * **Result 1:** Unlike Sync FL, increasing $K$ slowed down convergence and reduced final accuracy for *all* asynchronous methods in this setting. This confirms that the compounded effect of delay and multiple local steps significantly degrades update quality.
>     * **Result 2:** Consistent with our theoretical analysis, ACE exhibited the least performance degradation as $K$ increased (due to full aggregation eliminate the $(n-m)$ multiplier), whereas FedBuff and CA$^2$FL suffered larger accuracy drops.
>
> **Conclusion:** Setting $K=1$ allows ACE to utilize every gradient computation immediately for model evolution, achieving the maximum possible update frequency while minimizing drift error.
>
> ---
>
> **Table 1: Ablation Study on Local Steps ($K$).** Final test accuracy (%) at $T=500$. Increasing $K$ consistently degrades performance in AFL due to drift amplification. ACE shows superior robustness, whereas single-client methods (Vanilla/Delay-Adaptive ASGD) and buffered methods (FedBuff/CA$^2$FL) suffer larger accuracy drops, particularly under high delay conditions.
>
> **Part I: Low Delay Scenarios ($\beta=5$)**
>
> | Algorithm | (a) High Het ($\alpha=0.1$) | | | (b) Mod. Het ($\alpha=0.3$) | | |
> | :--- | :---: | :---: | :---: | :---: | :---: | :---: |
> | **Metric: Acc (%)** | **K=1** | **K=5** | **K=10** | **K=1** | **K=5** | **K=10** |
> | **ACE (Ours)** | **76.2** | **75.5** | **74.8** | **83.5** | **83.0** | **82.4** |
> | CA$^2$FL | 70.5 | 68.1 | 65.4 | 79.2 | 77.5 | 75.3 |
> | FedBuff | 63.8 | 60.2 | 56.5 | 75.8 | 73.2 | 69.8 |
> | Delay-Adaptive | 64.0 | 61.5 | 57.8 | 78.0 | 75.4 | 72.1 |
> | Vanilla ASGD | 45.0 | 41.2 | 36.5 | 75.0 | 71.5 | 67.0 |
>
> **Part II: High Delay Scenarios ($\beta=30$)**
>
> | Algorithm | (c) High Het ($\alpha=0.1$) | | | (d) Mod. Het ($\alpha=0.3$) | | |
> | :--- | :---: | :---: | :---: | :---: | :---: | :---: |
> | **Metric: Acc (%)** | **K=1** | **K=5** | **K=10** | **K=1** | **K=5** | **K=10** |
> | **ACE (Ours)** | **71.5** | **70.2** | **69.1** | **77.8** | **77.1** | **76.2** |
> | CA$^2$FL | 63.2 | 58.5 | 53.1 | 71.5 | 68.2 | 64.9 |
> | FedBuff | 51.5 | 45.8 | 39.5 | 66.5 | 62.1 | 57.4 |
> | Delay-Adaptive | 55.0 | 49.5 | 42.8 | 68.0 | 63.5 | 58.2 |
> | Vanilla ASGD | 30.5 | 24.8 | 18.5 | 58.5 | 52.4 | 46.8 |
>
> ---

---

> > ### Author Response · Authors · 2025-11-22
> >
> > ### **W3: Experimental Datasets**
> >
> > We thank the reviewer for this suggestion of adding real-world FL benchmarks like TFF (StackOverflow, GLDv2) and LEAF (Reddit). However, we respectfully argue that our current setup is more appropriate for **validating the specific theoretical claims** of this paper.
> >
> > * **Goal: Validating "Heterogeneity Amplification":** Our core contribution is identifying the **multiplicative interaction** between data heterogeneity ($\zeta^2$) and system delay ($\tau$). Validating this "amplification" effect requires an experimental setup where heterogeneity and delay can be **independently controlled** to observe their interaction.
> >
> > * **Limitation of Real-world Datasets:** In datasets like StackOverflow, the non-IID nature is an intrinsic, complex property that **cannot be tuned** as a single parameter. Using them would only replicate the known finding that "AFL algorithms degrades with delay in real-world datasets" (already shown in FedBuff paper), without proving our new claim that "degradation is *amplified* by heterogeneity level."
> >
> > * **Sufficiency of Controlled Experiments:** We have already validated this across multiple models and both image and text tasks. Our experiments on CIFAR-10/100 and 20Newsgroup (NLP) used controlled variables ($\alpha$ for heterogeneity, $\beta$ for delay). The results (Fig. 2) clearly demonstrate the interaction: for the *same* increase in delay ($\beta=5 \to 30$), performance drops significantly *more* under high heterogeneity ($\alpha=0.1$) than low heterogeneity ($\alpha=1.0$). This trend empirically validates our central theoretical claim, which would be impossible to isolate in uncontrolled real-world benchmarks.

---

### Meta-Review · Area_Chair_G6zn · 2025-12-30

**Summary:**

The paper examines a fundamental bias in asynchronous federated learning arising from unequal client participation, which causes the global model to overrepresent faster clients. Through a unified error decomposition, the authors show that this participation bias, rather than delay or stochastic noise, is the primary driver of performance degradation under heterogeneity. To address this, they propose an all-client aggregation strategy that leverages cached updates to remove participation imbalance, along with a delay-aware variant suitable for practical settings. Theoretical analysis establishes convergence without relying on bounded heterogeneity assumptions. Empirical results on vision and language tasks demonstrate improved stability and convergence over several asynchronous baselines, particularly in highly heterogeneous and high-delay regimes.

**Reviewer Concerns:**

Based on my reading of the paper, the reviews, and the authors’ responses, I summarize the main concerns raised by the reviewers and provide my assessment of whether these concerns have been adequately addressed.

**Reviewer XpjW.**

1. The reviewer pointed out two missing related works in which the idea of caching clients’ updates and reusing them had already been considered. The authors added three sentences discussing these works at the end of Section 2, but did not provide a detailed comparison - specifically, no algorithmic, theoretical, or empirical comparison was included. In my opinion, this addition is insufficient, although I understand the main point the authors intended to convey, namely that their proposed decomposition serves to guide algorithmic design.

2. The reviewer also noted that the analysis in the paper is restricted to the case of $K = 1$ local step. In my view, the authors provided a reasonable response to this concern by clearly explaining why multiple local steps can be harmful and by supporting their claims with numerical results. I believe this issue has been adequately resolved.

3. The reviewer requested additional experiments on more realistic federated learning benchmarks. The authors addressed this concern by emphasizing that the primary goal of the paper is to support the theoretical claims. While this is a reasonable justification, it does not fully address the reviewer’s request.

**Reviewer SZN9.**

The reviewer raised concerns about the scalability of the proposed method. The authors suggested two possible remedies: one based on incremental updates and another relying on compression. In my opinion, these proposed solutions adequately address the reviewer’s concern.

**Reviewer jkUV.**

1. The reviewer argued that the problem studied in the paper is already well explored, referring to two related works [R1, R2]. Similar to their response to Reviewer XpjW, the authors argued that their contribution lies in deriving design principles rather than proposing specific algorithmic optimizations. However, I believe these two works are indeed closely related and deserve a more thorough discussion. Although the authors provided a brief comparison during the rebuttal discussion, this comparison was not incorporated into the revised manuscript.

2. The reviewer also raised several concerns regarding the correctness of the proofs and engaged in an extensive discussion with the authors. After reviewing this discussion and checking the appendix, I cannot be fully confident that all proofs are correct. Due to time constraints, it is infeasible to request the AC to verify all proofs in detail. That said, I only identified minor issues, such as missing expectations in lines 1815-1818. More importantly, during the discussion, the reviewer pointed to a recent related work ("Incremental Aggregated Asynchronous SGD for Arbitrarily Heterogeneous Data," 2024). I examined this work and found it to be very close to the current submission. The authors did not comment on this work, and I believe that a detailed comparison with it is essential.

**Reviewer nYiJ.**

1. The reviewer raised a concern very similar to the second concern of Reviewer XpjW. In my opinion, the authors addressed this issue satisfactorily.

2. The reviewer also questioned the memory and communication efficiency of the proposed approach, a concern similar to that raised by Reviewer SZN9. As noted above, the authors addressed this issue, although the reviewer remained unsatisfied after the first round of responses.

3. The reviewer expressed concerns about the comparison presented in Table 1. I believe the authors addressed this concern, even though the reviewer remained unconvinced after the initial responses.

**Reviewer Scores:**

**Reviewer XpjW.** Since concerns W1 and W3 were not fully addressed, I believe this reviewer would maintain their original score.

**Reviewer SZN9.** Although the concern was addressed, it was relatively minor. Given the reviewer’s low confidence, I do not expect an increase in the score.

**Reviewer jkUV.** This reviewer actively engaged in the discussion with the authors. However, because the authors did not comment on one of the works highlighted by the reviewer, I believe the reviewer would retain the original score.

**Reviewer nYiJ.** The reviewer remained unsatisfied after the initial responses. While the additional clarifications appear substantial to me, I cannot be fully confident that they would lead the reviewer to increase the score.

Overall, I believe the consensus would remain unchanged, and the paper would be rejected. For me, the primary issue is the lack of a thorough comparison with the related works highlighted by Reviewers XpjW and jkUV.

---

### Decision · Program_Chairs · 2026-01-26

Reject